# Intensified continental chemical weathering and carbon-cycle perturbations linked to volcanism during the Triassic–Jurassic transition

Jun Shen [1 ✉], Runsheng Yin [2], Shuang Zhang[3], Thomas J. Algeo [1,4,5], David J. Bottjer[6], Jianxin Yu [4 ✉], Guozhen Xu[1], Donald Penman[7], Yongdong Wang [8 ✉], Liqin Li[8], Xiao Shi[9], Noah J. Planavsky [10], Qinglai Feng[1] & Shucheng Xie[4]

Direct evidence of intense chemical weathering induced by volcanism is rare in sedimentary successions. Here, we undertake a multiproxy analysis (including organic carbon isotopes, mercury (Hg) concentrations and isotopes, chemical index of alteration (CIA), and clay minerals) of two well-dated Triassic–Jurassic (T–J) boundary sections representing high- and low/middle-paleolatitude sites. Both sections show increasing CIA in association with Hg peaks near the T–J boundary. We interpret these results as reflecting volcanism-induced intensification of continental chemical weathering, which is also supported by negative mass-independent fractionation (MIF) of odd Hg isotopes. The interval of enhanced chemical weathering persisted for ~2 million years, which is consistent with carbon-cycle model results of the time needed to drawdown excess atmospheric $CO_2$ following a carbon release event. Lastly, these data also demonstrate that high-latitude continental settings are more sensitive than low/middle-latitude sites to shifts in weathering intensity during climatic warming events.

[1] State Key Laboratory of Geological Processes and Mineral Resources, China University of Geosciences, 430074 Wuhan, Hubei, P.R. China. [2] State Key Laboratory of Ore Deposit Geochemistry, Institute of Geochemistry, Chinese Academy of Sciences, 550081 Guiyang, Guizhou, P.R. China. [3] Department of Oceanography, Texas A&M University, College Station, TX 77843, USA. [4] State Key Laboratory of Biogeology and Environmental Geology, China University of Geosciences, 430074 Wuhan, Hubei, P.R. China. [5] Department of Geology, University of Cincinnati, Cincinnati, OH 45221-0013, USA. [6] Department of Earth Sciences, University of Southern California, Los Angeles, CA 90089, USA. [7] Department of Geosciences, Utah State University, Logan, UT 84321, USA. [8] State Key Laboratory of Palaeobiology and Stratigraphy, Nanjing Institute of Geology and Palaeontology, and Center for Excellence in Life and Paleoenvironment, Chinese Academy of Sciences, 210008 Nanjing, Jiangsu, P.R. China. [9] College of Earth Sciences, Jilin University, Changchun, 130061 Jilin, P.R. China. [10] Department Geology and Geophysics, Yale University, New Haven, CT 06520-8109, USA. ✉email: shenjun@cug.edu.cn; jianxinyu@cug.edu.cn; ydwang@nigpas.ac.cn

Eruptions of large igneous provinces (LIPs) are regarded as triggers of several major biocrises in Earth history[1]. However, the causal links between LIP volcanism, environmental perturbations (e.g., carbon-isotope excursions, chemical weathering, oceanic anoxia, and acidification), and biotic turnover (e.g., mass extinction) remain poorly understood. The trigger of the ~201 Ma Triassic–Jurassic (T–J) boundary biocrisis, one of the "Big Five" Phanerozoic mass extinctions[2], is widely thought to have been the eruption of massive lava flows and emplacement of sills linked to the Central Atlantic Magmatic Province (CAMP)[3,4]. The CAMP is a LIP that produced $2–3 \times 10^6$ km$^3$ of magmatic deposits across eastern North America and adjacent areas of Pangea during a ~600-kyr interval[2]. CAMP volcanism, which triggered numerous environmental perturbations associated with the T–J boundary mass extinction, is regarded as the ultimate cause of this biocrisis[2–6]. It is inferred to have emitted large quantities of isotopically light carbon as carbon dioxide and/or methane to the atmosphere, thus leading to negative carbon isotope excursions (CIEs) in both inorganic and organic reservoirs at a global scale[4,6–9]. Increased $CO_2$ concentrations in the atmosphere[10,11] contributed to climatic warming, oceanic anoxia[12], seawater acidification[13,14], and intensified chemical weathering on land during the Early Jurassic[15–17].

Continental chemical weathering acts as a potential link between volcanism and marine environmental perturbations[18–21]. Enhanced chemical weathering, e.g., due to elevated temperatures and $CO_2$ concentrations linked to volcanism, can: (1) increase riverine nutrient fluxes to the ocean, stimulating surface productivity and marine anoxia[1,22]; and (2) lower atmospheric $CO_2$ concentrations[21]. Earlier studies provided some evidence of intensified chemical weathering around the T–J boundary based on Os isotopes[15,23,24], chemical index of alteration (CIA) data[25], palynological assemblages[17], and clay-mineral abundances[16,26]. However, the links between volcanism, carbon isotope excursions, and continental chemical weathering are largely inferential and have not been robustly demonstrated to date mainly owing to a lack of suitable volcanic proxies in sedimentary successions.

Mercury (Hg) concentrations and isotopes are widely used to track volcanic[5,27–29] and terrestrial inputs[27,30,31] to ancient depositional systems. Elemental mercury has a low vapor pressure that renders it susceptible to volatilization, leading to atmospheric dispersal. It has relatively short residence times in the Earth-surface system (~0.5–2 yr in the atmosphere, and a few hundred years in seawater), making it ideal for recording geologically short ($<10^3$ yr) Hg-emission events[32]. During large volcanic eruptions, such as LIPs, normal buffering mechanisms can be overwhelmed by massive Hg inputs via atmospheric transport, leading to Hg enrichments in diverse facies globally[28,33,34]. However, terrestrial materials (e.g., plants, soils) can also have elevated Hg concentrations that may serve as a significant source of Hg to shallow-marine and lacustrine facies[27,30–32]. Furthermore, mass-independent fractionation (MIF) of odd Hg isotopes (i.e., $\Delta^{199}$Hg) can be used to identify certain processes (e.g., photochemical reduction) that influence Hg cycling (see review by Blum et al.[35]). $\Delta^{199}$Hg values are near-zero for direct volcanic emissions from the deep Earth, distinguishing them from terrestrial and atmospheric fluxes, which generally show negative and positive $\Delta^{199}$Hg values, respectively (see review by Blum et al.[35]). Reservoir-specific MIF values can be used to interpret Hg sources, as has been done for sections of the Toarcian OAE[31], the Permian–Triassic boundary[27,28,30], and the T–J boundary[5,36].

Although earlier T–J boundary studies have analyzed Hg concentrations in both marine and terrestrial sections (Fig. 1), these records are largely concentrated around central Pangea. However, data from the eastern margin of the Tethys Ocean, which would enable a truly global-scale overview, are lacking.

Because sedimentary Hg enrichments can be affected by different sources and depositional processes[30,31,37], additional constraints are needed, such as those provided by Hg isotopes. To date, Hg isotopic data have been generated for the T–J boundary in only two studies of three marine sections[5,36] (Fig. 1). However, terrestrial ecosystems are generally more responsive than marine ecosystems to continental volcanic effects (especially climatic warming[38]), making the investigation of Hg isotopes in terrestrial T–J boundary sections promising.

Here, we examine cause-and-effect relationships among volcanism, chemical weathering, and the global carbon cycle during the T–J transition based on an integrated multiproxy analysis of two terrestrial T–J boundary sections in western China, i.e., the high-latitude Haojiagou (HJG) site in Xinjiang Autonomous Region, and the low/middle-latitude Qilixia (QLX) site in Sichuan Province (Fig. 1, Supplementary Note 1). We use Hg isotopes to determine the dominant sources of Hg to the study sections, organic carbon isotopes ($\delta^{13}C_{org}$) to monitor carbon-cycle disturbances, as well as CIA and clay-mineral assemblages to evaluate weathering intensity changes during the T–J transition (Supplementary Data 1). We use the Long-term Ocean-Atmosphere-Sediment CArbon cycle Reservoir (LOSCAR) Model to quantify the relationship of carbon-cycle changes to atmospheric $CO_2$ concentrations and silicate weathering rates. Our analysis provides new insights into cause-and-effect relationships among volcanism, chemical weathering, and the global carbon cycle during the T–J transition.

## Results

In the HJG section, organic carbon isotopes ($\delta^{13}C_{org}$) show relatively uniform background values of −25‰ to −23‰ with a pronounced negative shift to −27‰ near the T–J boundary (at 30–70 m) (Fig. 2a). Around the T–J transition (−50–240 m), a total of three negative carbon isotope excursions (CIEs) are present below (at −20 to 20 m, 1–2‰ in magnitude), close to (at 30–70 m, 3–4‰ in magnitude), and above the T–J boundary (at 100–170 m, 3–4‰ in magnitude). Mercury concentrations (Hg) range from 3 to 101 ppb and show large sample-to-sample variations. A few spikes (e.g., >80 ppb) are present at 90–100 m and 170–190 m (Fig. 2b). From background values of <50 ppb/wt.% (at −180–10 m), the ratio of mercury to total organic carbon (Hg/TOC) rises to >100 ppb/wt.% (at 10–50 m) near the T–J boundary (Fig. 2c), and some smaller Hg/TOC peaks (e.g., 50–100 ppb/wt.%) are present in the Lower Jurassic (at 50–160 m). $\Delta^{199}$Hg shows negative values (−0.3‰) in the uppermost Triassic (at −180-10 m) with a shift toward slightly less negative values (at 10–50 m) near the T–J boundary and distinctly higher values (> - 0.20‰) in the Lower Jurassic (at 130–430 m) (Fig. 2d). There is a positive excursion in the CIA from 75–85 (mean value of 80 ± 3) in the Upper Triassic (at −180 to 10 m) to 85–95 though the T–J transition interval (at 10–180 m), followed by a return to lower values (55–75, mean value of 70 ± 6) in the upper part of the section (at 200–470 m; Fig. 2e). The kaolinite content shows a steady and slight increase near the T–J boundary and above it (at 10–180 m). Although it is unclear, the smectite content increases abruptly in the Lower Jurassic Sinemurian stage (at 200–450 m). However, other clay minerals [e.g., chlorite, illite, and illite-smectite mixed layers (I/S)] show limited variation through the section (Fig. 2f).

In the QLX section, around the T–J transition (280–345 m), the $\delta^{13}C_{org}$ profile exhibits Upper Triassic background values of −27‰ to −24‰, followed by three negative excursions to −26‰ to −27‰ that are located below (at 280–300 m, 1–2‰ in magnitude), close to (310–320 m, 2–3‰ in magnitude), and above the T–J boundary (325–345 m, 2–3‰ in magnitude) (Fig. 2g).

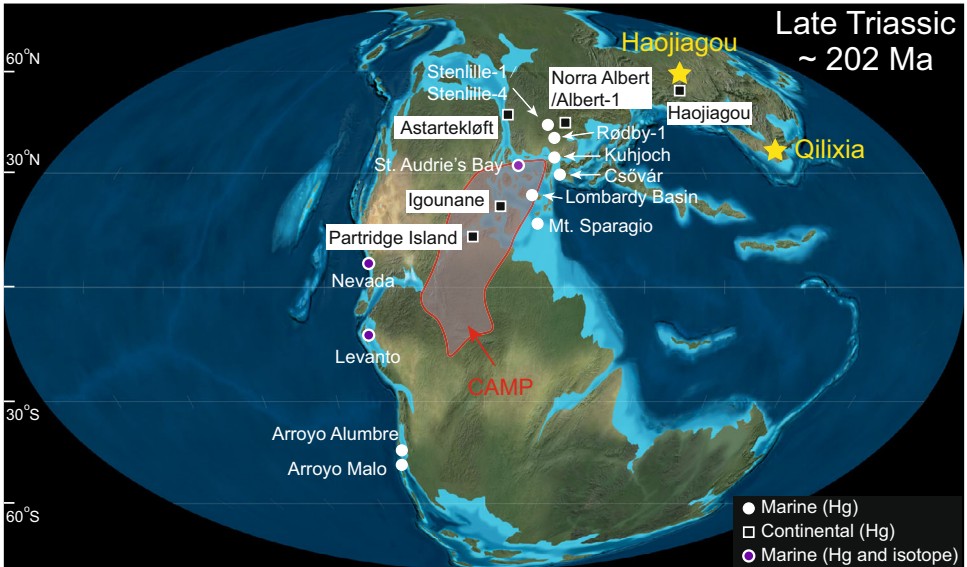

**Fig. 1 Geobal paleogeography of the Late Triassic (~200 Ma).** Adapted from Ron Blakey, http:// https://deeptimemaps.com/, © 2016 Colorado Plateau Geosystems Inc. Yellow stars represent the study sites, including. The Haojiagou section (~60°N, high latitude) on the North China Craton, and the Qilixia section (~30–40°N, low/middle latitude) on the South China Craton. Circles and squares represent other marine and continental sections, respectively, for which mercury data have been generated, including Hg concentrations and isotopes for Nevada[5, 36], St. Audrie's Bay[34, 36], and Levanto[36]; and Hg concentrations for Arroyo Malo[34], Astartekløft[34], Igounane[34], Kuhjoch[34], Partridge Island[34]; Stenlille-1/Stenlille-4[46], Rødby-1[46], Norra Albert/Albert-1[46], Csővár[47], Arroyo Alumbre[9], Lombardy Basin[36], Mt. Sparagio[36], and Haojiagou[48].

Mercury contents are lower (<400 ppb) in the Upper Triassic (at 0–290 m), except for a spike (to 611 ppb) at 60.62 m. Higher Hg concentrations (>400 ppb) are observed in the Uppermost Triassic to Lower Jurassic (at 290–345 m) (Fig. 2h). Multiple Hg/TOC peaks of >200 ppb/wt.% (max. 1218 ppb/wt.%) are present in the uppermost Triassic to lowermost Jurassic (at 300–345 m), compared to generally low background values (<200 ppb/wt.%) in the Upper Triassic (at 0–295 m) (Fig. 2i). $\Delta^{199}Hg$ background values are relatively higher at QLX (0‰ to +0.1‰, 0–295 m) than at HJG. Negative $\Delta^{199}Hg$ excursions from ~0‰ to −0.3‰ are present below (~300–310 m) and at the T–J boundary (~320 m) (Fig. 2j). CIA increases from background (0–310 m) values of 71–90 (mean 80 ± 4) to 82–92 (mean 85 ± 3) near the T–J boundary and above it (310–345 m) (Fig. 2k). The kaolinite content increases significantly near the T–J boundary (31 ± 12%, at 290–345 m) from low background values (<15%, at 0–290 m), whereas the contents of other clay minerals (e.g., chlorite, illite, and I/S) decrease from the Upper Triassic to the Lower Jurassic (Fig. 2l).

## Discussion

Carbon-isotope (inorganic and organic) and mercury records in sedimentary successions have been widely used to infer volcanism during the T–J transition. The CAMP coincided with ~3–6‰ negative CIEs in both carbonates and organic matter, which have been used to constrain its onset and duration[4,6,9,39,40]. The sources of isotopically light carbon are inferred to have been volcanic gases and/or thermogenic gases generated through magmatic intrusions into organic-rich strata[4] or dissociation of seafloor methane clathrates[7]. Atmospheric $CO_2$ concentrations during the T–J transition increased by a factor of 2–4× relative to pre-T–J boundary values (i.e., from 1000–2000 ppm to 2000–4000 ppm[11,41,42]). The presence of negative CIEs in carbonate and organic carbon isotope profiles of both marine and terrestrial T–J boundary sections serves to demonstrate the global extent of the underlying carbon-cycle perturbations[4,6–9,39,40]. Furthermore, these emissions of carbon-based greenhouse gases

had deleterious bio-environmental effects: (1) higher temperatures—various marine and terrestrial proxies suggest an average global temperature rise of 3–4 °C[10,11,41], (2) increased wildfire frequency in terrestrial habitats, attributed to climatic warming[43–45], and (3) oceanic acidification due to higher carbon dioxide concentrations in seawater[13,14].

Mercury, another promising proxy for volcanic inputs to the sediment, has been analyzed in various marine and continental T–J boundary sections ($n = 17$) that were mostly geographically proximal to the CAMP[5,9,34,36,46–48] (Fig. 1). Zero or near-zero MIFs associated with elevated Hg enrichments near the T–J boundary in marine facies representing a range of water depths support volcanic sources of Hg, thus linking these records to the CAMP[5,36]. Furthermore, multiple Hg peaks near the T–J boundary at many sites are indicative of episodic eruptions of the CAMP[5,9,34,36,46,47].

In this study, organic CIEs in the HJG and QLX sections are consistent with volcanism-induced carbon-cycle perturbations to terrestrial environments (Fig. 2a, g). Organic carbon isotopic variation in terrestrial systems is complex and can be influenced by many factors, such as carbon sources, $pCO_2$-dependent fractionation, and diagenesis[49]. The magnitude of the negative carbon isotope excursions near the T–J transitions ranges from 1‰ to 8‰ in various depositional settings[6,9]. The most significant negative excursions are (1) the "Initial CIE" close to the T–J boundary (ICIE), and (2) the "Main CIE" in the lowermost Jurassic (MCIE), although a small (~ 1‰) "Precursor CIE" is also present in the uppermost Triassic (PCIE) of the two study sections (Fig. 2a, g). The distributions of these CIEs in T–J boundary sections globally[6,8,9], is strong evidence that they are primary signals. Local environmental influences resulted in, at most, a limited overprint due to the weak correlation between carbon isotopes and TOC. Although we do not know the exact cause for the large negative excursions of carbon isotopes in the background interval (e.g., at ~ 110 m and 150 m for QLX, and at 260–300 m for HJG), they were likely due to other factors[49]. Commonalities between the CIEs of the two study sections (i.e.,

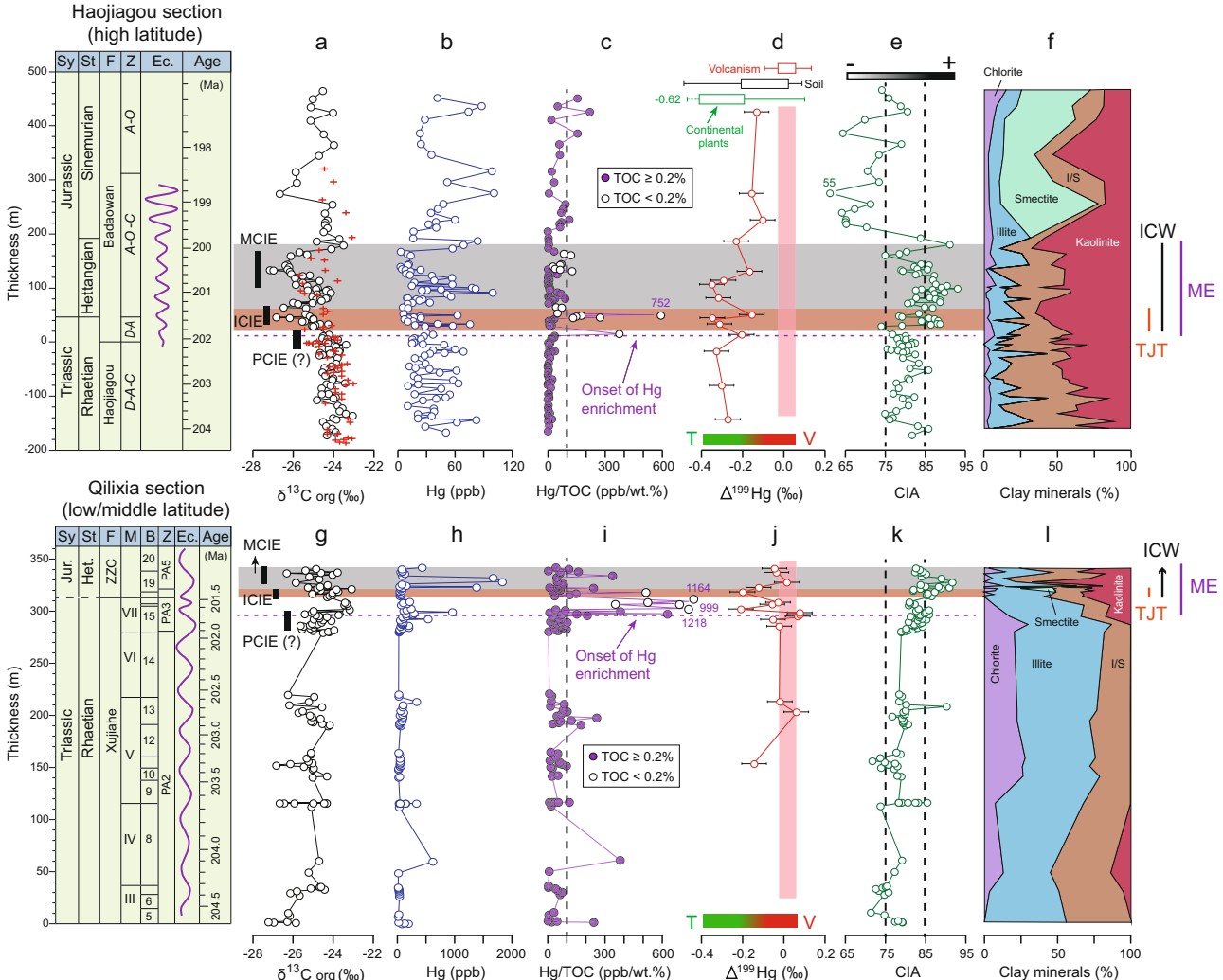

**Fig. 2 Profiles of Haojiagou (Upper, HJG) and Qilixia (Lower, QLX) sections.** (**a**, **g**) Organic carbon isotope ($\delta^{13}C_{org}$, ‰); (**b**, **h**) Mercury concentrations (Hg, ppb); (**c**, **i**) Ratios of mercury to total organic carbon (Hg/TOC, ppb/wt.%); (**d**, **j**) Mass independence fractionation of odd-Hg isotope ($\Delta^{199}Hg$, ‰); (**e**, **k**) Chemical index of alteration (CIA) and (**f**, **l**) clay minerals. The eccentricity cycle and ages are from Sha et al.[53] and Li et al.[70] for Haojiagou and Qilixia respectively. The red crosses in represent $\delta^{13}C_{org}$ data from Sha et al.[53]. Open and purple filled circles in c and i represent samples with TOC < 0.2 wt.% and ≥ 0.2 wt.%, respectively. ICW Intense chemical weathering interval; TJT Triassic–Jurassic transition; ME mercury-enriched interval. The red and green bar at the base of column d and j represent the volcanic (V) and terrestrial (T) compositions of $\Delta^{199}Hg$ respectively. The arrow for the ICW represents the uncompleted records in the Early Jurassic for QLX. The horizon bars of the $\Delta^{199}Hg$ profiles represent standard deviation (2σ) values. Abbreviations: Sy System, St (sub)stage, F formation, B bed, Z sporomorph assemblage zone, Ec. eccentricity cycle, M member, Jur. Jurassic, Het. Hettangian, ZZC Zhenzhuchong, ICIE Initial carbon isotope excursion, MCIE Main carbon isotope excursion, PCIE precursor carbon isotope excursion. Note: full geochemical data are in Supplementary Figs. 2–3. Source data are provided as a Source Data file.

multi-phased character and similar magnitudes) are consistent with the hypothesis that they were the product of releases of large amounts of [13]C-depleted carbon to the atmosphere by volcanism and related processes.

Profiles of both raw Hg (Fig. 2b, h) and normalized concentrations (Hg/TOC) (Fig. 2c, i) for the two study sections document increased inputs of Hg above background levels around the T–J transition. Owing to its strong affinity for the organic fraction of sediments, Hg concentrations are generally reported on a TOC-normalized basis[50], although other minerals (e.g., sulfides, clays) can be the dominant host of Hg[37,51]. In the present study sections, Hg is hosted mainly by organic matter at QLX (Supplementary Note 2), validating the use of TOC-normalized Hg concentrations (i.e., Hg/TOC) in this section. The TOC values for samples yielding higher Hg/TOC (e.g., >100 ppb/wt.%) are greater than the threshold value of 0.2 wt.%[52] for all but

six samples, and five of these samples have TOC values ≥ 0.15 wt.% (Fig. 2i). Significantly, both low-TOC and high-TOC samples yield high Hg/TOC values (Fig. 2i, Supplementary Fig. 3d), and, thus, the pattern of the Hg/TOC profile at QLX would not change significantly if these six samples were excluded.

At HJG, the dominant host of Hg is uncertain owing to non-significant correlations with proxies for organic matter, sulfides, and clay minerals (see Supplementary Note 2). Despite this uncertainty, we chose to utilize TOC rather than TS or Al for Hg normalizations, which has the advantage of maintaining equivalency of data display with the QLX section as well as many earlier Hg studies[29,37]. The variability in Hg/TOC peaks near the ICIE at HJG may be partly due to low-TOC values (Fig. 2c), and, given the low sampling density of this section (116 samples in ~650 m strata), it is possible that we inadvertently failed to sample the T–J boundary beds with the highest TOC content. A higher-

resolution study of HJG (61 samples in ~90 m strata) yielded Hg/TOC peaks for the samples with high TOC values (e.g., ≥0.20 wt.%) around the T–J boundary[48] (Supplementary Fig. 2e), although the pattern of secular variation is much the same as in the present study. Given the similarity of Hg/TOC records from both this and earlier studies[48], it appears that the Hg enrichment interval around the T–J boundary at HJG has been reliably identified in the present study as well.

The Hg/TOC peak associated with the ICIE at HJG and QLX has an equivalent in other T–J boundary sections[5,34,36,46]. This observation suggests a major increase in actively cycled Hg in conjunction with global-scale volcanism at the T–J boundary[29]. The onset of Hg enrichment (ME) coincided with the PCIE (i.e., below the ICIE), similar to the Hg record from the Arroyo Alumbre section[9], likely due to linkage between the carbon cycle and intrusive emplacement of dikes and sills before the main eruptive stage of CAMP magmatism[3]. The presence of several Hg/TOC peaks (e.g., >100 ppb/wt.%, Fig. 2c, i) near the T–J boundary of the study sections implies multiple stages of CAMP volcanism, as has been inferred from coeval marine and terrestrial records[5,34,36,46].

Hg isotopes in the study sections provide evidence of atmospheric transport (of presumably volcanically sourced) Hg during the T–J transition (Figs. 2d, j and 3a, b). The $\Delta^{199}$Hg/$\Delta^{201}$Hg slope is between 1.0 and 1.36 for both sections (Fig. 3c),

consistent with photoreduction of aqueous Hg(II) driven by natural dissolved organic matter[35]. Below the T–J transition, the $\Delta^{199}$Hg values are lower at HJG (–0.4‰ to –0.2‰) compared to QLX (–0.2‰ to –0.05‰), suggesting a higher proportion of terrestrially-sourced Hg (e.g., from plants and soil), probably derived from abundant coal deposits in the Upper Triassic of HJG[53,54]. The $\Delta^{199}$Hg values during the mercury-enriched interval exhibit significant increases compared to background levels in both study sections, from lower values in the Upper Triassic (–0.4‰ to –0.2‰ for HJG, and –0.2‰ to –0.05‰ for QLX) to higher values around the T–J boundary (–0.2‰ to –0.1‰, and –0.05‰ to +0.05‰, respectively) (Fig. 2d, j). This pattern, implying an increased flux of volcanically sourced Hg (with near-zero $\Delta^{199}$Hg) through the atmosphere (purple dashed arrow in Fig. 3a, b), has a counterpart in less negative $\Delta^{199}$Hg signals (e.g., from ≤–0.24‰ to ≥–0.17‰) in shallow-marine T–J sections[5,36]. However, the present study sections exhibit lower $\Delta^{199}$Hg values near the T–J transitions than deep-water marine sites[5,36], which were influenced by the variations of terrestrial inputs (Fig. 3a, b).

Increased continental chemical weathering is associated with many episodes of LIP volcanism in Earth history[1]. Previous Os-isotope studies of the T–J transition documented two distinct stages: (1) decreasing $^{187}$Os/$^{188}$Os during the Rhaetian, reflecting inputs of mantle-derived unradiogenic Os associated with

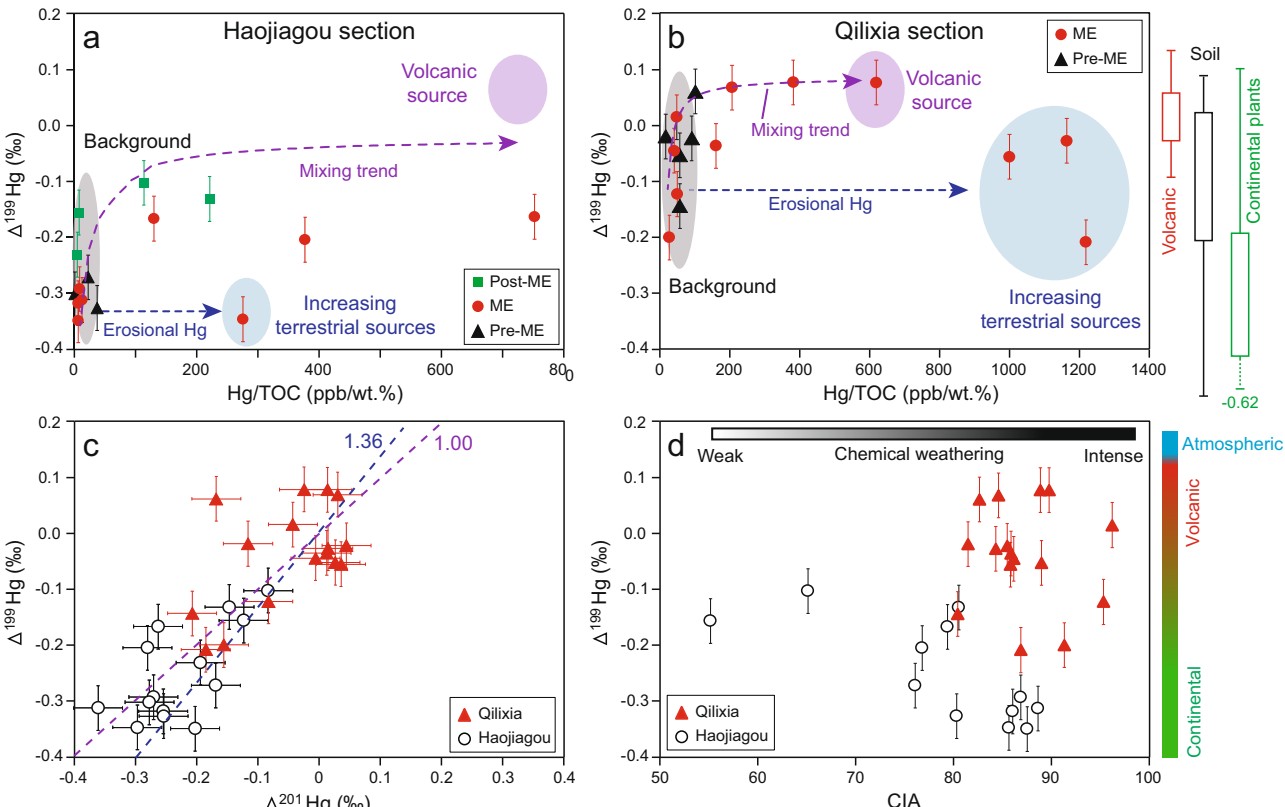

**Fig. 3 Crossplot of geochemical proxies.** $\Delta^{199}$Hg versus Hg/TOC for Haojiagou (**a**) and Qilixia (**b**) sections, as well as $\Delta^{199}$Hg versus $\Delta^{201}$Hg (**c**) and CIA (**d**) for HJG and QLX sections. The gray, purple, and blue ellipses in (**a**) and (**b**) represent the background, volcanic, and terrestrial endmember sources, respectively. The volcanic endmember is based on the most positive values of MIF ($\Delta^{199}$Hg = ~0.1‰) and most elevated Hg/TOC values (~600–800 ppb/wt.%) in the study sections, which likely reflect dominant volcanic influence. For the terrestrial endmember, we assumed values based on the maximum Hg/TOC (~300 ppb/wt.% and 1000–1200 ppb/wt.% for HJG and QLX, respectively) and $\Delta^{199}$Hg values similar to the background (~−0.3‰ to −0.4‰ and −0.2‰ to 0‰ for HJG and QLX, respectively). The purple dashed arrows in a and b represent a two-component mixing model between the background (low Hg/TOC, negative $\Delta^{199}$Hg) and the volcanic endmember (high Hg/TOC, positive $\Delta^{199}$Hg). The blue dashed arrows in a and b represent increasing terrestrial Hg inputs (high Hg/TOC, negative $\Delta^{199}$Hg) into the system. The range of $\Delta^{199}$Hg values for volcanisms, soil, and continental plants and continents are from Yin et al.[58]. The bars of the $\Delta^{199}$Hg and $\Delta^{201}$Hg distributions represent standard deviation (2σ) values. ME mercury-enriched interval. Source data are provided as a Source Data file.

emplacement of the CAMP; and (2) increasing $^{187}Os/^{188}Os$ during the Hettangian, reflecting inputs of more radiogenic Os related to increased continental weathering[15,23,24,55,56]. An increase of seawater $^{87}Sr/^{86}Sr$ from the upper Rhaetian to the Hettangian was also linked to enhanced chemical weathering and the associated elevated flux of radiogenic Sr from continents to the ocean[23]. In Nevada and Peru, shallow-marine carbonate ramp facies disappear at the T–J boundary and are replaced by siliceous sponge-dominated "glass ramp" cherts for two million years. This benthic ecosystem regime shift has been attributed to elevated dissolved silica flux from enhanced weathering of silicate rocks and soil erosion during the volcanogenic greenhouse of the Early Jurassic[17,57].

Mercury records in the two study sections document intense continental weathering during the T–J transition. Sources of Hg can be evaluated based on a multi-endmember mixing model, yielding distinct amounts and isotopic compositions for the pre-ME, ME, and post-ME intervals (Fig. 3a, b). During the pre-ME, both sections exhibit lower Hg/TOC with negative $\Delta^{199}Hg$ values, indicative of background terrestrial sources, as seen in sediments dominated by terrestrial sources of Hg[35,58]. During the ME, Hg/TOC, and $\Delta^{199}Hg$ covariation suggest two separate mixing trends with background Hg as one endmember (gray ellipse in Fig. 3a, b) and the other endmember being either atmospherically transported volcanic Hg (purple ellipse in Fig. 3a, b; characterized by higher Hg/TOC and near-zero or slightly positive $\Delta^{199}Hg$ values) or terrestrial Hg (blue ellipse in Fig. 3a, b; characterized by higher Hg/TOC and unchanged $\Delta^{199}Hg$ values to the pre-ME). Besides the volcanic sources of Hg, elevated Hg coupled with limited $\Delta^{199}Hg$ variations during the ME relative to the background sediments likely indicates a large influx of terrestrial Hg to the study sections (blue dashed arrow in Fig. 3a, b). Similar mixing trends of multiple sources of Hg were also reported from terrestrial and shallow-water settings near the Permian–Triassic boundary, the largest mass extinction of the Phanerozoic, which was associated with the Siberian Traps LIP[27,30].

The CIA profiles of the two study sections provide further evidence of intense continental chemical weathering during the T–J transition. In an A-CN-K diagram (Fig. 4), samples from each section plot approximately along a line representing the ideal weathering trend during the initial moderate weathering stage (i.e., characterized by preferential loss of Na and Ca), before

turning sharply towards the Al apex at the onset of an advanced weathering stage marked by accelerated loss of K (Fig. 4). This pattern suggests that source materials were generally consistent throughout each study section, a precondition for using CIA to track the evolution of weathering conditions in sedimentary successions (see Supplementary Note 3). CIA increases from the pre-ICW (75–85) to the ICW (85–95) in both sections, providing direct evidence of intensified continental chemical weathering during the T–J transition (Fig. 2e, k). A strong negative correlation between CIA and $\Delta^{199}Hg$ for HJG ($r = -0.66$, $n = 12$, $p < 0.05$), also supports elevated terrestrial Hg fluxes due to continental chemical weathering (Fig. 3d).

Clay-mineral compositions provide further evidence of intensified chemical weathering immediately following the extinction event. Diagenesis can alter clay-mineral assemblages in deep-time sedimentary systems[59], as evidenced by K-addition (Fig. 4) and a high percentage of illite in illite/smectite (I/S) mixed-layer minerals (mostly 80–95%) for both sections. However, discrete clay minerals (especially kaolinite and chlorite) are not commonly affected by diagenesis and can preserve climatic and weathering information[60]. With these considerations in mind, the noticeable increase in kaolinite content around the extinction interval in both study sections (e.g., mean value increases from 27 to 53% and 2 to 31% for HJG and QLX, respectively, Fig. 2f, l) and its general correspondence to elevated CIA values point to stronger chemical weathering intensity in terrestrial habitats.

The near-synchronicity of the onset of major volcanism (evidenced by elevated Hg loading), intensified chemical weathering (evidenced by CIA and clay minerals), and carbon-cycle disturbances (evidenced by $\delta^{13}C_{org}$) in the two study sections provides direct evidence of cause-and-effect relationships among these processes near the T–J boundary. Volcanism released large quantities of carbon, resulting in a major carbon-cycle perturbation at a global scale. On the other hand, volcanic-related environmental perturbations (e.g., rising temperatures) are likely to have contributed to an increase in continental chemical weathering rates, drawing down atmospheric $CO_2$ levels. More specifically, evidence for an intensification of local weathering conditions is consistent with Hg spikes at the high-latitude site (HJG), although the low/middle-latitude site (QLX) is marked by a slight delay in weathering intensification (Fig. 2). Based on the current age model (see "Materials and methods"), the offset at the

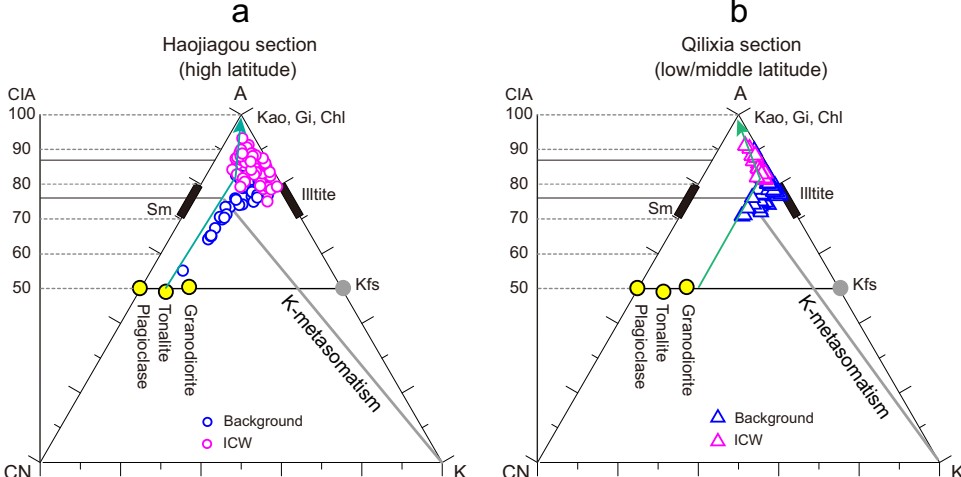

**Fig. 4 Ternary diagrams of A-CN-K for the Haojiagou. (a)** and Qilixia (**b**) sections. Purple and blue symbols represent samples from intensely chemically weathered (ICW) and background intervals, respectively. The green arrows represent weathering trends. A = Al$_2$O$_3$, CN = CaO* + Na$_2$O, K = K$_2$O; Chl chlorite, Gi gibbsite, Kao kaolinite, Kfs K-feldspar, Sm smectite. Other details as in Supplementary Figs. 2 and 3. Source data are provided as a Source Data file.

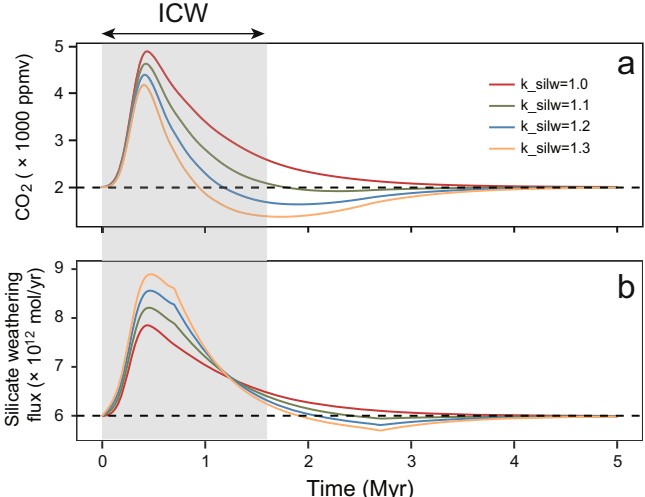

**Fig. 5 Model results of LOSCAR for the T–J boundary carbon-cycle perturbation. The weatherability (k_silw) factors were setted to 1.0, 1.1, 1.2, and 1.3. a** Atmospheric $p$CO$_2$ level. **b** Silicate weathering flux. The gray shaded rectangle represents the interval of intense chemical weathering (ICW), yielding elevated silicate weathering fluxes. The horizontal dashed lines represent the background values of atmospheric $p$CO$_2$ (**a**) and silicate weathering flux (**b**) before the CAMP eruptions.

low/middle-latitude site corresponds to a time interval of ~200 kyr (Fig. 2), suggesting a protracted shift in tropical-subtropical conditions relative to high-latitude settings. However, this pattern is consistent with predictions of more rapid temperature increases at high-latitude sites during the initial stages of warming (similar to the modern[61]). More pronounced and more rapid temperature shifts in high-latitude settings could translate into an earlier signal for increased weathering intensity (Fig. 2).

Intensified chemical weathering could draw down atmospheric CO$_2$ by silicate weathering at a million-year timescale following a major emission event such as the CAMP eruptions[11,42]. To investigate the response of the global carbon cycle to CAMP volcanogenic carbon inputs, we utilized the Long-term Ocean-Atmosphere-Sediment CArbon cycle Reservoir (LOSCAR) Model[62] (Fig. 5). LOSCAR is suitable for investigation of the effects of carbon-cycle perturbations on atmospheric CO$_2$ concentrations and silicate weathering rates at timescales ranging from centuries to millions of years[62–65]. Following the $p$CO$_2$ reconstructions for the end-Triassic[11] and the modeling practice of Heimdal et al.[4], we set the initial steady-state atmospheric $p$CO$_2$ as 2000 ppm. A total of 24,000 Gt carbon with an average $\delta^{13}$C of −18.8‰ was released in the model, following Heimdal et al.[4]. The carbon release is assumed to follow a Gaussian pattern during the 600-kyr-long eruption interval[2,3]. Given the possibility of weathering rate changes linked to eruption of highly weatherable CAMP basalts, we explored the effects of increasing the silicate weathering rate in LOSCAR with an additional weatherability coefficient (k_silw). It was set to increase linearly from 1.0 to a range of values higher than 1.0 (i.e., 1.1, 1.2, 1.3) through the 600-kyr eruption interval and then to decrease linearly back to 1.0 within 1–2 Myr following termination of volcanism. Detailed model descriptions and sensitivity tests can be found in Supplementary Note 4.

Our LOSCAR model outputs show that higher silicate weathering rates could have been maintained for ~1 to 3 million years following the eruption, depending on k_silw values (Fig. 5), which roughly agrees with the duration of the high-CIA interval at HJG (~1.6 Myr, Fig. 2e) as well as the timeframe of Os records[23] and the drawdown of CO$_2$ to background values[42]. Following the

onset of CAMP volcanism, CO$_2$ levels first rose due to continuous emissions of carbon and then decreased due to strong silicate weathering triggered by high CO$_2$ levels (as well as increased availability of basalt for weathering). Our model shows that the extra carbon sourced from CAMP was fully removed from the atmosphere by silicate weathering within ~1 to 3 million years following the onset of the volcanic event. By comparing the time frames of the atmospheric $p$CO$_2$ perturbation and the corresponding silicate weathering response (driven by variation in k_silw values) with the observed temperature record[10] and the recorded weathering anomalies (~1.6 Myr) in this study, we infer that the optimal k_silw value is between 1.0 and 1.1 (i.e., a very slight increase in weatherability, Fig. 5). This finding clashes with the view that basalt emplacement triggered a large increase in average crustal weatherability (which can also be thought of as the strength of the silicate weathering feedback), driving widespread cooling[66]. Despite the fact that CAMP resulted in large-scale (~10$^7$ km$^2$) emplacement of basalt at low to middle paleolatitudes, the silicate weathering feedback through the T–J boundary seems not to require additional changes (i.e., adding an extra k_silw term to create a stronger weathering feedback) other than its default $p$CO$_2$ dependence (i.e., higher $p$CO$_2$ leads to increased silicate weathering). This result may seem surprising, given that fresh basalts generally react faster with atmospheric CO$_2$ than other silicate rock types. However, weathering rates can be slowed by surface passivation[67] in natural settings. In other words, CAMP resulted in massive low/middle-latitude basalt emplacement[11,23] but our findings indicate that this magmatic outpouring resulted in only a minor shift in the behavior of the silicate weathering feedback immediately following the eruptions.

In this study, Hg concentrations and Hg isotopes provide the first evidence for intense volcanism from two terrestrial sections in China (far from the CAMP area) near the T–J transition. Enhanced continental chemical weathering during the latest Triassic to earliest Jurassic is evidenced by changes in Hg sources, increased CIA, and larger proportions of kaolinite in clay-mineral assemblages from both low/middle- and high-latitude continental sites in China. The underlying cause was climatic warming due to a contemporaneous rise in atmospheric carbon dioxide levels linked to the CAMP eruptions. In turn, intensified chemical weathering (more sensitive to high-latitude settings) may have been a major cause of marine oceanic perturbations leading to biotic stress and mass extinction during the Triassic–Jurassic transition. Simulations using the LOSCAR model show that intensified chemical weathering could have drawn atmospheric CO$_2$ back down to its pre-extinction background level over an interval of ~1 to 3 million years, conforming well to the time-frame for the carbon-cycle excursion and weathering anomalies identified in the study section.

## Methods

Terrestrial deposits of Late Triassic to Early Jurassic age are well developed in the inland basins (e.g., Junggar and Sichuan) of western China. Detailed palynological work has been carried out in these basins, allowing well interbasinal correlations. The two study sections are Haojiagou (HJG, Xinjiang Autonomous Region), representing a high-latitude (~60°N; note: ~200-Ma paleolatitude) lacustrine succession in the Junggar Basin, and Qilixia (QLX, Sichuan Province), representing a low/middle-latitude (~30°N) fluvial succession in the Sichuan Basin (Fig. 1).

The HJG section (base at ~43.641°N, 87.221°E, and top at 43.665°N, 87.203°E) is located on the southern margin of the Junggar Basin in the Xinjiang Autonomous Region, northwestern China (Fig. 1). The section contains >1 km thick deposits of the Upper Triassic Haojiagou and Lower Jurassic Badaowan formations, of which a ~650-m interval straddling the T–J boundary is studied here (Fig. 2, Supplementary Fig. 1). The Haojiagou Formation consists mainly of mudstone and siltstone with numerous coal and black mudstone interbeds, whereas the Badaowan Formation consists of siltstone and sandstone with a few coal and mudstone interbeds. Four sporomorph assemblages were recovered (in ascending order): the *Dictyophyllidites-Aratrisporites-Cycadopites* (D-A-C), *Dictyophyllidites-Aratrisporites* (D-A), *Alisporites-Osmundacidites-Cyathidites* (A-O-C), and *Araucariacites-*

*Osmundacidites* (A-O) assemblages (Supplementary Fig. 1). The T–J boundary extinction horizon is marked by an abrupt decrease of sporomorph diversity[53]. This section has been the subject of floral[68], sedimentological[68,69], and orbital-cyclicity studies[53], making it well suited for the present study due to near-continuous sedimentation and the availability of a high-resolution biostratigraphic-astrochronological framework[53].

The QLX section (31.197ºN, 107.744ºE), one of the most complete and well-exposed T–J boundary outcrops in the Sichuan Basin, is located ~17 km south of Xuanhan County of Dazhou City, along the Xuanhan-Kaijiang Highway. The Xujiahe Formation (~500 m thick) is composed of sandstone, siltstone, mudstone, and coal seams yielding abundant plant fossils (Supplementary Fig. 1). This formation is subdivided into seven lithologic members, with Members I, III, V, and VII composed mainly of mudstones, and Members II, IV, and VI of sandstones[54]. A high-resolution biostratigraphic[54] and astrochronological[70] study provided the framework for the present study. A recent study distinguished five palynological assemblages spanning Norian to Hettangian–Sinemurian, and assemblages PA2 and PA3 are for the Norian to Rhaetian age, PA4 for T–J transition, and PA5 for Hettangian to Sinemurian age (Fig. 2, Supplementary Fig. 1).

The age models for the study sections are well estimated based on biostratigraphic and astrochronological data from earlier studies (see Supplementary Note 1). The ~650-m-thick study interval at HJG represents an interval of ~7 Myr (204-197 Ma) based on the obliquity-cycle model of Sha et al.[53]. The ~350 m study interval at QLX represents ~3.5 Myr based on the obliquity-cycle model of Li et al.[70]. A discrete interval of intense chemical weathering (ICW) near the T–J mass extinction horizon, present at 20–180 m in the HJG section and at 310–340 m in the QLX section, is estimated to have lasted ~1.6 Myr at HJG, but only ~0.6 Myr at QLX due to the incomplete Lower Jurassic stratigraphic record of the latter (Fig. 2).

We correlated the two study sections based on the following considerations. First, detailed palynological zonations are available for both HJG and QLX. The end–Triassic extinction horizon and T–J boundary can be recognized based on abrupt floral turnovers in each section (Supplementary Fig. 1). Furthermore, the palynological assemblage at the base of the Zhenzhuchong Formation at QLX can be correlated with that at the base of the Badaowan Formation at HJG—they share many miospore genera including *Dictyophyllidites*, *Concavisporites*, *Asseretospora*, *Cyathidites*, *Chasmatosporites*, and *Quadraeculina*[54] (Supplementary Fig. 1). Significantly, the palynological assemblages in both sections record fern spore spikes composed of only a few genera that have been documented from the T–J transition globally[54]. Second, organic carbon isotope profiles for each section exhibit similar features, i.e., three negative excursions—the PCIE, ICIE, and MCIE (Fig. 2, Supplementary Fig. 1), which also permit correlations to other continental and marine T–J boundary sections[6,9]. Finally, studies of Milankovitch orbital periodicities have yielded high-resolution cyclostratigraphic frameworks for both HJG[53] and QLX[70] (Fig. 2, Supplementary Fig. 1).

Samples were trimmed to remove visible veins and weathered surfaces and pulverized to ~200 mesh in an agate mortar for geochemical analysis. Aliquots of each sample were prepared for different analytical procedures. Organic carbon isotopes ($n = 120$ and $n = 104$ for Haojiagou and Qilixia, respectively) were analyzed at the State Key Laboratory of Geological Processes and Mineral Resources, China University of Geosciences (Wuhan). Samples were reacted offline with 100% $H_3PO_4$ for 24 h at 250 °C, and then the carbon isotope composition of the generated $CO_2$ was measured on a Finnigan MAT 253 mass spectrometer. All isotope data are reported as per mille (‰) variation relative to the Vienna Pee Dee belemnite (VPDB) standard. The analytical precision is better than ±0.1‰ for $\delta^{13}C$ based on duplicate analyses.

Hg concentrations ($n = 116$ and $n = 105$ for Haojiagou and Qilixia, respectively) were analyzed using a Direct Mercury Analyzer (DMA80) at Yale University[71]. About 150 mg for siltstone samples and 100 mg for mudstone samples were used in this analysis. Results were calibrated to the Marine Sediment Reference Material MESS-3 (80 ppb Hg). One replicate sample and a standard were analyzed for every ten samples. Data quality was monitored via multiple analyses of MESS-3, yielding an analytical precision (2σ) of ±0.5% of reported Hg concentrations.

Carbon and sulfur concentrations ($n = 122$ and $n = 105$ for Haojiagou and Qilixia, respectively) were measured using an Eltra 2000 C–S analyzer at the University of Cincinnati. Data quality was monitored via multiple analyses of the USGS SDO-1 standard with an analytical precision (2σ) of ±2.5 wt.% and ±5 wt.% for reported values of carbon and sulfur, respectively. An aliquot of each sample was digested in 2 N HCl at 50 °C for 12 h to dissolve carbonate minerals, and the residue was analyzed for total organic carbon (TOC), with total inorganic carbon (TIC) obtained by difference.

Major element abundances ($n = 122$ and $n = 106$ for Haojiagou and Qilixia, respectively) were determined by X-ray fluorescence (XRF) analysis of pressed powder pellets using a wavelength-dispersive Rigaku 3040 XRF spectrometer at the University of Cincinnati. Results were calibrated using both USGS and internal laboratory standards. Analytical precision based on replicate analyses was better than ±2% for major elements.

A subset of samples ($n = 15$ and $n = 16$ for Haojiagou and Qilixia, respectively) were analyzed for Hg isotopes at State Key Laboratory of Ore Deposit Geochemistry, Institute of Geochemistry, Chinese Academy of Sciences, Guiyang per methods as reported in Shen et al.[28]. Hg isotopic results are expressed as delta (δ) values in units of per mille (‰) variation relative to the bracketed NIST 3133 Hg

standard, as follows:

$$\delta^{202}Hg = [(^{202}Hg/^{198}Hg)_{sample}/(^{202}Hg/^{198}Hg)_{standard} - 1] \times 1000 \qquad (1)$$

Any Hg-isotopic value that does not follow the theoretical mass-dependent fractionation (MDF) was considered an isotopic anomaly caused by mass-independent fractionation (MIF). MIF values were calculated for $^{199}Hg$ and expressed as per mille deviations from the predicted values based on the MDF law:

$$\Delta^{199}Hg = \delta^{199}Hg - 0.252 \times \delta^{202}Hg \qquad (2)$$

Analytical uncertainty was estimated based on the replication of the UM-Almadén secondary standard solutions and full procedural analyses of MESS-2.

A total of 82 bulk-rock and correspondent clay-fraction sample pairs were analyzed ($n = 52$ and $n = 30$ for Haojiagou and Qilixia, respectively). X-ray diffraction (XRD) analysis has been performed on both randomly-oriented bulk-rock powders and oriented clay aggregates (<2 μm) using a TD-3500 X-ray Diffractometer with CuKα radiations operating at 40 kV and 25 mA at the State Key Laboratory of Geological Processes and Mineral Resources, China University of Geosciences (Wuhan)[72]. Besides analysis upon air-drying of samples as performed for bulk-rock analysis, two additional runs including after saturation with ethylene-glycol and upon heating at 490 °C for 2 h. were also performed for clay aggregates. Identification and semi-quantifications of bulk-rock minerals were mainly based on the (001) peak of each mineral species. Major mineral species include quartz (Q, 4.26 Å), K-feldspar (Kfs, 3.24–3.25 Å), plagioclase (Pl, 3.18–3.20 Å), hematite (2.69 Å) and calcite (3.02 Å). Clay minerals were mainly identified both by their diagnostic peak series and by the peak shifts between different runs. On air-drying traces illite is recognized by the (10 Å, 5 Å, 3.33 Å) peak series which exhibit no shift after treatments; chlorite is characteristic of the (14.2 Å, 7.1 Å, 4.7 Å, 3.54 Å) peaks that upon heating reduce their intensities or even disappear; kaolinite is represented by the presence of both 7.18 Å and 3.58 Å peaks; and illite-smectite mixed-layer (I/S) is identified by peaks between 11–14 Å which shift towards left after saturation with ethylene-glycol and reduce to 10 Å after heating. Semi-quantification of clay minerals was also mainly based on their (001) peak area. Relative proportion between chlorite and kaolinite was determined by the 3.54/3.58 Å peak area ratio.

## Data availability
The authors declare that the main data supporting the findings of this study are available within the Source Data file. Additional data are available from the corresponding author upon request.

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

## Acknowledgements

The authors are grateful to Shenghui Deng for assistance with the field works, Mingsong Li for the discussions of timeframe of Qilixia section, and Linhao Fang for providing data of Haojiagou section. This research was supported by the Natural Science Foundation of China (92055201, 42072037) (J.S.), (41790454, 41688103) (Y.D.W.), and (42072009) (L.Q.L.), Strategic Priority Research Program (B) of the Chinese Academy of Sciences (XDB18000000, XDB26000000) (Y.D.W. and L.Q.L.), the State Key Laboratory of Palaeobiology and Stratigraphy (193124, 20172103, 20191103, 20192101) (J.S., Y.D.W., and L.Q.L.), 111 Project from National Bureau of Foreign Experts and the Ministry of Education of China (BP0820004) (S.C.X.) and China Postdoctoral Science Foundation Grant (2018M630888) (G.Z.X.). This work is a contribution to IGCP Project 739.

## Author contributions

J.S. conceived the study and designed it with R.S.Y., S.Z., and J.X.Y.; J.S., J.X.Y., and X.S performed the field works; J.S. performed Hg concentration, carbon isotope, and major element analyses; R.S.Y. analyzed Hg isotopes; Y.D.W. and L.Q.L worked on the biostratigraphy; S.Z. and D.P. ran the LOSCAR modeling; J.S., T.J.A., D.J.B., and N.J.P. wrote the paper with significant input from S.Z., G.Z.X., Q.L.F., and S.C.X.

## Competing interests

The authors declare no competing interests.
