## [Peer Review File · Nature Communications]

Intensified continental chemical weathering and carbon cycle perturbations linked to volcanism during the Triassic-Jurassic transitionReviewers' Comments:

Reviewer #1:

Remarks to the Author:

Dear authors,

I have read your manuscript with great interest. It is certainly an interesting study that should be published. The question, I think, is not whether CAMP emissions changed the global climate and enhanced weathering and erosion, but how and to what extent in different areas. Here, your manuscript can be an important contribution.

However, I think the manuscript itself needs some additional work. I have a series of comments and concern that I hope will help you revising your paper. You will find the detailed comments listed after this letter.

In summary, these are my main issues:

First, give credit where credit is due, and please do not just cite papers of colleagues you side with. In one instance you cite irrelevant papers. I assume, but this may be a mistake in your citation system?

Secondly. The plots of the geochemistry for the HJG locality in Fig. 2 are inconsistent with the data provided in the supplement. There are a couple of data points that are in the plots, that do not occur in the Table. In addition, according to the Supplementary Table the CIA data has been corrected, but there is no explanation anywhere in the Methods section or in the paper why this has been necessary or how they were corrected. Please make sure that the data you supply match the plots in your figures.

Thirdly, the localities are said to be well-constrained by palynology, i.e. spores and pollen, and that this allows interbasinal correlation, but this is neither discussed in the text or shown in a figure. Since the palynozonations for the two sections are not the same, it is impossible for the reader to correlate between the sections. You need to provide some sort of correlation figure and explanatory text for this. Preferably there should be some palynoevents that can be used for correlation. Although, as a palynologist, I realise that this may be very difficult to find as the two localities seem to be in different climate zones. But then again, how can you then say that they allow interbasinal correlation?

I recommend moderate revision.

Kind regards

Sofie Lindström

General comments:

Replace all hyphens in e.g. Triassic-Jurassic, 2-3 x 10⁶ with en-dash “-”.

Specific comments.

Line 37-38. I suggest that you end the sentence after “boundary”. Begin a new sentence with “We interpret this as reflecting volcanism-induced...”.

Line 49. Replace “the trigger” with “triggers”. Replace “multiple” with “several”.

Line 53. Why Tr-J and not T-J? Since the term “Tertiary” doesn't exist anymore.

Line 54-55. Well, actually not only the eruptions of lavas but also intrusive magmatic activity and degassing, see e.g. Heimdal et al. (2020).

Line 63. Replace “Higher” with “Increased”.

Line 65–66. I know you are probably limited by the number of references you can use, but it is not fair to the authors of the original papers to only refer to a review paper, so if possible add a reference or two here.

Line 68. Here is another one.

Line 70–71. Hm, yes, but I think eutrophication was already mentioned by Bond and Wignall (2014).

Line 72–75. Also, van de Schootbrugge et al. (2020), and Ahlberg et al. (2003).

Line 88. Yes, these are nice papers, but maybe you should first refer to Sanei et al. (2012), which was the first paper that suggested the link between Hg and LIPs.

Line 89. Insert “can” after “(e.g., plants, soils)” and “may” before “serve”.

Line 93. I am curious about the term “direct volcanic emissions”. As far as I know magma, and hence also lava, contain fairly low amounts of Hg unless it is assimilated from sedimentary rocks. Maybe you need to explain what you mean by “direct volcanic emissions”.

Line 98–99. What exactly do the numbers in brackets refer to? Are they the number of localities analyzed from marine and terrestrial environments respectively? If so, they are wrong and you should consult the literature again, including supplementary information.

Line 105–106. I am not sure this statement is accurate. After all, several papers have shown that the terrestrial ecosystem, although perturbed, recovers faster than the marine ecosystems, e.g. at the PTB (papers by Schneebeli), the TJB (e.g. Lindström et al. 2012). Plants are very resilient.

Line 113–116. This does not make sense compared to Fig 2. First you say that you have a pronounced negative shift near the TJB at 20–60m and then that one of the CIEs (that one) is at 30–70m.

Line 125–127. Actually, in the H₂G section kaolinite increases steadily through the Rhaetian. You should comment on this.

Line 126–128. You should comment on the pronounced increase in smectite in the Sinemurian. What is this related to?

Line 149. There are other papers dealing with the timing and onset etc of this LIP, and not all of them are in agreement with Korte and Ruhl. Unless you specifically want to add a sentence at the beginning of the paper referring to the variety of papers dealing with this subject and a statement that you are herein using the papers by Korte et al. and Ruhl et al. as your main stratigraphic framework, you should refer to some of them here.

Line 153. Why is the Korte et al. (2009) paper cited here? You mean Korte et al (2019) again?

Line 156–157. I think there have been other papers dealing with this subject before Korte et al. (2019).

Line 159. The first paper to suggest a 3-4 degree Celsius rise was McElwain et al. (1999). Please give credit where credit is due! If you need to lose a reference to squeeze in McElwain’s paper, you could do without one of Schaller’s papers here.

Line 161. Here you could also cite Marynowski and Simoneit (2009) and Petersen and Lindström (2012).

Line 162. And how about van de Schootbrugge et al. (2007)?

Line 182–183. Why is ONLY Percival et al. (2017) mentioned here? You don't think all the above mentioned papers on lines 178–179 add to the knowledge of volcanic induced Hg-loading?

Line 193. Other studies which you don't want to mention because?? There are papers by Sanei, Grasby, Bergquist, Fantasia, Jones, and myself with co-authors, and many others to choose from. For the TJB there are only two papers that deal with terrestrial Hg loading, and that is Percival et al. (2017) and Lindström et al. (2019). If you want to discuss more about the terrestrial environment and Hg, there is always also a paper on the Devonian–Carboniferous extinction by Marshall et al. 2020.

Line 197–199. Several of your high Hg/TOC peaks are the product of very low TOC values. You need to discuss this. It does not necessarily mean that they are not real, but you need to show the readers that you are aware of this.

Line 204–206. This is an oversimplification. In fact, as shown by Blum et al. (2014) there is overlap in $\Delta^{199}\text{Hg}$ in coals, plant litter, total gaseous Hg, rainfall-coal impacted, lichens, etc etc. Besides, why would there be "more" continental input from plants and soil? They are both terrestrial successions, right? So isn't it really a question of preservation of the organic matter to which the Hg is bound?

Line 210–213. Thibodeau et al. (2016) actually show rather consistent $\Delta^{199}\text{Hg}$ values with only slight variations from 0.05 ‰ to -0.05 ‰. At least from your supplementary table, none of your values in HJG comes close to that, and only a couple of values in QLX are close to but not within that range. I think you need to discuss what the cause of these differences are.

Line 226–230. The Os-isotope records are complicated and vary significantly between the epicontinental sea and Panthalassa.

Line 230–233. There are also other papers referring to Sr-isotopes, e.g. Callegaro et al. (2012) and Kovacz et al. (2020). The Kovacz et al. (2020) paper is very interesting as it suggests alternative interpretations.

Line 235–236. And here you could mention van de Schootbrugge et al. (2020) on soil erosion and weathering.

Line 238–239. Why does the Hg records record intense continental weathering? Terrestrial Hg derived from what source? You need to explain this better.

Line 254–255. There is an empty () after HJG.

Line 280. "today"

Line 296. Here you could also consult Landwehrs et al. (2020).

Line 340. I guess here you mean in this particular area, i.e. China? But this is really a repetition of what is on line 332. Why not merge these two sentences?

Line 346–347. It says, both in the abstract and here, that the sections are well-dated, and that this allows interbasinal correlation, but no correlation between them is shown in this paper. How robust is the spore-pollen stratigraphy in western China? Here, there are two quite different palynozonations from the two areas. Besides that, there are also additional papers dealing with the same interval from the two basins, presenting alternative zonations, e.g. Zhang et al. (2020) and Huang (2006). It would be good if you could comment on this.

Line 362. Sha et al. (2011) is not in the reference list.

Line 403. What is the detection limit for the instrument for Carbon and Sulfur, respectively? Some of the TOC values are very low, <0.20 wt %. In most cases it is these low TOC values that result in the large peaks in Hg/TOC, so I think it is important that you clarify this to the readers.

Line 619–620. We actually analyzed Hg from a primarily terrestrial succession also, the Norra Albert quarry and the Albert-1 core (a core drilled inside the quarry) in southern Sweden (Lindström et al. 2019).

In addition, Percival et al. (2017) also analyzed samples from the terrestrial succession at Astartekløft in East Greenland.

Comments on FIG. 2:

What are the red crosses on the Haojiagou C-isotope curve?

The C-isotope curve for Haojiagou does not look the same on the figure as when the actual values from the Supplementary table are plotted. There are values on the Figure that don't exist in the table, e.g. the neg C-isotope peak at c. -95m.

Comments on FIG. 3. There is either something wrong with your plots or with your Supplementary table, e.g. the data in the table shows values down to -0.67. Please make sure that the plots shows the correct data. Specify whether you have plotted against your CIA or Corrected CIA values.

Reviewer #2:

Remarks to the Author:

See attached PDF

Review: Intensified continental chemical weathering and carbon cycle perturbations linked to volcanism during the Triassic-Jurassic transition

Authors: Shen, J. et al.

This paper integrates a number of different datasets and methods in order to investigate the cause-and-effect relationships between CAMP, chemical weathering and the carbon cycle around the T-J boundary. They examine two new terrestrial sections (both high and low latitude) from China where various proxies can be analyzed, which allows to directly compare carbon cycle perturbations (carbon isotopes), volcanism (Hg data) and weathering (Hg, CIA and clay mineralogy). One issue with tying together the relationship between CAMP, the end-Triassic mass extinction and the associated carbon cycle perturbations, is that the various proxies are from different sections, and correlating these have proven to be difficult. For example, CAMP lavas are emplaced in continental deposits, but most C-isotopic records are from marine sections. Furthermore, there exists different interpretations of the isotope data considering the timing of carbon cycle perturbations and CAMP activity (e.g., Korte et al., 2019 vs. Lindström et al., 2017). Producing data of proxies for weathering, volcanism and carbon cycle perturbations from the same section prevents the issue of correlating, and is therefore of great importance for the end-Triassic, where these relationships are yet not fully understood. This point alone provides novelty to this contribution. In addition, they use carbon cycle modelling to test whether or not increased weathering due to the emplacement of CAMP can explain the observed drawdown of atmospheric CO₂ within the same timeframe as the observed intense weathering interval. I believe the main goal combined with the used methods and approaches, provides novelty to this contribution. Therefore, I believe this paper is appropriate for Nature Communications. However, I do believe the authors need to address some major concerns before it is ready for publication.

While the datasets and selected approach in this study are convincing, my major concern is the way the manuscript is organized and written. First of all, you have to convey more clearly in the introduction 1) what do we already know, 2) what knowledge are we missing, 3) what are you going to do to close the current knowledge gaps, and 4) why is this important. You have to be more specific in the introduction considering the work you have actually done. For example, the results-section starts with the description of carbon isotope data, but it is never mentioned in the introduction that you are presenting C isotope curves. Therefore, the importance of presenting this data is not clear at all. Also, the LOSCAR modelling suddenly appears at the end of the discussion, but this has barely been mentioned before. It is not clear why you have included this modeling. Why is it important? The way it is written now, it seems like it is just something you threw in there at the end. Furthermore, the different data and conclusions must be much better tied together. For example, in the way the discussion is written, the sections about the different proxies or methods stand completely alone. The whole point of this study is to integrate different proxies and methods in order to say something about the relationships between CAMP, chemical weathering and the carbon cycle around the T-J boundary. This would not be possible if you only presented carbon isotopes, or if you only did modeling. It is therefore a matter of reorganization, and to better convey your message. Lastly, I think the manuscript would greatly benefit from a read through by a native English speaker.

Specific comments:

Main Text

Line 33: "...organic carbon isotopes.."

Lines 33-37: please rewrite, this sentence is too long.

Line 38: "... near the T-J boundary, **reflecting** a ..."

Lines 39-40: delete "in both sections."

Lines 40-41: "... 2 million years, **which is** consistent with model results..." Also, it is not clear that these are carbon cycle model results that you have produced. This is important!

Line 41: I would use the word "lastly" instead of further.

Lines 51-52: "...volcanism, environmental perturbations (e.g., carbon-isotope excursions, chemical weathering, anoxia, and acidification), **and** biotic turnovers (e.g., mass extinctions) remain poorly...»

Lines 53-57: please rewrite, this sentence is too long. Also, the eruptions of lavas did not trigger the end-Triassic crisis, but rather the volatile emissions.

Lines 57-59: The way it reads now it seems like the synchronicity itself is considered the main trigger between CAMP and the extinction, which is not the case.

Lines 61-62: "... leading to negative carbon isotope excursions (CIEs) in both inorganic and organic matter at a global scale..."

Line 63: I would write "increased" instead of higher.

Line 64: I would write "warming" instead of greenhouse climate.

Lines 67-68. I would write "represents" or "act as" instead of "is". Also, you need to explain why this is a link.

Lines 68-72: chemical weathering can lower CO₂ through weathering? Explain why CO₂ concentrations decrease due to weathering. I think what you are trying to say in the following sentences is that the elevated CO₂ concentrations (that eventually lead to increased weathering through elevated temperatures and runoff) is sourced from volcanic emissions?

Lines 72-77: this is why your study is so important! There is already strong evidence for weathering, volcanism, and carbon cycle changes, but the problem is that the different datasets must be correlated between sections. And this has been proven quite difficult. However, the sections you study represents an opportunity to provide all these different proxies without worrying about correlations. This has to be communicated much more clearly.

Lines 79-81: I would write the sentence like this: Mercury (Hg) concentrations and isotopes are widely used to track volcanic (Thibodeau et al., 2016; Grasby et al., 2017; Shen et al., 2019a) and terrestrial inputs (Shen et al., 2019b; Them et al., 2019) in ancient depositional systems.

Lines 81-85: please rewrite, this sentence is too long.

Line 86: "... eruptions, such as LIPs,..."

Line 87: "..., leading to Hg enrichments in diverse facies globally..."

Lines 90-97: I would try to shorten this, and to make a better link between the previous sentences. The main message here is that Hg isotopes are necessary to identify the source of the Hg.

Line 98-100: I would write these sentences like this: "... boundary studies have analyzed Hg concentrations in both marine and terrestrial sections (Fig. 1), these records mainly originate from the central Pangean region. However, data from areas more distal from CAMP (e.g., the eastern margin of the Tethys Ocean) are lacking."

Line 101: rewrite "can be due to".

The introduction should end with a separate paragraph where you explain specifically what you are going to do, why it is important, and why your data and approach is novel. This paragraph should be based on what you have written previously in your entire introduction. This includes some of the arguments you make in lines 98-107, but also in lines 72-77.

Line 113: like previously mentioned, it is never written in the introduction that you are presenting C isotope curves, so the first time I read the results-section I was wondering why are you talking about carbon isotopes?

Line 115: "**A total of** three negative carbon...". Also, I think you should mention the magnitude of each CIE, so that they can be easily compared to the precursor/Marshi, initial/Spelae and main/Tilmanni CIEs observed in other sections.

Line 118: this is a detail, but Fig. 2A should appear before Fig. 2B.

Line 117-119: is the TOC measured in % or wt.% ? In the excel table, the TOC content is also described as % and not wt.%.

Lines 116-119: please rewrite this sentence.

Line 122: I would write: "**there is an excursion in the** CIA from 75-85...". Also, say if the excursion is negative or positive.

Lines 125-126: "**The** kaolinite content..."

Lines 129-130: again, I think you should say the actual magnitude of each CIE.

Line 138: "**The** kaolinite content increases..."

Line 153: there must be a better way to say this

Lines 147-173: What is the important message you want to convey with this section? I believe it is that the T-J boundary CIEs found in several sections have been interpreted to have occurred as a result of carbon emissions from CAMP. The CIEs you see in the two sections from your

study can be correlated to the other CIEs. So, this tells us that carbon cycle perturbations are connected to volcanism (i.e., CAMP). However, this message gets lost. I think you should move the part about the CIEs from HJG and QLX being comparable to the precursor/initial/main CIEs found elsewhere (lines 163-168) up to the beginning of this section (to line 147). This is what is important! How your data connects to previous data. And then you can say that the CIEs have been attributed to the release of carbon gases from CAMP, and make the connection between the CIEs and volcanism. Lines 157-162 feel out of place to me, I do not understand why this information is important and how it connects with the rest of the section.

Line 176: I would write it like this: “Mercury has previously been used to track volcanic inputs to various marine and continental Tr-J boundary sections that were”

Lines 176-182: Again, like the previous section (carbon isotope excursions), you start the section with what other people have done. Start with your data instead, this is what is important! Explain/interpret what your data means by using the data and conclusions from the other studies.

Line 184: refer to Fig. 2B, G.

Lines 185-190: “Owing to its..... valid procedure for the evaluation of volcanic Hg inputs.» This feels misplaced. Maybe move this to the methods-section or SI?

Line 191: “... of such excursions..” which excursions do you mean?

Line 193-196: I do not understand what you mean here: The 10^5 year gap between the onset of the Hg enrichment (ME) and the initial CIE is the result of early extrusive emplacement of dikes and sills before the main eruptive stage. Do you mean that the onset of Hg enrichment represents the earliest CAMP stage (the earliest sills and dykes), while the next Hg enrichments, overlapping with the initial CIE, represents the main eruptive CAMP stage? Why do you not mention that the onset of Hg enrichment actually overlaps with the precursor CIE? I would say that is very important information, which further argues for a link between carbon cycle changes and volcanism. Also, dikes and sills are *intrusive* not extrusive (line 195).

Line 198: “... Tr-J boundary of the study sections likely represent multiple...”

Lines 200-201: Do you mean in general or your data specifically? Why is the Hg presumably volcanically sourced? I assume this is what you explain further down in the section, but you cannot conclude with this before you have explained why. Give arguments as to why you think the Hg is volcanically sourced, and then you can make a conclusive statement like this. Also, you refer to Figs. 2 and 3, but these figures have a lot of information. Be specific, which parts of these figures show what you are trying to say?

Line 204: I would write “compared to” instead of “than at”.

Line 205: what do you mean by more continental Hg inputs? Do you mean that there is a higher abundance of terrestrial sourced Hg in HJG compared to QLX, or compared to a volcanic source, or both?

Line 207: I would write “compared to that of the” instead of “over” background levels.

Lines 208-209: You already presented these values (-0.4 ‰ to -0.2 ‰ for HJG, and -0.2 ‰ to -0.05 ‰ for QLX).

Line 211: "... volcanically sourced Hg (**with** near-zero $\Delta^{199}\text{Hg}$ values) to the sediments..."

212-213: This has already been written (lines 179-181).

Lines 214-222: This section seems out of place. Combine these results with the other Hg data supporting the same conclusions. Instead of separating the measured $\Delta^{199}\text{Hg}$ values and the mixing model, explain why the source is terrestrial or volcanic, based on these results combined. And then you can say that this is also supported by others (e.g., Thibodeau et al., 2016).

Lines 225-237: Again, start with your data, and then use other studies to back up your conclusions. Also, I feel like this could be substantially shortened. For example:

Increased continental weathering associated with the Tr-J boundary is documented by increasing ratios of both $^{187}\text{Os}/^{188}\text{Os}$ and $^{87}\text{Sr}/^{86}\text{Sr}$ during the Hettangian (Cohen and Coe, 2002, 2007; Kuroda et al., 2010), as well as the replacement of shallow-marine carbonates by siliceous sponge-dominated "glass ramp" cherts in Nevada (USA) and Peru (Ritterbush et al., 2014).

Lines 238-239: Why do the Hg records document intense continental weathering? I assume this is what you explain further down in the section, but you cannot conclude with this before you have explained why. Give arguments as to why you think the Hg records document intense continental weathering, and then you can make a conclusive statement like this.

Line 246: Also refer to Fig. S4.

Line 254: There is an empty parenthesis

Lines 257-262: please rewrite, this sentence is too long.

Line 258: remove "e.g."

Lines 266-282: This should be at the end of the discussion. This section reads like a conclusion. Merge this paragraph with your conclusion.

Line 269-270: "volcanism **acted** as a trigger **by releasing** large..." Also, volcanism acted as a trigger of what? Be specific.

Lines 273-282: here you talk about the fact that your data suggests that high-latitude continental settings are more sensitive to shifts in weathering intensity compared to low-latitude settings. This is interesting and important information (you even mention this in the abstract). However, I feel like this information suddenly appears (a bit out of the blue) as part of this conclusion-type paragraph. I think you should try to better explain why this is the case before you conclude.

Line 274: what do you mean by “in step”?

Lines 276-277: I would write this instead: Surprisingly, with the current age model, the offset in the low latitude site could correspond to a time interval of ~ 200 kyrs (Fig. 2), suggesting a very protracted shift in tropical conditions relative to high latitude settings.

Line 280: «today» is misspelled.

Lines 285-287: a consequence of chemical weathering is the consumption of CO₂ which is drawn down by silicate weathering? Please rewrite this sentence.

Line 288: «... weathering **was the increase in organic...**» Is it really necessary to mention organic carbon burial, since you do not focus on this in other parts of the manuscript? Maybe you can delete this sentence.

Lines 292-296: This feels misplaced. Maybe move this to the methods-section or SI?

Lines 296-299: I would write this instead: “A total of 24,000 Gt CO₂ with an average δ¹³C of -18.8 ‰ was released in the model, following Heimdal et al. (2020). The carbon release is assumed to follow a Gaussian pattern during the 600-kyr-long eruption interval (Blackburn et al., 2013; **Davies et al., 2017**). N.B.: Note that Heimdal et al. (2020) argues for the release of 24,000 Gt *carbon*, not 24,000 Gt CO₂.”

Lines 299-304: please rewrite, this sentence is too long.

Line 308: specify Fig. 2D

Lines 308-311: are you talking about your model results, or observed *p*CO₂ values? Or both? Please specify.

313: “... to 3 million years **following the onset of the** volcanic event.”

Lines 313-316: Please rewrite this sentence.

Line 320: “... how it did during the Palaeocene–Eocene Thermal Maximum (PETM; Zeebe et al., 2009).” Also, how did the silicate weathering feedback function during the PETM? This is not obvious to the reader, unless they are experts on the PETM and NAIP.

Line 322: “given that basalt have tremendous capability to sequester carbon”: please rewrite this sentence. Basalt does not directly sequester carbon, and also, the word tremendous should be replaced.

Lines 316-326: This section is confusing to me. You have previously said that there was intensified chemical weathering during the T-J boundary (as evidenced by ¹⁸⁷Os/¹⁸⁸Os, ⁸⁷Sr/⁸⁶Sr, Hg data, CIA, clay compositions), and in the abstract you say “the period of enhanced chemical weathering persisted for ~ 2 million years consistent with model results of the time needed to drawdown atmospheric CO₂ levels”, and in conclusions you say “simulations using the LOSCAR model show that intensified chemical weathering could have drawn down atmospheric CO₂ back down to its pre-extinction background level over an interval of ~1 to 3 million years”. But in lines 316-326 you say that basalt emplacement did not drive significant

shifts in weatherability, and that weathering rates are slowed down, and that there was a minor shift in the behavior of the silicate weathering feedback. I am not sure what your intention is with lines 316-326? Is it to say that you would have expected the need of even higher weathering rates? The important point is that the LOSCAR modeling suggests that high silicate weathering rates can be maintained for 1-3 million years, which agrees with the duration of the high CIA interval, and that the $p\text{CO}_2$ values are back to background levels within this time period. Right? Maybe move this part to the SI, or shorten it to a few sentences.

Line 330-331: what do you mean by provide the first evidence for intense volcanism? In general, or from terrestrial sections in China?

Line 346: you refer to the Ordos Basin, but in the following sentences you only mention the Junggar Basin (HJG section) and the Sichuan Basin (QLX section). Which section is found in the Ordos Basin?

Lines 346-347: you use the word “detailed” twice in the same sentence.

Line 353: “The section contains > **1 km thick deposits** of the Upper...”

Line 377: “are” well estimated

Line 389: replace “following which”

Line 413: “A subset **of** samples...”

Line 433: What do you mean by “extra” data? What data is this? Also, use the word “additional” instead.

Line 448: “... **performed** the field work”

Line 614: it should be yellow stars, not yellow triangles

Line 626: organic carbon isotopes should be A and F

Line 629: move all abbreviations to the bottom.

Line 631: “... and ages **are** from...”

Line 634: “...green bar **at** the base of column C and H represents the”

Line 635: “The arrow for the ICW represents the”

Line 639: spell out “ME”

Line 639: it is confusing with the red color for both the volcanic source and also for samples with mercury enrichment. Also, what do the blue and purple circles represent?

Line 641: either use both abbreviations (HJG and QLX) or spell out both section names (Haojiagou and Qilixia).

Line 644: “members”

Line 644: “into” the system instead “to”.

Line 646: “See” the web version instead of “Refer to”

Line 649: is it possible to add observed $p\text{CO}_2$ values? (for example in Schaller et al., 2011, 2012, in the supplementary material, there are tables with the $p\text{CO}_2$ values and “absolute time” in myr). It would be interesting to see the modeled $p\text{CO}_2$ curves plotted against observed values

Supplementary Information

Lines 36 -37: You mention a Dalongkou section. Is this supposed to be the Haojiagou section?

Lines 54-58: please rewrite, this sentence is too long.

Line 101: you refer to PETM boundary conditions. Do you mean that you used the paleo set-up as opposed to the modern set-up of LOSCAR? If so, you should write that instead, because it makes no sense to use PETM conditions for the T-J boundary.

Line 110: you say that you enhance the climate sensitivity to make the simulations more robust, and you do this by setting the temperature change for a doubling of atmospheric CO_2 to 3 °C. Is this the parameter TSNS? If so, please mention this, so it is easy to compare with those listed Table S1. Also, can you please explain why doing this makes the simulations more robust? That is not clear to me.

Lines 115-116: There are other studies that suggest different values for both the carbon release magnitude and isotopic composition than Heimdal et al. (2020). It is fine to use the values from Heimdal et al. (2020), (I believe testing different emission scenarios is beyond the scope of the paper, so I understand why you chose to use values from just one study), but I think you should mention why you chose this particular emission magnitude, as opposed to values from other studies. Why do you find the values from Heimdal et al. (2020) more realistic? Again, note that Heimdal et al. (2020) argues for the release of 24,000 Gt *carbon*, not 24,000 Gt CO_2 .

Line 117: specify that it is “model time” 0. Also, I think you should add Davies et al. (2017), which provided additional high precision U-Pb ages of CAMP intrusives.

Lines 118-123: I do not fully understand the k_{silw} parameter. You say that this is multiplied with the “silicate weathering rate” in LOSCAR. What is the “silicate weathering rate”? Is this rate different from the nsi , and if so, how?

Is the k_{silw} written into the LOSCAR code by you, or are you doing this multiplication “manually”?

Why is the k_{silw} not listed in Table S1?

You say that $k_{silw} = 1$ is unlikely to be the real case. Why?

You say that k_{silw} was increased linearly from 1 to “higher values”. Do you mean increase to 1.1, 1.2 and 1.3? Please be specific.

Line 125: I would write “the” different instead of “those”.

Line 138 and 153: Please upload figures with better resolution (Fig. S1 and S2)

Line 166: Spell out the abbreviation of “TS”

Line 177: What details? I do not understand the connection between this and Figure 1.

I'm not an expert on this, but I believe the recommended threshold for TOC concentrations in order to report Hg/TOC values is 0.2 wt.% (see Grasby et al., 2016; Jones et al., 2019). Is this taken into account?

Reviewer #3:

Remarks to the Author:

This manuscript investigates two new terrestrial records of the Triassic–Jurassic boundary for mercury concentrations and isotopes, before combining them with chemical index of alteration (CIA) data to report increased weathering rates during the TJ mass extinction. The exact role of weathering during this major event remains largely unknown (though widely speculated on), so this is certainly an interesting topic. Since the Hg isotope papers of Grasby et al. (2017, *Geology*) and Them et al. (2019, *EPSL*) I have been intrigued about the possibility of using Hg isotopes to chart increased runoff of terrestrial mercury during times of enhanced weathering, so personally, I welcome the approach being attempted here. Unfortunately, however, I think there are a number of issues with the study as it is currently presented, meaning that I cannot recommend it for publication in *Nature Communications*.

I have issues with both the carbon isotope and mercury data as currently presented. For the Qilixias section in particular, the proposed negative excursions in $\delta^{13}\text{C}$ are no greater (and are often lower) in magnitude than some of the 'background' variations between 0–250 m. It doesn't help that there are large stratigraphic gaps in the dataset. For the Hg/TOC, the authors don't present the raw Hg concentrations, only the Hg/TOC ratios, but from looking at the supplementary files it is clear that there is no increase in Hg concentrations in the strata that show high Hg/TOC ratios. In fact, in both sections the Hg/TOC peaks appear to be largely caused by the fact that those sediments have the lowest TOC contents in the section. Indeed, many of the strata that record the Hg/TOC peaks feature TOC contents less than 0.2 wt%, which has been argued by Grasby et al. (2016, *Geological Magazine*) to be the cutoff for reliably normalizing Hg against TOC due to the high percentage uncertainty on those low values. If samples for which TOC < 0.2 wt% are removed from the Hg/TOC data plots, following the protocol of Grasby et al., then the Hg/TOC peaks either disappear (for Haojiagou) or are reduced such that they are similar to background variability (for Qilixia). Thus, I'm not convinced that the postulated mercury elevations are actually genuine, and they're certainly not as robust as the authors claim. These authors have published several papers on mercury before now, so I am surprised that they are presenting the data in this way. I know that some workers in the mercury community are skeptical about the 0.2 wt% cutoff proposed by Grasby et al., but if the authors are going to adopt this stance, then they have to be forthright and say this. And certainly include the Hg concentrations in the main figure.

The main conclusion of this manuscript is that the combined CIA and Hg isotope data document an increase in weathering (and runoff of Hg) during the TJ extinction event, which I agree with, and that the correlation with the Hg peak ties this enrichment to the CAMP volcanism. Even assuming that this enrichment is true (see above), the authors make no reference to the main conclusion of Them et al.'s 2019 paper on the Toarcian. In that paper, a similar Hg/TOC peak correlative with negative MIF values was also interpreted as showing increased runoff of terrestrial Hg, but crucially, whilst the authors did not rule out the possibility that the Hg was originally derived from coeval Karoo-Ferrar volcanism but had been recycled via the terrestrial realm, there was no proof of a volcanic source. Why should these two TJ records be any different? How can the authors be certain that any Hg increase is not simply the result of the enhanced weathering, completely independent of any CAMP emissions. The Them et al. paper is cited elsewhere in this manuscript, so the authors are clearly aware of the study, and I am surprised (again) that it is not discussed more fully.

Finally, I think that in places the authors are overselling the novelty of their paper. To clarify, their methodological approach of combining Hg isotopes and CIA on TJ records is novel, but there are other statements in parts of the text that seem to be trying to imply that this study is more novel than it actually is. For example, mercury is repeatedly stated to be a 'novel' proxy. In the last 5 years there has been a huge body of work published using mercury to study LIP volcanism (including 4 or 5 works on the TJ alone), so I don't think it can be classed as that novel a tool any more. The statement that Hg data do not exist at sites distal from CAMP is incorrect: I would hardly describe any of Nevada, Astartekløft, or the Argentinian sites as being proximal to CAMP, so this study is not presenting the first data from a distal site. The implication that links between volcanism and weathering have not

been robustly determined is also misleading. A number of studies have used a range of techniques to show that the CAMP basalts were rapidly weathered following their eruption (Cohen and Coe, 2002, 2007; Kuroda et al., 2010; Palfy and Zajzon, 2012), so it is clear that more weathering on the continents (at least in that area) was taking place immediately following the eruptions, and directly in response to the emplacement of those basalts. Schaller et al. (2012) even proposes this as a cause of CO₂ drawdown post event. I acknowledge that this does not prove a link between volcanism and weathering away from the CAMP, but that's not how the statement currently reads in my view.

MINOR COMMENTS:

L. 55: just the eruption of massive lavas? It's been argued by a few studies now that the intrusive magmas (and volatiles released from sediments that they were emplaced into) were potentially the more significant trigger (e.g., Svensen et al., 2009; Heimdal et al., 2018). And the dates of some of the low Ti sills that intrude the Amazonas Basin (which is full of evaporites as well as other sedimentary rocks) match the extinction date almost perfectly (Davies et al., 2017), whereas most/all lavas postdate the onset of the extinction.

L. 63: Several workers are not wholly convinced as to how reliably quantitative the CO₂ estimates based on stomatal index data or pedogenic carbonates truly are. I think the general trends (of an increase across the TJ transition) and timing are not in doubt, just the precise values. So it may be safer to leave the 4000ppm out.

L. 67–72: The idea that weathering potentially links volcanism and marine environmental perturbations in this way greatly predates the works of Penman et al. (2020) or Shen et al. (2015). The broad brushstrokes of this mechanism are mentioned by both Algeo et al. (2011, Palaeo-3) and Jenkyns (2010, G-cubed), and various aspects of it had been discussed before then (e.g., Cohen et al., 2004). Earlier citations would be better.

L. 75–78: See earlier point. There's lots of evidence linking CAMP volcanism with weathering, by the fact that the basalts were intensely weathered soon after eruption. If the authors are indicating that we don't have evidence for increased weathering on a global scale, away from the CAMP basalts themselves, then they must state this more clearly.

L. 79: Hg isn't really that novel these days.

L. 93–94: In so far as we can tell from the (very) limited studies on modern volcanoes. Most (if not all) of which were on arc volcanoes I believe, which are hardly analogous with LIPs.

L. 95–97: And the Toarcian OAE (Them et al., 2019).

L. 100: See point above. I would say that all of Nevada, Astartekløft, and the Argentinian sites are distal from the CAMP. Yes, none of them are from the eastern margin of the Tethys, but that's not the only place in the world distal from the CAMP.

L. 110: Looking at Figure 1, Qilixia appears to be north of 30N degrees. I would suggest that this is mid latitude rather than low latitude.

L. 115–116: I'm not at all convinced by the proposed PCIE, which consists of only 2 or 3 data points, and is the same order of magnitude as many background variations, although admittedly the authors do have a question mark next to it. Also the clearest excursion of all at Haojiagou (260–360 m) is ignored. I know that it's in Sinemurian strata, and thus too late to be connected to the TJ extinction, but to avoid confusing the reader, the authors should make clear that they are talking about CIEs close to the TJ boundary in the results section.

L. 129–131: See earlier point. There is a lot of variation in the Qilixia $\delta^{13}\text{C}$ background, and the proposed CIEs aren't really any greater in magnitude. Not convincing for me.

L. 153: See earlier point about how reliable these numbers are believed to be.

L. 160: Wildfires are certainly a source of Hg to the surface environment in the modern day, and it has been suggested that this could also have been the case during major events (Them et al., 2019). Is there any evidence for charcoal in these terrestrial sections?

L. 176: See earlier point about the novelty of Hg.

L. 193: Cite these other studies.

L. 200–204: Blum et al. (2014) state that the plot of ^{199}Hg vs ^{201}Hg MIF is appropriate for samples with MIF of greater than ± 0.3 . Most of the samples in this study do not. So is this interpretation valid? Also, my understanding is that the normal line of evidence for atmospheric transport and deposition of Hg is positive MIF. So how do these two datasets, in which MIF values are overwhelmingly negative, support atmospheric transport? Either the authors' interpretation is wrong, or I'm missing a step in their reasoning. Either way, their logic needs to be made clear.

L. 212: change 'sections' to 'section'. Only one TJ section has been previously studied for Hg isotopes.

L. 229–230: The rise in Os isotope values documented by Cohen and Coe (2007) continues into the Sinemurian. Also, whilst it might represent a long-term increase in continental weathering rates, it could equally reflect a reduction in the levels of primitive basalts being weathered (if, for example, the amount of CAMP available to be weathered was gradually declining over time, which is very likely to have occurred).

L. 239–241: Why does elevated Hg coupled with limited MIF variation relative to background signify a large flux of terrestrial Hg to the sediments? In both Grasby et al. (2017) and Them et al. (2019), terrestrial Hg input is inferred from a peak in Hg combined with a large change in MIF to more negative values.

L. 254: Something is missing in the brackets here.

L. 266–296: Alternatively, it could just reflect a large input of terrestrial Hg to these areas, completely independently of any volcanism (following the model of Them et al., 2019). The authors should discuss in more detail why they have ruled out this option, and cite the 2019 paper appropriately.

L. 280: Change 'toady' to 'today'.

L. 288: 'organic matter' or 'organic carbon' would be better than just 'organic'.

L. 314: Strictly speaking, McElwain et al. (1999), calculated CO_2 concentrations based on stomatal index data, and interpreted the likely temperature change based on those values. They didn't present direct evidence of temperature changes per say. As far as I'm aware, there is actually no palaeotemperature record that directly shows warming at the onset of the TJ extinction, just lots of evidence for increased atmospheric carbon, which presumably would have caused warming.

L. 318: Change to 'Despite the fact that...'

L. 320: Change 'how it did in' to 'during', and 'event' to 'events'.

L. 322: 'basalt' should be 'basalts'.

L. 336–338: How exactly does the weathering cause the oceanic perturbations that triggered biotic stress? I don't think this has been mentioned in the discussion at least. Are the authors invoking the model of weathering = nutrient runoff = eutrophication and anoxia? The problem with this is that evidence for anoxia during the TJ is patchy at best. It definitely doesn't seem to be as widespread as during the PT extinction or the later OAEs, and a lot of places only seem to start recording black shales in the earliest Jurassic, too late to have been the cause of biotic stress.

L. 340–342: How can this model of drawing down CO₂ over 1–3 million years conform with evidence of weathering from the two sections, given that only one of the sections (Haojiagou) has data from sediments deposited that amount of time after the event? Reword.

Figure 1 and caption: Change 'Lgounane' to 'Igounane'. Check throughout the manuscript.

Point by point response to the three reviewers:

We thank the three reviewers for comments which have led to significant improvements to our manuscript. We think we have adequately addressed all the comments from the reviewers and revised the manuscript accordingly. Please note that changes to the text are track-edited (see the WORD file named "main text with marked"), and the line numbers in our response letter refer to those of the revised manuscript.

Dear Dr Shen,

Thank you again for submitting your manuscript "Intensified continental chemical weathering and carbon cycle perturbations linked to volcanism during the Triassic-Jurassic transition" to Nature Communications. We have now received reports from 3 reviewers and, after careful consideration, we have decided to invite a major revision of the manuscript.

As you will see from the reports copied below, the reviewers raise important concerns. We find that these concerns limit the strength of the study, and therefore we ask you to address them with additional work. Without substantial revisions, we will be unlikely to send the paper back to review. In particular, while reviewers agree this work is an important contribution to resolve questions around volcanism and weathering at the Tr-J boundary, it must be fully and transparently contextualized within the existing literature which includes references that present opposing interpretations. Furthermore, Reviewers #2 and #3 note that it does not appear the established threshold of 0.2 wt% for Hg/TOC values was applied here, which must be taken into account in your interpretations. We also agree with reviewers' suggestions for additional supporting evidence, such as Reviewer #1's recommendation for a figure and explanatory text on interbasinal palynoevent correlation.

Reply: Many thanks for the positive consideration of the MS. We responded to the reviews from the three reviewers point-by-point below. The most important changes are: 1) we added the raw Hg concentrations in Figure 2 (B and H), and we categorized the Hg/TOC values using different symbols according to TOC content (e.g., < 0.2 wt%, and ≥ 0.2 wt%, Fig. 2 C,J), in order to address comments from the second and third reviewers; 2) we added statements about the impact of TOC variation on Hg/TOC values in lines 186-195 in the main text and lines 40-78 (Text S2) in the supplementary text; 3) we added a figure (Figure S1) to correlate the two sections, as reviewer 1 suggested; and 4) we added a paragraph (Text S1, lines 25-38) to discuss the correlations between two sections in the two basins in the supplementary text.

If you feel that you are able to comprehensively address the reviewers' concerns, please provide a point-by-point response to these comments along with your revision. Please show all changes in the manuscript text file with track changes or colour highlighting. If you are unable to address specific reviewer requests or find any points invalid, please explain why in the point-by-point response.

Important: In addition to the above, you must comply with the following editorial requests; we will not be able to proceed with your revised manuscript otherwise. Please also see the Nature Communications formatting instructions, which you may find useful while preparing your revised manuscript.

REVIEWER COMMENTS

Reviewer #1 (Remarks to the Author):

Dear authors,

I have read your manuscript with great interest. It is certainly an interesting study that should be published. The question, I think, is not whether CAMP emissions changed the global climate and enhanced weathering and erosion, but how and to what extent in different areas. Here, your manuscript can be an important contribution.

However, I think the manuscript itself needs some additional work. I have a series of comments and concerns that I hope will help you revise your paper. You will find the detailed comments listed after this letter.

Reply: We are grateful for these positive comments. Yes, the spatial and temporal effects to various settings by the CAMP are important to understand the volcanic-induced global ecosystem perturbations near the TJB.

In summary, these are my main issues:

First, give credit where credit is due, and please do not just cite papers of colleagues you side with. In one instance you cite irrelevant papers. I assume, but this may be a mistake in your citation system?

Reply: Many thanks for the reminder. We updated the citations throughout the MS.

Secondly. The plots of the geochemistry for the HJG locality in Fig. 2 are inconsistent with the data provided in the supplement. There are a couple of data points that are in the plots, that do not occur in the Table. In addition, according to the Supplementary Table the CIA data has been corrected, but there is no explanation anywhere in the Methods section or

in the paper why this has been necessary or how they were corrected. Please make sure that the data you supply match the plots in your figures.

Reply: 1) We updated the carbon isotope profile of HJG in Fig. 2A (open circles). The red "+" in Fig. 2A represent organic carbon isotopes from a previous study of HJG (Sha et al., 2015, PNAS). We clarified this in the caption of figure 1.

2) In fact, the CIA correction might not be necessary in either of our studied sections because each sample set fit very well with an ideal weathering trend in its respective A-CN-K diagram, indicating minor post-burial K-metamorphism (Fig. S5, and Text S3). That's why we didn't mention much about the correction in the previous version main manuscript. We decided to not apply correction for our samples to avoid the confuse. We now removed the CIA_{corr} column in Dataset S1 and the CIA_{corr} profiles in Figs. S2 and S3.

Thirdly, the localities are said to be well-constrained by palynology, i.e. spores and pollen, and that this allows interbasinal correlation, but this is neither discussed in the text or shown in a figure. Since the palynozonations for the two sections are not the same, it is impossible for the reader to correlate between the sections. You need to provide some sort of correlation figure and explanatory text for this. Preferably there should be some palynoevents that can be used for correlation. Although, as a palynologist, I realise that this may be very difficult to find as the two localities seem to be in different climate zones. But then again, how can you then say that they allow interbasinal correlation?

Reply: Thanks for raising this issue. The two study sections are well-studied for palynology as well as other biostratigraphic markers that we mentioned in the main text (lines 352-383) and Text S1 in the SI. This is an important reason why we chose these two specific sections for our study.

Yes, as the reviewer said, it is hard to find the same plant fossils in two climate zones' settings. Several studies (e.g., Lu & Deng, 2005, 2009; Huang, 2006; Sha et al., 2011, 2015) have been carried out, regarding to the palyno-stratigraphy of the Junggar Basin. Lu & Deng (2005), Huang (2006), Lu & Deng (2009) recognized several Triassic-Jurassic palynological assemblages throughout the Haojiagou section. Combing the above reported studies, Deng et al. (2010) summarized an integrated palynostratigraphy framework. Palynological assemblages in the same section from Sha et al., (2011, 2015) further supplements the palynological sequences in this area. Zhang et al. (2020) only focused on a 10-meter-thick lignite bed at the Haojiagou section (~ 250 m below the TJB), which belonging to the *Alisporites-Chordasporites-Chasmatosporites* Assemblage of Lu and Deng (2005) and *Concavisporites-Dictyophyllidites-Chasmatosporites-Cycadopites* Assemblage of Huang (2006).

Based on these well-established palynological sequences in the Junggar Basin, Li et al. (2020) made palynological and palaeobotanical correlations between the Sichuan Basin and the Junggar Basin, although much differences exist due to two separate floristic regions, the so called South China Floristic Region represented by the Sichuan Basin and the North China Floristic Region represented by the Junggar Basin.

We correlated the two sections based on: **1)** palynology data. Detailed palynological analyses were carried out for HJG and QLX, although there are different palynozonations from the two areas (see above). The turnover and/or extinction horizon and TJB can be recognized based on the abrupt turnover of the plants in each section (Fig. S1); **2)** similar organic carbon isotope trends in both sections (e.g., three phases of negative excursion, PCIE, ICIE, MCIE), which can be correlated to each other and to other continental and/or marine TJB sections (e.g., Ruhl et al., 2020, ESR); and **3)** astrochronological constraints based on Milankovich cycles, which provide an internal (floating) timescale for HJG (Sha et al., 2015, PNAS) and QLX (Li M et al., 2017, EPSL). We added a figure (Fig. S1) and supplementary text (Text S1) to discuss dating and stratigraphic correlations.

I recommend moderate revision.

Kind regards
Sofie Lindström

General comments:

Replace all hyphens in e.g. Triassic-Jurassic, 2-3 x 10⁶ with en-dash “–”.

Reply: Many thanks for the reminders. We changed hyphens to en-dash throughout the Ms.

Specific comments.

Line 37–38. I suggest that you end the sentence after “boundary”. Begin a new sentence with “We interpret this as reflecting volcanism-induced...”.

Reply: Modified.

Line 49. Replace “the trigger” with “triggers”. Replace “multiple” with “several”.

Reply: Modified.

Line 53. Why Tr-J and not T-J? Since the term “Tertiary” doesn’t exist anymore.

Reply: Modified. We rephrased Tr-J to T-J through the main text and SI.

Line 54–55. Well, actually not only the eruptions of lavas but also intrusive magmatic activity and degassing, see e.g. Heimdal et al. (2020).

Reply: Agreed. We added the “emplacement of sills” in lines 54-55.

Line 63. Replace “Higher” with “Increased”.

Reply: Modified.

Line 65–66. I know you are probably limited by the number of references you can use, but it is not fair to the authors of the original papers to only refer to a review paper, so if possible add a reference or two here.

Reply: We cited the review paper due to the limited number of references allowed by NC (no more than 70). However, we followed the reviewer’s suggestion by adding special references for various environmental perturbations (such as Jost et al., 2017 van de Schootbrugge et al., 2007 and Greene et al., 2012 for anoxia and acidification) instead of citing only one review paper here.

Line 68. Here is another one.

Reply: Fair point. We added specific references (Cohen et al., 2004; Jenkyns, 2010, Algeo et al., 2011) in addition to the review paper here.

Line 70–71. Hm, yes, but I think eutrophication was already mentioned by Bond and Wignall (2014).

Reply: Modified. We added Bond and Wignall (2014) here.

Line 72–75. Also, van de Schootbrugge et al. (2020), and Ahlberg et al. (2003).

Reply: Modified. We added these two references here.

Line 88. Yes, these are nice papers, but maybe you should first refer to Sanei et al. (2012), which was the first paper that suggested the link between Hg and LIPs.

Reply: Modified. We added Sanei’s paper here.

Line 89. Insert “can” after “(e.g., plants, soils)” and “may” before “serve”.

Reply: Modified.

Line 93. I am curious about the term “direct volcanic emissions”. As far as I know magma, and hence also lava, contain fairly low amounts of Hg unless it is assimilated from sedimentary rocks. Maybe you need to explain what you mean by “direct volcanic emissions”.

Reply: We agree. Here we refer to the Hg isotope compositions of the Hg from the deep Earth (near zero MIF, Blum et al., 2014; Yin et al., 2016). Based on the Hg isotope records (near zero) near the TJB for the three marine sections at various water-depths (Thibodeau et al., 2016; Yager et al., 2021), the Hg from the deep-Earth, yielding near-zero $\Delta^{199}\text{Hg}$ values, likely contributed to the Hg peaks at sites near the CAMP (Fig. 1). Of course, the intrusion of sedimentary rocks by sills could also have been a significant Hg source to the atmosphere, but this scenario does not affect the main conclusions of the present work. We modified the sentence to “for direct volcanic emissions from the deep Earth” (lines 85-86) to make the meaning clearer.

Line 98–99. What exactly do the numbers in brackets refer to? Are they the number of localities analyzed from marine and terrestrial environments respectively? If so, they are wrong and you should consult the literature again, including supplementary information.

Reply: Sorry for the error. We have updated the number of the studied sections based on the publications of relative Hg records near the TJB in figure 1 (also adding two recent publications, Yager et al., 2021, ESR; Zhang et al., 2021, Acta Sedimentologica Sinica).

Line 105–106. I am not sure this statement is accurate. After all, several papers have shown that the terrestrial ecosystem, although perturbed, recovers faster than the marine ecosystems, e.g. at the PTB (papers by Schneebeli), the TJB (e.g. Lindström et al. 2012). Plants are very resilient.

Reply: Yes, we agree that the terrestrial ecosystem is easily affected by volcanism due to lack of seawater buffering but is also very resilient (e.g., plants). Terrestrial ecosystems (especially for high latitude regions) are more sensitive to environmental perturbations (warming) than the ocean system nowadays (Bekryaev et al., 2010, Journal of Climate). We modified this sentence, and added Lindstrom et al. (2012) here.

Line 113–116. This does not make sense compared to Fig 2. First you say that you have a pronounced negative shift near the TJB at 20–60m and then that one of the CIEs (that one) is at 30–70m.

Reply: Sorry for the error. The pronounced negative shift near the TJB is at 30-70 m in thickness.

Line 125–127. Actually, in the H₂G section kaolinite increases steadily through the Rhaetian. You should comment on this.

Reply: Modified. We rephrased the sentence to “The kaolinite content shows a steady and slight increase near the T–J boundary and above it”.

Line 126–128. You should comment on the pronounced increase in smectite in the Sinemurian. What is this related to?

Reply: We still do not know the exact cause for the abrupt increase of smectite content at 200-450 m, which represents the Sinemurian stage. It is possibly related to changes of climate or terrestrial inputs. It does not affect the main conclusions of the present work, but this could be investigated in future works in multiple settings to test whether this was a local or global phenomena. We made this clear in lines 130-132.

Line 149. There are other papers dealing with the timing and onset etc of this LIP, and not all of them are in agreement with Korte and Ruhl. Unless you specifically want to add a sentence at the beginning of the paper referring to the variety of papers dealing with this subject and a statement that you are herein using the papers by Korte et al. and Ruhl et al. as your main stratigraphic framework, you should refer to some of them here.

Reply: It is still debated about the relationships between the CIE and the onset and timing of CAMP. We did not mean that we used Korte and Ruhl's model in our study, and we just cited them as a review paper here. We have added other references (e.g., Zaffani et al., 2018; Heimdal et al., 2020) about the relationships between the timing and onset of the LIP.

Line 153. Why is the Korte et al. (2009) paper cited here? You mean Korte et al (2019) again?

Reply: Sorry for the error. We removed this citation.

Line 156–157. I think there have been other papers dealing with this subject before Korte et al. (2019).

Reply: Modified. Yes, there are many papers reporting the negative CIEs near the TJB. We cited only the review paper here because of the limited references policy. Now we have added more references (e.g., Ruhl and Kürschner, 2011; Ruhl et al., 2011, 2020; Zaffani et al., 2018; Korte et al., 2019; Heimdal et al., 2020).

Line 159. The first paper to suggest a 3-4 degree Celsius rise was McElwain et al. (1999). Please give credit where credit is due! If you need to lose a reference to squeeze in McElwain's paper, you could do without one of Schaller's papers here.

Reply: We added McElwain's paper here.

Line 161. Here you could also cite Marynowski and Simoneit (2009) and Petersen and Lindström (2012).

Reply: Many thanks. We cited these two papers here.

Line 162. And how about van de Schootbrugge et al. (2007)?

Reply: Many thanks. We cited Schootbrugge's paper here.

Line 182–183. Why is ONLY Percival et al. (2017) mentioned here? You don't think all the above mentioned papers on lines 178–179 add to the knowledge of volcanic induced Hg-loading?

Reply: We updated the citations here.

Line 193. Other studies which you don't want to mention because?? There are papers by Sanei, Grasby, Bergquist, Fantasia, Jones, and myself with co-authors, and many others to choose from. For the TJB there are only two papers that deal with terrestrial Hg loading, and that is Percival et al. (2017) and Lindström et al. (2019). If you want to discuss more about the terrestrial environment and Hg, there is always also a paper on the Devonian–Carboniferous extinction by Marshall et al. 2020.

Reply: We did not mention the special paper here because of the limited number of references. We have added a review paper (Grasby et al., 2019, ESR) now.

Line 197–199. Several of your high Hg/TOC peaks are the product of very low TOC values. You need to discuss this. It does not necessarily mean that they are not real, but you need to show the readers that you are aware of this.

Reply: Thanks for this suggestion. The best procedure for normalizations of Hg in sediments is still not certain. Organic matter is widely assumed to be the dominant mineral host of Hg in sediments, and Hg/TOC is widely used to normalize Hg (e.g., Grasby et al., 2019, ESR). However, other minerals, such as clays (Sial et al., 2013, Paleo-3; Shen et al., 2019, Geology) and sulfides (Shen et al., 2019, 2020, EPSL) can be the dominant host for Hg under "special" conditions, complicating the use of TOC to normalize Hg.

In the present work, as the plots in Figure S4 show, the stronger correlation ($r = +0.76$, $n = 105$, $p < 0.01$) between Hg and TOC at QLX is evidence that organic matter is the dominant host for Hg in this section. However, the host of Hg at HJG is still unclear because of weak correlations between Hg-TOC, Hg-Al, and Hg-TS (as stated in lines 67-78 in supplementary materials).

Hg/TOC is suitable to normalize Hg at QLX. The TOC values for the samples yielding higher Hg/TOC (e.g., >100 ppb/wt.%) values are various, ranging from < 0.2% to $\geq 0.2\%$. TOC values (0.19, 0.15, 0.16, 0.18, 0.18) are close to the threshold value (e.g., 0.2 wt.%) for five higher Hg/TOC samples (e.g., Hg/TOC > 350 ppb/wt.%) during the mercury enrichment interval (Fig. 2G), but high Hg/TOC values (e.g., >100 ppb/wt.%) are also exhibited in many other samples (10 samples) that have TOC higher than 0.2 wt.%. So

even if low TOC affects the magnitude of Hg/TOC for these five samples, the pattern of secular variation of Hg/TOC at QLX is not significantly changed.

For HJG, it is hard to normalize Hg correctly owing to inability to determine the dominant host of Hg in these sediments. The variabilities in Hg/TOC peaks near the ICIE are partly a consequence of low TOC values (Fig. 2C). Owing to the low stratigraphic resolution of our long section (116 samples in ~ 650 m strata), we may not have sampled beds with high TOC near the TJB. For comparison, a higher-resolution study (61 samples in ~ 90 m strata near the TJB) of Hg in the same outcrop was carried out recently (Zhang et al., 2021, *Acta Sedimentologica Sinica*). Their data show higher Hg/TOC values near the TJB in a pattern similar to the interval with higher Hg/TOC values in the present work (Fig. 2C). Given the similar Hg/TOC record from an independent study, it appears that the Hg enrichment interval at HJG has been reliably determined in our study as well, although the magnitude of Hg/TOC could be vary.

We added the raw Hg and Hg/TOC data from Zhang et al. (2021), the raw Hg values of the study sections (B and H) as well as modified the Hg/TOC symbols (C and J) to reflect TOC content (e.g., <0.2 % and $\geq 0.2\%$) in figure 2. We also added discussion about the influence of TOC content on Hg/TOC in lines 186-195 in the main text and lines Text S2 (lines 40-78) in the supplementary text.

Line 204–206. This is an oversimplification. In fact, as shown by Blum et al. (2014) there is overlap in $\Delta^{199}\text{Hg}$ in coals, plant litter, total gaseous Hg, rainfall-coal impacted, lichens, etc etc. Besides, why would there be “more” continental input from plants and soil? They are both terrestrial successions, right? So isn't it really a question of preservation of the organic matter to which the Hg is bound?

Reply: Yes, there is overlap of $\Delta^{199}\text{Hg}$ among coal, plant litter, and lichens to some degree, and it is hard to distinguish them. The Hg in terrestrial sediment mainly consist of two sources, including terrestrials and atmosphere. There are remarked differences between the $\Delta^{199}\text{Hg}$ from terrestrial systems (negative values) and from the atmosphere (positive values) (see summary figure (figure 2) in Shen et al., 2019, ESR). The $\Delta^{199}\text{Hg}$ values in sediments affect by the proportions of the two sources, yielding more negative values in sediments with high proportions of terrestrial inputs, and more positive values with high proportion of atmosphere loading. We modified the statement to clarify that HJG contains a larger proportion of terrestrial-sourced Hg than QLX (lines 205-208).

The degree of preservation of OM can affect Hg concentrations but cannot affect $\Delta^{199}\text{Hg}$ values, which have distinct variations between terrestrial and atmospheric sources.

Line 210–213. Thibodeau et al. (2016) actually show rather consistent $\Delta^{199}\text{Hg}$ values with only slight variations from 0.05 ‰ to -0.05 ‰. At least from your supplementary table, none of your values in HJG comes close to that, and only a couple of values in QLX are close to but not within that range. I think you need to discuss what the cause of these

differences are.

Reply: Yes, the $\Delta^{199}\text{Hg}$ values are lower at HJG and QLX than in relatively deep-water offshore settings (e.g., Nevada and Levanto, Thibodeau et al., 2016; Yager et al., 2021), which likely received Hg dominantly from seawater (thus, positive $\Delta^{199}\text{Hg}$). However, the $\Delta^{199}\text{Hg}$ values are in a similar range to those of nearshore settings (St Audries Bay, Yager et al., 2021), which received more terrestrial-sourced Hg. The Hg in marine sediments comes mainly from three sources: seawater, terrestrial inputs, and atmospheric inputs. For deep-water offshore settings close to the CAMP (Nevada and Levanto), the near-zero $\Delta^{199}\text{Hg}$ in sediment could result from: 1) atmospheric transfer of volcanic Hg ($\Delta^{199}\text{Hg}$ close to zero); and 2) a mixing of terrestrial (negative values) and marine (positive) sources in equal amounts, yielding $\Delta^{199}\text{Hg}$ near zero in the sediments.

The present study sections are terrestrial (Hg mainly source from terrestrial and atmosphere) and located far from CAMP. The Hg proportion from terrestrial inputs is higher than that from atmospheric inputs, resulting in negative $\Delta^{199}\text{Hg}$ values (see the detailed replies to the last comments). To reply to the reviewer's question, the proportions of terrestrial versus atmospheric Hg are different for the HJG and QLX sections. We made this clear in lines 205-216.

Line 226–230. The Os-isotope records are complicated and vary significantly between the epicontinental sea and Panthalassa.

Reply: Yes, the Os isotope records are complex in spatial-temporal scale. However, it's beyond the scope of the present work. We just cited that intense chemical weathering near the TJB was evidenced by Os isotopes here.

Line 230–233. There are also other papers referring to Sr-isotopes, e.g. Callegaro et al. (2012) and Kovacz et al. (2020). The Kovacz et al. (2020) paper is very interesting as it suggests alternative interpretations.

Reply: We added the suggested references here.

Line 235–236. And here you could mention van de Schootbrugge et al. (2020) on soil erosion and weathering.

Reply: Many thanks. We added the statement about soil erosion and added Schootbrugge's paper here (lines 229-230).

Line 238–239. Why does the Hg records record intense continental weathering? Terrestrial Hg derived from what source? You need to explain this better.

Reply: Good question. The terrestrial ecosystem (e.g., forest, soil) is a significant Hg reservoir in the modern world (Selin, 2009), characterized by negative $\Delta^{199}\text{Hg}$ values

(Blum et al., 2014). Intense chemical weathering would increase the runoff of continental materials as well as Hg to oceans. So elevated Hg in nearshore sections could result from increased terrestrial runoff, but not from volcanic inputs (e.g., Grasby et al., 2017, *Geology*; Them et al., 2019, *EPSL*; Shen et al., 2019, *EPSL*). Besides, Hg isotopes, especially $\Delta^{199}\text{Hg}$ values, differ between terrestrial (negative values) and volcanism-induced atmospheric sources (near zero or positive values). Integrated use of Hg concentrations and isotopes could help to track Hg sources in sediments. In the present study, the $\Delta^{199}\text{Hg}$ values during the high-Hg interval can be divided into two groups (Fig. 3A, 3B), one group having $\Delta^{199}\text{Hg}$ values similar to the background samples, which show the similar sources of the elevated Hg (e.g., terrestrial reservoirs) (blue arrow dashed lines in Fig. 3A, 3B). The other group have more positive values (close or a little higher than zero) evidence atmosphere sources, which could be related to volcanic eruption (purple arrow dashed lines in Fig. 3A, 3B). We made this clear in lines 231-246.

The terrestrial Hg could be derived from: 1) the sediments on land, such as soil; 2) the re-cycled Hg^0 from the atmosphere (could be related to volcanisms). Both of these two sources are having negative $\Delta^{199}\text{Hg}$.

Line 254–255. There is an empty () after HJG.

Reply: Sorry for the error. We added the r values for the co-efficient values for the correlations between CIA and $\Delta^{199}\text{Hg}$ for HJG in figure 3D.

Line 280. “today”

Reply: Modified.

Line 296. Here you could also consult Landwehrs et al. (2020).

Reply: We added Landwehrs`s paper here.

Line 340. I guess here you mean in this particular area, i.e. China? But this is really a repetition of what is on line 332. Why not merge these two sentences?

Reply: Not exactly. The drawn CO_2 from the LOSCAR model was based on the elevated chemical weathering in global scale, not just in a local region. We show the enhanced continental chemical weathering lasting 1-2 million year in line 332 (original version), while we stated the elevated chemical weathering from the LOSCAR model can draw the CO_2 concentrations over the similar timeframe. We rephrased the sentence to avoid the repetition.

Line 346–347. It says, both in the abstract and here, that the sections are well-dated, and that this allows interbasinal correlation, but no correlation between them is shown in this paper. How robust is the spore-pollen stratigraphy in western China? Here, there are two

quite different palynozonations from the two areas. Besides that, there are also additional papers dealing with the same interval from the two basins, presenting alternative zonations, e.g. Zhang et al. (2020) and Huang (2006). It would be good if you could comment on this.

Reply: As we replied above (pages 3-4 in this file), the two studied sections are well-studied for palynology and other works as we mentioned in the main text (lines 352-383) and Text S1 in the SI. This is the reason why we chose these two sections for our study. We correlated the two sections based on: 1) the palynology data. Detailed palynology works were carried out for HJG (e.g., Sha et al., 2015, PNAS) and QLX (Li et al., 2020, PPP), respectively. The turnover/extinction events and TJB can be recognized in each section; 2) organic carbon isotopes also exhibit similar trends (e.g., three negative excursions, PCIE, ICIE, MCIE) in both sections as well as with other continental and/or marine sections; 3) The Milankovitch cycles also provide an internal timeframe for both HJG (Sha et al., 2015, PNAS) and QLX (Li et al., 2017, EPSL). We added a figure (Fig. S1) and a section (Text S1) to discuss this issue in the SOM.

Line 362. Sha et al. (2011) is not in the reference list.

Reply: Many thanks. We added the paper in the reference list.

Line 403. What is the detection limit for the instrument for Carbon and Sulfur, respectively? Some of the TOC values are very low, <0.20 wt %. In most cases it is these low TOC values that result in the large peaks in Hg/TOC, so I think it is important that you clarify this to the readers.

Reply: The detection limit for the C and S is 0.05 wt.% for Eltra 2000 C-S analyzer. See the detailed replies above about the effects by lower TOC to the higher Hg/TOC (see replies to the similar comment in the pages 8-9 in this file).

Line 619–620. We actually analyzed Hg from a primarily terrestrial succession also, the Norra Albert quarry and the Albert-1 core (a core drilled inside the quarry) in southern Sweden (Lindström et al. 2019).

In addition, Percival et al. (2017) also analyzed samples from the terrestrial succession at Astartekløft in East Greenland.

Reply: Fair point. We updated the marine and terrestrial sites in figure 1.

Comments on FIG. 2:

What are the red crosses on the Haojiagou C-isotope curve?

The C-isotope curve for Haojiagou does not look the same on the figure as when the actual values from the Supplementary table are plotted. There are values on the Figure that don't exist in the table, e.g. the neg C-isotope peak at c. -95m.

Reply: As we replied above, the red "+" represents organic carbon isotopes from previous publications (Sha et al., 2015, PNAS). We made clear about this in the caption of Fig. 2. We updated the carbon isotope profile of the present work for HJG in Fig. 2 (column A, open circles).

Comments on FIG. 3. There is either something wrong with your plots or with your Supplementary table, e.g. the data in the table shows values down to -0.67. Please make sure that the plots shows the correct data. Specify whether you have plotted against your CIA or Corrected CIA values.

Reply: Not exactly. The referee refers to the -0.67 value in the table that is the $\delta^{199}\text{Hg}$ value for HJG (column U and lines 44), and QLX (Column U, lines 64). We plotted $\Delta^{199}\text{Hg}$ to Hg/TOC (A, B), $\Delta^{201}\text{Hg}$ (C), and CIA (D) in figure 3. We plotted $\Delta^{199}\text{Hg}$ and again CIA in panel D of Figure 3.

Reviewer #2 (Remarks to the Author):

See attached PDF (We copied the reviews from the PDF to this word file)

Review: Intensified continental chemical weathering and carbon cycle perturbations linked to volcanism during the Triassic-Jurassic transition

Authors: Shen, J. et al.

This paper integrates a number of different datasets and methods in order to investigate the cause-and-effect relationships between CAMP, chemical weathering and the carbon cycle around the T-J boundary. They examine two new terrestrial sections (both high and low latitude) from China where various proxies can be analyzed, which allows to directly compare carbon cycle perturbations (carbon isotopes), volcanism (Hg data) and weathering (Hg, CIA and clay mineralogy). One issue with tying together the relationship between CAMP, the end-Triassic mass extinction and the associated carbon cycle perturbations, is that the various proxies are from different sections, and correlating these have proven to be difficult. For example, CAMP lavas are emplaced in continental deposits, but most C-isotopic records are from marine sections. Furthermore, there exists different interpretations of the isotope data considering the timing of carbon cycle perturbations and CAMP activity (e.g., Korte et al., 2019 vs. Lindström et al., 2017). Producing data of proxies for weathering, volcanism and carbon cycle perturbations from the same section prevents the issue of correlating, and is therefore of great importance for the end-Triassic, where these relationships are yet not fully understood. This point alone provides novelty to this contribution. In addition, they use carbon cycle modelling to test

whether or not increased weathering due to the emplacement of CAMP can explain the observed drawdown of atmospheric CO₂ within the same timeframe as the observed intense weathering interval. I believe the main goal combined with the used methods and approaches, provides novelty to this contribution. Therefore, I believe this paper is appropriate for Nature Communications. However, I do believe the authors need to address some major concerns before it is ready for publication.

Reply: We are extremely grateful for the positive comments.

While the datasets and selected approach in this study are convincing, my major concern is the way the manuscript is organized and written. First of all, you have to convey more clearly in the introduction 1) what do we already know, 2) what knowledge are we missing, 3) what are you going to do to close the current knowledge gaps, and 4) why is this important. You have to be more specific in the introduction considering the work you have actually done. For example, the results-section starts with the description of carbon isotope data, but it is never mentioned in the introduction that you are presenting C isotope curves. Therefore, the importance of presenting this data is not clear at all.

Reply: Modified. We revised the Introduction extensively (lines 100-111) to more clearly follow the organization proposed by the reviewer.

Also, the LOSCAR modelling suddenly appears at the end of the discussion, but this has barely been mentioned before. It is not clear why you have included this modeling. Why is it important? The way it is written now, it seems like it is just something you threw in there at the end.

Reply: Modified. We revised the Introduction and added the rationale for application of the LOSCAR model, which is to quantify the relationship of C cycle, atmosphere CO₂ and silicate weathering rates (lines 100-111).

Furthermore, the different data and conclusions must be much better tied together. For example, in the way the discussion is written, the sections about the different proxies or methods stand completely alone. The whole point of this study is to integrate different proxies and methods in order to say something about the relationships between CAMP, chemical weathering and the carbon cycle around the T-J boundary. This would not be possible if you only presented carbon isotopes, or if you only did modeling. It is therefore a matter of reorganization, and to better convey your message.

Reply: First, we have modified the Discussion by replacing headers like "Carbon isotopes" and "Mercury records" with ones that indicate the scientific issue under discussion. Second, we have integrated our discussion of the various proxies used more effectively than in the original manuscript. Third, we have modified the Introduction to indicate the role of carbon isotopes in our study.

Lastly, I think the manuscript would greatly benefit from a read through by a native English speaker.

Reply: Modified. The co-author Thomas J Algeo has thoroughly edited the revised MS (he also edited the original version thoroughly, notwithstanding any minor language errors that might have slipped through).

Specific comments:

Main Text

Line 33: "...organic carbon isotopes.."

Reply: Modified.

Lines 33-37: please rewrite, this sentence is too long.

Reply: Modified.

Line 38: "... near the T-J boundary, reflecting a ..."

Reply: Modified. We divided this sentence into two based on the comments of the first reviewer.

Lines 39-40: delete "in both sections."

Reply: Modified. We deleted "in both sections"

Lines 40-41: "... 2 million years, which is consistent with model results..." Also, it is not clear that these are carbon cycle model results that you have produced. This is important!

Reply: Modified.

Line 41: I would use the word "lastly" instead of further.

Reply: Modified. We rephrased the "Further" to "Lastly" here.

Lines 51-52: "...volcanism, environmental perturbations (e.g., carbon-isotope excursions, chemical weathering, anoxia, and acidification), and biotic turnovers (e.g., mass extinctions) remain poorly...»

Reply: Modified.

Lines 53-57: please rewrite, this sentence is too long. Also, the eruptions of lavas did not trigger the end-Triassic crisis, but rather the volatile emissions.

Reply: Modified. we separated the sentence into two, and we added the statements about the “emplacement of sills” (lines 52-57).

Lines 57-59: The way it reads now it seems like the synchronicity itself is considered the main trigger between CAMP and the extinction, which is not the case.

Reply: We have improved the wording of this sentence.

Lines 61-62: “... leading to negative carbon isotope excursions (CIEs) in both inorganic and organic matter at a global scale...”

Reply: Modified.

Line 63: I would write “increased” instead of higher.

Reply: Modified.

Line 64: I would write “warming” instead of greenhouse climate.

Reply: Modified. We changed “greenhouse climate” to “warming”.

Lines 67-68. I would write “represents” or “act as” instead of “is”. Also, you need to explain why this is a link.

Reply: We replaced “is” by “act as”. We stated that continental chemical weathering acts as a link in lines 65-70.

Lines 68-72: chemical weathering can lower CO₂ through weathering? Explain why CO₂ concentrations decrease due to weathering. I think what you are trying to say in the following sentences is that the elevated CO₂ concentrations (that eventually lead to increased weathering through elevated temperatures and runoff) is sourced from volcanic emissions?

Reply: Chemical weathering, especially of basalts, can lower CO₂ by the reaction: $\text{CaSiO}_3(\text{s}) + 2\text{CO}_2(\text{g}) + \text{H}_2\text{O} \rightarrow \text{Ca}^{2+}_{(\text{aq})} + 2\text{HCO}_3^{-}_{(\text{aq})} + \text{SiO}_2(\text{aq})$. This process is thought to play a significant role in regulation of the pCO₂ in the atmosphere at timescales of tens of thousands years during Earth history (Kump et al., 2000, Annu. Rev. Earth Planet. Sci.).

Yes, we were trying to show that volcanism-induced elevated temperatures and CO₂ concentrations could result in intensified chemical weathering. We made this clear in lines 65-70.

Lines 72-77: this is why your study is so important! There is already strong evidence for weathering, volcanism, and carbon cycle changes, but the problem is that the different datasets must be correlated between sections. And this has been proven quite difficult. However, the sections you study represents an opportunity to provide all these different proxies without worrying about correlations. This has to be communicated much more clearly.

Reply: Yes, we agree with that demonstration of accurate temporal relationships among these processes is a key reason why our work is so important. We made clear in lines 70-73.

Lines 79-81: I would write the sentence like this: Mercury (Hg) concentrations and isotopes are widely used to track volcanic (Thibodeau et al., 2016; Grasby et al., 2017; Shen et al., 2019a) and terrestrial inputs (Shen et al., 2019b; Them et al., 2019) in ancient depositional systems.

Reply: Modified.

Lines 81-85: please rewrite, this sentence is too long.

Reply: Modified. We divided the sentence into two.

Line 86: "... eruptions, such as LIPs,..."

Reply: Modified.

Line 87: "... , leading to Hg enrichments in diverse facies globally..."

Reply: Modified.

Lines 90-97: I would try to shorten this, and to make a better link between the previous sentences. The main message here is that Hg isotopes are necessary to identify the source of the Hg.

Reply: Yes, we would like to show that Hg isotopes are important to track Hg sources. We do not think it can be shortened more. This part includes three sentences: the first one is a "topic" sentence to show the Hg isotope is useful; the second one to show that $\Delta^{199}\text{Hg}$ varies among different reservoirs (why it is useful), and the third one to offer a few examples during Earth history to show that it is promising to track Hg sources.

Line 98-100: I would write these sentences like this: "... boundary studies have analyzed Hg concentrations in both marine and terrestrial sections (Fig. 1), these records mainly

originate from the central Pangean region. However, data from areas more distal from CAMP (e.g., the eastern margin of the Tethys Ocean) are lacking.”

Reply: Many thanks. We modified this sentence.

Line 101: rewrite “can be due to”. The introduction should end with a separate paragraph where you explain specifically what you are going to do, why it is important, and why your data and approach is novel. This paragraph should be based on what you have written previously in your entire introduction. This includes some of the arguments you make in lines 98-107, but also in lines 72-77.

Reply: Modified. We have revised the last paragraph of the Introduction to more fully address the goals and importance of our study. See lines 100-111.

Line 113: like previously mentioned, it is never written in the introduction that you are presenting C isotope curves, so the first time I read the results-section I was wondering why are you talking about carbon isotopes?

Reply: We have modified the Introduction to indicate the role of carbon isotopes in our study.

Line 115: “A total of three negative carbon...”. Also, I think you should mention the magnitude of each CIE, so that they can be easily compared to the precursor/Marshi, initial/Spelae and main/Tilmanni CIEs observed in other sections.

Reply: Modified. We added the magnitude of each CIE in lines 116-118.

Line 118: this is a detail, but Fig. 2A should appear before Fig. 2B.

Reply: Sorry for the error. We added Fig. 2A in line 116 before the first appearance of Fig. 2B in line 120.

Line 117-119: is the TOC measured in % or wt.% ? In the excel table, the TOC content is also described as % and not wt.%.

Reply: The units of TOC are wt.%, we modified the units of TOC in the main text and tables.

Lines 116-119: please rewrite this sentence.

Reply: Modified. We modified the depth of the intervals yielding various Hg/TOC various.

Line 122: I would write: “there is an excursion in the CIA from 75-85...”. Also, say if the excursion is negative or positive.

Reply: Modified.

Lines 125-126: "The kaolinite content..."

Reply: Modified.

Lines 129-130: again, I think you should say the actual magnitude of each CIE.

Reply: Fair point. We added the actual magnitude of each CIE (lines 134-136).

Line 138: "The kaolinite content increases..."

Reply: Modified.

Line 153: there must be a better way to say this

Reply: We have modified this sentence.

Lines 147-173: What is the important message you want to convey with this section? I believe it is that the T-J boundary CIEs found in several sections have been interpreted to have occurred as a result of carbon emissions from CAMP. The CIEs you see in the two sections from your study can be correlated to the other CIEs. So, this tells us that carbon cycle perturbations are connected to volcanism (i.e., CAMP). However, this message gets lost. I think you should move the part about the CIEs from HJG and QLX being comparable to the precursor/initial/main CIEs found elsewhere (lines 163-168) up to the beginning of this section (to line 147). This is what is important! How your data connects to previous data. And then you can say that the CIEs have been attributed to the release of carbon gases from CAMP, and make the connection between the CIEs and volcanism. Lines 157-162 feel out of place to me, I do not understand why this information is important and how it connects with the rest of the section.

Reply: We have modified the Discussion by replacing headers like "Carbon isotopes" and "Mercury records" with ones that indicate the scientific issue under discussion. Also, we have integrated our discussion of the various proxies used more effectively than in the original manuscript.

Line 176: I would write it like this: "Mercury has previously been used to track volcanic inputs to various marine and continental Tr-J boundary sections that were"

Reply: Modified.

Lines 176-182: Again, like the previous section (carbon isotope excursions), you start the

section with what other people have done. Start with your data instead, this is what is important! Explain/interpret what your data means by using the data and conclusions from the other studies.

Reply: The Discussion section has been completely re-organized. We have followed the reviewer's guidance closely.

Line 184: refer to Fig. 2B, G.

Reply: Modified.

Lines 185-190: "Owing to its..... valid procedure for the evaluation of volcanic Hg inputs.» This feels misplaced. Maybe move this to the methods-section or SI?

Reply: It is a brief introduction to the normalized Hg values, and we would like to keep it in the main text. We fully discussed the host minerals of Hg and the methods of Hg normalization in the SI (Text S2).

Line 191: "... of such excursions.." which excursions do you mean?

Reply: Modified. We refer to the initial CIE here.

Line 193-196: I do not understand what you mean here: The 105 year gap between the onset of the Hg enrichment (ME) and the initial CIE is the result of early extrusive emplacement of dikes and sills before the main eruptive stage. Do you mean that the onset of Hg enrichment represents the earliest CAMP stage (the earliest sills and dykes), while the next Hg enrichments, overlapping with the initial CIE, represents the main eruptive CAMP stage? Why do you not mention that the onset of Hg enrichment actually overlaps with the precursor CIE? I would say that is very important information, which further argues for a link between carbon cycle changes and volcanism. Also, dikes and sills are intrusive not extrusive (line 195).

Reply: Good point. We modified the text here, and we replaced "extrusive" with "intrusive".

Line 198: "... Tr-J boundary of the study sections likely represent multiple..."

Reply: Modified.

Lines 200-201: Do you mean in general or your data specifically? Why is the Hg presumably volcanically sourced? I assume this is what you explain further down in the section, but you cannot conclude with this before you have explained why. Give arguments as to why you think the Hg is volcanically sourced, and then you can make a conclusive statement like this. Also, you refer to Figs. 2 and 3, but these figures have a lot of information. Be specific, which parts of these figures show what you are trying to say?

Reply: The first sentence is the topic sentence of the paragraph, but not the conclusion. We prefer to use a “topic sentence” to identify the key idea in each discussion paragraph. This is an effective writing style—it may differ from the reviewer’s writing style, but s/he should respect our choice. We added special panels for the citations of figures 2 and 3.

Line 204: I would write “compared to” instead of “than at”.

Reply: Modified.

Line 205: what do you mean by more continental Hg inputs? Do you mean that there is a higher abundance of terrestrial sourced Hg in HJG compared to QLX, or compared to a volcanic source, or both?

Reply: Sorry for being unclear. What we want to express is that the proportion of Hg from terrestrial sources was greater at HJG compared to QLX. We have revised lines 205-208 to make this clear. See also the detailed reply above (pages 10-11 in this file).

Line 207: I would write “compared to that of the” instead of “over” background levels.

Reply: Modified. We replaced “over” with “compared to that of the”.

Lines 208-209: You already presented these values (-0.4 ‰ to -0.2 ‰ for HJG, and -0.2 ‰ to -0.05 ‰ for QLX).

Reply: Yes, we indeed presented the $\Delta^{199}\text{Hg}$ values for the difference of the pre-extinction interval between the two sections in line 206. We also presented these data here for comparison with the background and elevated values of the extinction interval near the TJB (lines 210-211).

Line 211: “... volcanically sourced Hg (with near-zero $\Delta^{199}\text{Hg}$ values) to the sediments...”

Reply: Modified.

212-213: This has already been written (lines 179-181).

Reply: Modified. We revised the sentence to “This pattern is consistent with an increased flux of volcanically sourced Hg (with near-zero $\Delta^{199}\text{Hg}$) through the atmosphere (purple dashed arrow in Fig. 3A, B), also corresponding to $\Delta^{199}\text{Hg}$ signals in shallow-marine T–J sections (Thibodeau et al., 2016; Yager et al., 2021). However, the present study sections exhibit lower $\Delta^{199}\text{Hg}$ values than deep-water marine sites (Thibodeau et al., 2016; Yager et al., 2021), (Fig. 3 A, B).”

Lines 214-222: This section seems out of place. Combine these results with the other Hg data supporting the same conclusions. Instead of separating the measured $\Delta^{199}\text{Hg}$ values and the mixing model, explain why the source is terrestrial or volcanic, based on these results combined. And then you can say that this is also supported by others (e.g., Thibodeau et al., 2016).

Reply: Modified. We reorganized the Discussion section. See the above detailed reply regarding the sources of terrestrial and volcanic Hg (pages 10-11 in this file)

Lines 225-237: Again, start with your data, and then use other studies to back up your conclusions. Also, I feel like this could be substantially shortened. For example: Increased continental weathering associated with the Tr-J boundary is documented by increasing ratios of both $^{187}\text{Os}/^{188}\text{Os}$ and $^{87}\text{Sr}/^{86}\text{Sr}$ during the Hettangian (Cohen and Coe, 2002, 2007; Kuroda et al., 2010), as well as the replacement of shallow-marine carbonates by siliceous sponge-dominated “glass ramp” cherts in Nevada (USA) and Peru (Ritterbush et al., 2014).

Reply: The first sentence is the topic sentence of the paragraph, but not the conclusion. We prefer to use a “topic sentence” to identify the key idea in each discussion paragraph. This is an effective writing style—it may differ from the reviewer’s writing style, but s/he should respect our choice.

Lines 238-239: Why do the Hg records document intense continental weathering? I assume this is what you explain further down in the section, but you cannot conclude with this before you have explained why. Give arguments as to why you think the Hg records document intense continental weathering, and then you can make a conclusive statement like this.

Reply: About why do the Hg records document intense continental weathering, refer the detail replies in pages 10-11 in this file.

The writing style see preceding reply.

Line 246: Also refer to Fig. S4.

Reply: Modified.

Line 254: There is an empty parenthesis

Reply: Sorry for the error, we added the r value for the correlation between CIA and $\Delta^{199}\text{Hg}$ at HJG.

Lines 257-262: please rewrite, this sentence is too long.

Reply: Modified. we divided the sentence into two sentences.

Line 258: remove “e.g.,”

Reply: Modified. We removed “e.g.”.

Lines 266-282: This should be at the end of the discussion. This section reads like a conclusion. Merge this paragraph with your conclusion.

Reply: This paragraph is needed to sum up the results and interpretations of our geochemical data before we proceed to the modeling part of the study.

Line 269-270: “volcanism acted as a trigger by releasing large...” Also, volcanism acted as a trigger of what? Be specific.

Reply: Modified. We refer to volcanism as a cause for Earth-surface environmental perturbations. We removed “acted as a trigger” here for we stated it in the next sentence.

Lines 273-282: here you talk about the fact that your data suggests that high-latitude continental settings are more sensitive to shifts in weathering intensity compared to low-latitude settings. This is interesting and important information (you even mention this in the abstract). However, I feel like this information suddenly appears (a bit out of the blue) as part of this conclusion altype paragraph. I think you should try to better explain why this is the case before you conclude.

Reply: Yes, the sensitive variation to the environmental perturbations for different latitude is one of the highlights for the present work. Besides, in the conclusions, we stated this issue in one paragraph in the discussion section (lines 269-284) behind the discussion about the intensified chemical weathering in the two sections. This is a further understanding about the spatial-and-temporal variations of chemical weathering by volcanism in different latitudes.

Line 274: what do you mean by “in step”?

Reply: We meant the intensification of chemical weathering coincided with the Hg spike. We rephrased “in step” as “is consistent with” in lines 276-277.

Lines 276-277: I would write this instead: Surprisingly, with the current age model, the offset in the low latitude site could correspond to a time interval of ~ 200 kyrs (Fig. 2), suggesting a very protracted shift in tropical conditions relative to high latitude settings.

Reply: Modified. We rephrased this sentence as suggested.

Line 280: «today» is misspelled.

Reply: Sorry for the error, modified.

Lines 285-287: a consequence of chemical weathering is the consumption of CO₂ which is drawn down by silicate weathering? Please rewrite this sentence.

Reply: Revised.

Line 288: «... weathering was the increase in organic...» Is it really necessary to mention organic carbon burial, since you do not focus on this in other parts of the manuscript? Maybe you can delete this sentence.

Reply: Modified. We deleted this sentence as suggested.

Lines 292-296: This feels misplaced. Maybe move this to the methods-section or SI?

Reply: Not exactly. This is a brief introduction of the LOSCAR to the readers. We present the model in more detail in the SI (Text S4).

Lines 296-299: I would write this instead: "A total of 24,000 Gt CO₂ with an average δ¹³C of -18.8 ‰ was released in the model, following Heimdal et al. (2020). The carbon release is assumed to follow a Gaussian pattern during the 600-kyr-long eruption interval (Blackburn et al., 2013; Davies et al., 2017). N.B.: Note that Heimdal et al. (2020) argues for the release of 24,000 Gt carbon, not 24,000 Gt CO₂."

Reply: Modified. Many thanks for the suggestion. We rewrote the sentence, and we changed "CO₂" to "carbon" as suggested.

Lines 299-304: please rewrite, this sentence is too long.

Reply: Modified. we divided the sentence into two.

Line 308: specify Fig. 2D

Reply: Modified.

Lines 308-311: are you talking about your model results, or observed pCO₂ values? Or both? Please specify.

Reply: We refer to the various CO₂ levels yielded by our model. We made clear about it in the main text.

313: "... to 3 million years following the onset of the volcanic event."

Reply: Modified.

Lines 313-316: Please rewrite this sentence.

Reply: Modified. We rephrased this sentence.

Line 320: "... how it did during the Palaeocene–Eocene Thermal Maximum (PETM; Zeebe et al., 2009)." Also, how did the silicate weathering feedback function during the PETM? This is not obvious to the reader, unless they are experts on the PETM and NAIP.

Reply: We intended to convey the idea that, although massive basalt eruption will generally expose fresh basalts and thus increase silicate weathering, our model results actually don't support this conclusion. We tested several weatherability increases (1.1, 1.2, and 1.3) related to basalt extrusion, but the model results favor a change to just 1.1 (as discussed in the text). For the PETM simulations (Zeebe et al., 2009 as well as other classical PETM papers), silicate weatherability was held at 1.0 (i.e., no change) and the model ran well. Hence, we were stating that the T-J event also doesn't need a large change of weatherability to match the observed records, and that parameterization of weatherability in our model was similar to that for PETM simulations. However, to avoid confusion, we have deleted the PETM comparison.

Line 322: "given that basalt have tremendous capability to sequester carbon": please rewrite this sentence. Basalt does not directly sequester carbon, and also, the word tremendous should be replaced.

Reply: Thanks. We have modified the sentence to "given that fresh basalts react faster with atmospheric CO₂ compared to other silicate rock types".

Lines 316-326: This section is confusing to me. You have previously said that there was intensified chemical weathering during the T-J boundary (as evidenced by 187Os/188Os, 87Sr/86Sr, Hg data, CIA, clay compositions), and in the abstract you say "the period of enhanced chemical weathering persisted for ~ 2 million years consistent with model results of the time needed to drawdown atmospheric CO₂ levels", and in conclusions you say "simulations using the LOSCAR model show that intensified chemical weathering could have drawn down atmospheric CO₂ back down to its pre-extinction background level over an interval of ~1 to 3 million years". But in lines 316-326 you say that basalt emplacement did not drive significant shifts in weatherability, and that weathering rates are slowed down, and that there was a minor shift in the behavior of the silicate weathering feedback. I am not sure what your intention is with lines 316-326? Is it to say that you would have expected the need of even higher weathering rates? The important point is that the LOSCAR modeling suggests that high silicate weathering rates can be maintained for 1-3 million years, which agrees with the duration of the high CIA interval, and that the pCO₂ values are back to background levels within this time period. Right? Maybe move this part to the SI, or shorten it to a few sentences.

Reply: This is a very thoughtful suggestion. The reviewer is totally right. The main point for using the LOSCAR model in this paper is to support our observation that “high silicate weathering rates can be maintained for 1-3 million years, which agrees with the duration of the high CIA interval in HJG, and that the pCO₂ values return to background levels within this time period”. We are emphasizing the dilemma here for basalt weathering because we hope to offer insights on how silicate weathering behaves if the eruptive units are basalts instead of other silicate rock types. Many studies (e.g., Macdonald et al, 2019, Science; Jagoutz et al. 2016, PNAS) emphasized that basalt weathering consumes atmospheric CO₂ quickly and leads to global cooling. But it is still debatable (see caveat of Rugestein 2021 PNAS). As a test, we added an extra term k_{silw} to reflect this additional basalt effect, but our model shows that this basalt effect on weatherability is minimal; k_{silw} is within 1.1) and we think that the passivation effect prevents basalts from increasing weatherability dramatically. We regard this point as important to the reader, and that’s why we chose to keep these discussions in the main text. However, we have modified the text to make these points clearer at the reviewer’s suggestion.

Line 330-331: what do you mean by provide the first evidence for intense volcanism? In general, or from terrestrial sections in China?

Reply: We meant the first evidence of intense volcanism from terrestrial sections in China (line 332).

Line 346: you refer to the Ordos Basin, but in the following sentences you only mention the Junggar Basin (HJG section) and the Sichuan Basin (QLX section). Which section is found in the Ordos Basin?

Reply: We removed the Ordos basin here.

Lines 346-347: you use the word “detailed” twice in the same sentence.

Reply: Modified. We rephrase the second “detailed” with “well”

Line 353: “The section contains > 1 km thick deposits of the Upper...”

Reply: Modified.

Line 377: “are” well estimated

Reply: Modified.

Line 389: replace “following which”

Reply: We rephrased “following which” by “and then”

Line 413: "A subset of samples..."

Reply: Modified.

Line 433: What do you mean by "extra" data? What data is this? Also, use the word "additional" instead.

Reply: We rephrased "exact" by "additional". We mean any other background data including geochemical, sedimentary, and palynological data about the two sections we did, but not show in the present Ms.

Line 448: "... performed the field work"

Reply: Modified.

Line 614: it should be yellow stars, not yellow triangles

Reply: Modified.

Line 626: organic carbon isotopes should be A and F

Reply: Modified.

Line 629: move all abbreviations to the bottom.

Reply: Modified.

Line 631: "... and ages are from..."

Reply: Modified.

Line 634: "...green bar at the base of column C and H represents the"

Reply: Modified.

Line 635: "The arrow for the ICW represents the"

Reply: Modified.

Line 639: spell out "ME"

Reply: Modified. We added the full name of ME in the caption.

Line 639: it is confusing with the red color for both the volcanic source and also for samples with mercury enrichment. Also, what do the blue and purple circles represent?

Reply: Sorry for the confusion. The samples with red circles represent the mercury enrichment interval. They have two different sources as evidenced by Hg isotopes, including the increasing terrestrial sources (with increased Hg, but the similar negative Hg isotope to background), volcanic source (with increase Hg, but positive Hg isotope to the background). We made this clear in the caption of figure 3. We added statements about the gray, purple, and blue ellipses in the caption of figure 3.

Line 641: either use both abbreviations (HJG and QLX) or spell out both section names (Haojiagou and Qilixia).

Reply: Modified.

Line 644: "members"

Reply: Modified.

Line 644: "into" the system instead "to".

Reply: Modified.

Line 646: "See" the web version instead of "Refer to"

Reply: Modified.

Line 649: is it possible to add observed pCO₂ values? (for example in Schaller et al., 2011, 2012, in the supplementary material, there are tables with the pCO₂ values and "absolute time" in myr). It would be interesting to see the modeled pCO₂ curves plotted against observed values

Reply: Really good point. However, it is not necessary because the pCO₂ values in the models over the ~600 kyr of CAMP are from observed values (e.g., from 2000 to 5000 ppm, Schaller et al., 2011, 2012). The observed values certainly fit well to the values in the model. We stated this in lines 134-146 in the supplementary information.

Supplementary Information

Lines 36 -37: You mention a Dalongkou section. Is this supposed to be the Haojiagou section?

Reply: Sorry for the error. Yes, we refer to the Haojiagou section here.

Lines 54-58: please rewrite, this sentence is too long.

Reply: Modified.

Line 101: you refer to PETM boundary conditions. Do you mean that you used the paleo setup as opposed to the modern set-up of LOSCAR? If so, you should write that instead, because it makes no sense to use PETM conditions for the T-J boundary.

Reply: Yes, we used the paleo setup (i.e., PETM boundary conditions) instead of modern because the paleogeography near the TJB was more similar to that for the PETM. We note that there are still differences between the PETM and TJB boundary conditions. For example, the end-Triassic world featured one dominant ocean—the Panthalassic Ocean—whereas the configuration of LOSCAR for PETM has four oceans. However, as demonstrated by previous studies, variation in modeled $\delta^{13}\text{C}$ values for the surface layer of different oceans is small, indicating that the separate ocean boxes could represent the dominant epicontinental sea. So, we argue that the PETM setup with the modifications delineated in the text will likely capture the overall pattern of the carbon system during TJB. We have modified the text (lines 129-134) to clearly state that we used PETM boundary conditions.

Line 110: you say that you enhance the climate sensitivity to make the simulations more robust, and you do this by setting the temperature change for a doubling of atmospheric CO₂ to 3 °C. Is this the parameter TSNS? If so, please mention this, so it is easy to compare with those listed Table S1. Also, can you please explain why doing this makes the simulations more robust? That is not clear to me.

Reply: This climate sensitivity parameter is controlled by TSNS (0 means not using climate sensitivity, and 1 means we used the climate sensitivity, and in this case, it is 3 degree). So TSNS is a switch flag to denote whether we used the climate sensitivity or not. There are other parameters in LOSCAR to control the exact value of climate sensitivity (but 3 is the most commonly used one).

As a greenhouse gas, CO₂ is closely linked to the climate state. A higher CO₂ will lead to a higher surface temperature (assuming other conditions are the same). Without climate sensitivity being set up in LOSCAR, LOSCAR will assume no CO₂ influence on the temperature, and this is not correct. Hence, we turned on the climate sensitivity switch in our model. Now we have explicitly denoted this in the supplementary text: “we turned on the climate sensitivity (i.e., the temperature change for a doubling of atmospheric CO₂ was set to 3 °C; TSNS changed to 1 in Table S1)”.

Lines 115-116: There are other studies that suggest different values for both the carbon release magnitude and isotopic composition than Heimdal et al. (2020). It is fine to use the values from Heimdal et al. (2020), (I believe testing different emission scenarios is beyond the scope of the paper, so I understand why you chose to use values from just one study),

but I think you should mention why you chose this particular emission magnitude, as opposed to values from other studies. Why do you find the values from Heimdal et al. (2020) more realistic? Again, note that Heimdal et al. (2020) argues for the release of 24,000 Gt carbon, not 24,000 Gt CO₂.

Reply: Right. As the reviewer mentioned, testing different emission scenarios is beyond the scope of this paper. We chose Heimdal et al. (2020) mainly because they have tested a detailed emission scenario grounded by numerous complementary boundary conditions. Their model successfully replicated observed $p\text{CO}_2$ and $\delta^{13}\text{C}$ curves, and the carbon emission amounts are also in agreement with the thermal modeling results of previous studies. Hence, we adopted the carbon emission scenarios from Heimdal et al. (2020). Yes. Their estimate is 24,000 Gt carbon, not 24,000 Gt CO₂. Our model used 24,000 Gt carbon. We now modified the sentence to reflect that it is carbon, not CO₂.

Line 117: specify that it is “model time” 0. Also, I think you should add Davies et al. (2017), which provided additional high precision U-Pb ages of CAMP intrusives.

Reply: Modified. We now added Davies et al. (2017).

Lines 118-123: I do not fully understand the k_{silw} parameter. You say that this is multiplied with the “silicate weathering rate” in LOSCAR. What is the “silicate weathering rate”? Is this rate different from the nsi , and if so, how? Is the k_{silw} written into the LOSCAR code by you, or are you doing this multiplication “manually”? Why is the k_{silw} not listed in Table S1? You say that $k_{\text{silw}} = 1$ is unlikely to be the real case. Why? You say that k_{silw} was increased linearly from 1 to “higher values”. Do you mean increase to 1.1, 1.2 and 1.3? Please be specific.

Reply: The k_{silw} parameter represents the silicate weatherability. It is a measure of the collective reactivity of silicate rocks and is modified by a variety of factors, including the geographic distribution of lithology and topography, plant physiochemical responses, and total continental area, etc. Hence, basalt exposure might increase the k_{silw} term because fresh basalt (without considering the passivation effect) reacts faster with atmospheric CO₂ and will increase the silicate weathering rate. The default silicate weathering rate in LOSCAR is as follows:

$$F_{\text{si}} = F_{\text{si}0} * (p\text{CO}_2/p\text{CO}_2_0)^{nsi}.$$

Hence, no weatherability term is incorporated in the default weathering term. This k_{silw} is different from nsi as it is mainly a scaling factor and are not in the exponential part. The k_{silw} is manually added by us to the LOSCAR C code, and it will change through time.

When we fix k_{silw} as 1, it will take ~3 Myr from the system to be restored to its initial state following the carbon cycle perturbations, while our observation indicates ~1.6 Myr

restoration time, so we think k_{silw} as 1 is unlikely. By saying k_{silw} was increased linearly from 1 to “higher values”, we mean that it increases from 1 to 1, 1.1, 1.2 and 1.3, respectively. Each case is a different LOSCAR run. Now we have explicitly explained this in the supplementary text (Text S4).

Line 125: I would write “the” different instead of “those”.

Reply: Modified.

Line 138 and 153: Please upload figures with better resolution (Fig. S1 and S2)

Reply: Sorry for the lower resolution for Figures S1 and S2 (Fig. S2 and S3 in the new version). We upload the high-resolutions figures separate to the main text in the submit system.

Line 166: Spell out the abbreviation of “TS”

Reply: Modified. We spell out all abbreviations in the caption of figure S2.

Line 177: What details? I do not understand the connection between this and Figure 1. I’m not an expert on this, but I believe the recommended threshold for TOC concentrations in order to report Hg/TOC values is 0.2 wt.% (see Grasby et al., 2016; Jones et al., 2019). Is this taken into account?

Reply: Sorry for the typing error. We mean the vertical variation of CIA for the two sections in Figures S2 and S3 (new version). In answering the question raised by the first reviewer (pages 8-9 in this file), we discussed the effects on the Hg/TOC values by various TOC values in Text S2. We also added the raw Hg data to Figure 2.

Reviewer #3 (Remarks to the Author):

This manuscript investigates two new terrestrial records of the Triassic–Jurassic boundary for mercury concentrations and isotopes, before combining them with chemical index of alteration (CIA) data to report increased weathering rates during the TJ mass extinction. The exact role of weathering during this major event remains largely unknown (though widely speculated on), so this is certainly an interesting topic. Since the Hg isotope papers of Grasby et al. (2017, *Geology*) and Them et al. (2019, *EPSL*) I have been intrigued about the possibility of using Hg isotopes to chart increased runoff of terrestrial mercury during times of enhanced weathering, so personally, I welcome the approach being attempted here. Unfortunately, however, I think there are a number of issues with the study as it is currently presented, meaning that I cannot recommend it for publication in *Nature Communications*.

Reply: Many thanks for the positive view about the study topic and our integrated approach. “The exact role of weathering during this major event remains largely unknown (though widely speculated on), so this is certainly an interesting topic ” and “welcome the approach being attempted here”. We do not agree with the reviewer’s evaluation of the suitability of our study for publication in Nature Communications.

I have issues with both the carbon isotope and mercury data as currently presented. For the Qilixia section in particular, the proposed negative excursions in $\delta^{13}\text{C}$ are no greater (and are often lower) in magnitude than some of the ‘background’ variations between 0–250 m. It doesn’t help that there are large stratigraphic gaps in the dataset.

Reply: Fair point. The carbon isotope records in Qilixia are more complex from the Late Triassic to the Early Jurassic. There are many negative excursions in the Rhaetian background interval (e.g., ~ 125 m, and ~ 150 m), whose magnitudes are larger than the negative excursions near the T-J transition. However, organic carbon isotopes in terrestrial systems are complex and can be affected by many processes, such as carbon sources, $p\text{CO}_2$ -dependent fractionation factor, depositional environments (Hollander and McKenzie, 1991, *Geology*; Oehlert and Swart, 2014, *NC*). The magnitude of carbon isotope negative excursions near the T-J transitions are various, ranging from 1 ‰ to 8 ‰ in different settings (summarized by Ruhl et al., 2020, *ESR*). Although we do not know the exact cause for the large negative excursions of C isotopes in the background interval, they were likely due to other factors. This uncertainty does not affect the main conclusions of the present work, which are based on the 2-3 ‰ negative carbon isotope excursions within the T-J transition.

For the Hg/TOC, the authors don’t present the raw Hg concentrations, only the Hg/TOC ratios, but from looking at the supplementary files it is clear that there is no increase in Hg concentrations in the strata that show high Hg/TOC ratios. In fact, in both sections the Hg/TOC peaks appear to be largely caused by the fact that those sediments have the lowest TOC contents in the section. Indeed, many of the strata that record the Hg/TOC peaks feature TOC contents less than 0.2 wt%, which has been argued by Grasby et al. (2016, *Geological Magazine*) to be the cutoff for reliably normalizing Hg against TOC due to the high percentage uncertainty on those low values. If samples for which $\text{TOC} < 0.2 \text{ wt}\%$ are removed from the Hg/TOC data plots, following the protocol of Grasby et al., then the Hg/TOC peaks either disappear (for Haojiagou) or are reduced such that they are similar to background variability (for Qilixia). Thus, I’m not convinced that the postulated mercury elevations are actually genuine, and they’re certainly not as robust as the authors claim. These authors have published several papers on mercury before now, so I am surprised that they are presenting the data in this way. I know that some workers in the mercury community are skeptical about the 0.2 wt% cutoff proposed by Grasby et al., but if the authors are going to adopt this stance, then they have to be forthright and say this. And certainly include the Hg concentrations in the main figure.

Reply: We answered a similar question from the first reviewer (pages 8-9 in this file). Briefly, First, we added raw Hg concentrations to figure 2 (B, H). Second, we separated Hg/TOC by different symbols according to TOC content (e.g., < 0.2%, versus $\geq 0.2\%$, Fig. 2C, J). We discussed the effects on Hg/TOC values by variations of TOC in lines 186-194 in the main text and lines Text S2 in the supplementary text.

The main conclusion of this manuscript is that the combined CIA and Hg isotope data document an increase in weathering (and runoff of Hg) during the TJ extinction event, which I agree with, and that the correlation with the Hg peak ties this enrichment to the CAMP volcanism. Even assuming that this enrichment is true (see above), the authors make no reference to the main conclusion of Them et al.'s 2019 paper on the Toarcian. In that paper, a similar Hg/TOC peak correlative with negative MIF values was also interpreted as showing increased runoff of terrestrial Hg, but crucially, whilst the authors did not rule out the possibility that the Hg was originally derived from coeval Karoo-Ferrar volcanism but had been recycled via the terrestrial realm, there was no proof of a volcanic source. Why should these two TJ records be any different? How can the authors be certain that any Hg increase is not simply the result of the enhanced weathering, completely independent of any CAMP emissions. The Them et al. paper is cited elsewhere in this manuscript, so the authors are clearly aware of the study, and I am surprised (again) that it is not discussed more fully.

Reply: Fair points. We appreciate the reviewer's positive view of the main conclusions about "combined CIA and Hg isotope data document an increase in weathering (and runoff of Hg) during the TJ extinction event, and that the correlation with the Hg peak ties this enrichment to the CAMP volcanism".

We fully agree with the conclusions in Them's paper, and we cannot rule out Hg sources from volcanism to the continental baseline on our data. However, this does not affect our main conclusions (i.e., that Hg records intense chemical weathering and volcanism). The Hg spikes are likely to have been sourced from recycled Hg of the CAMP, as suggested by Them for the T-OAE, as well as Hg stored in continental reservoirs (e.g., soil). However, in the present work, $\Delta^{199}\text{Hg}$ values during the mass extinction interval can be divided into two groups, one group having similar values to the background interval, and the other one having more positive $\Delta^{199}\text{Hg}$ values. We interpret a dominant Hg source from the continents for the first group, and Hg from CAMP volcanic releases for the second group (see the detailed replies to this issue in pages 9-11 of this file). We made this clear in lines 212-216 and 231-246 and cite Them's work in lines 83 and 89.

Finally, I think that in places the authors are overselling the novelty of their paper. To clarify, their methodological approach of combining Hg isotopes and CIA on TJ records is novel, but there are other statements in parts of the text that seem to be trying to imply that this study is more novel than it actually is. For example, mercury is repeatedly stated to be a 'novel' proxy. In the last 5 years there has been a huge body of work published

using mercury to study LIP volcanism (including 4 or 5 works on the TJ alone), so I don't think it can be classed as that novel a tool any more. The statement that Hg data do not exist at sites distal from CAMP is incorrect: I would hardly describe any of Nevada, Astartekløft, or the Argentinian sites as being proximal to CAMP, so this study is not presenting the first data from a distal site. The implication that links between volcanism and weathering have not been robustly determined is also misleading. A number of studies have used a range of techniques to show that the CAMP basalts were rapidly weathered following their eruption (Cohen and Coe, 2002, 2007; Kuroda et al., 2010; Palfy and Zajzon, 2012), so it is clear that more weathering on the continents (at least in that area) was taking place immediately following the eruptions, and directly in response to the emplacement of those basalts. Schaller et al. (2012) even proposes this as a cause of CO₂ drawdown post event. I acknowledge that this does not prove a link between volcanism and weathering away from the CAMP, but that's not how the statement currently reads in my view.

Reply: Although we appreciate the positive comment: "their methodological approach of combining Hg isotopes and CIA on TJ records is novel", we disagree with the following points.

About the novelty of the study. We believe that the novelty of our study includes several aspects: 1) this is the first study of volcanic Hg inputs to terrestrial facies from the eastern margin of Tethys, which is the most distal of 17 settings yielding Hg data for the CAMP to date (even including recently reported sections in Yager et al. (2021)). 2) we integrated carbon cycle (carbon isotopes), volcanism (Hg data) and weathering (Hg, CIA and clay mineralogy) data from the same samples of two terrestrial sections to investigate cause-and-effect relationships between CAMP, chemical weathering and the carbon cycle around the T-J boundary. 3) we used carbon cycle modelling (LOSCAR) to test whether or not increased continental chemical weathering due to emplacement of CAMP can explain the observed drawdown of atmospheric CO₂ at timescales consistent with observations of intense chemical weathering at HJG.

Furthermore, we fully agree with the statements about the novelty and importance of this work from the first and second reviewers.

As the first reviewer said, "The question, I think, is not whether CAMP emissions changed the global climate and enhanced weathering and erosion, but how and to what extent in different areas. **Here, your manuscript can be an important contribution.**" (page 2 in this file).

As the second reviewer said: 1) "This paper integrates a number of different datasets and methods in order to investigate the cause-and-effect relationships between CAMP, chemical weathering and the carbon cycle around the T-J boundary. They examine two new terrestrial sections (both high and low latitude) from China where various proxies can be analyzed, which allows to directly compare carbon cycle perturbations (carbon

isotopes), volcanism (Hg data) and weathering (Hg, CIA and clay mineralogy). One issue with tying together the relationship between CAMP, the end-Triassic mass extinction and the associated carbon cycle perturbations, is that the various proxies are from different sections, and correlating these have proven to be difficult. For example, CAMP lavas are emplaced in continental deposits, but most C-isotopic records are from marine sections. Furthermore, there exists different interpretations of the isotope data considering the timing of carbon cycle perturbations and CAMP activity (e.g., Korte et al., 2019 vs. Lindström et al., 2017). Producing data of proxies for weathering, volcanism and carbon cycle perturbations from the same section prevents the issue of correlating, and is therefore of great importance for the end-Triassic, where these relationships are yet not fully understood. **This point alone provides novelty to this contribution. In addition, they use carbon cycle modelling to test whether or not increased weathering due to the emplacement of CAMP can explain the observed drawdown of atmospheric CO₂ within the same timeframe as the observed intense weathering interval. I believe the main goal combined with the used methods and approaches, provides novelty to this contribution.**” (pages 13-14 in this file) 2) “there is already strong evidence for weathering, volcanism, and carbon cycle changes in numerous settings near TJB, but the problem is that the different datasets must be correlated between sections. And this has been proven quite difficult. However, **the sections in the present work represents an opportunity to provide all these different proxies without worrying about correlations.**” (page 17 in this file)

Secondly, Hg concentrations for the TJB may be not “novel” (17 sections in 7 studies as shown in figure 1) as the reviewer said. However, elevated Hg in sediments can result from many processes (e.g., volcanism, terrestrial inputs, local redox conditions), resulting in complex causation between Hg peaks and volcanism (e.g., Percival et al., 2018, AJS; Them et al., 2019, EPSL; Shen et al., 2020, EPSL). Mercury isotopes are a promising tool to track Hg sources (Blum et al., 2014), and would be useful to reconstruct sources and cycle processes of Hg (we made clear in lines 74-90). Our work is the first time to provide Hg isotope for two continental TJB sections (this is likely to be novel), because Hg isotopes have been reported from only one marine setting when we submitted this Ms (Thibodeau et al., 2016, NC), although Hg isotopes have recently been reported for two marine settings (Yager et al., 2021, ESR).

Thirdly, we stated the two sections of the present study are distal from CAMP because they are located on the eastern and northern margins of the Tethys. The sections (Nevada, Astartekløft, or the Argentinian) mentioned by the reviewer are indeed far from the CAMP to some degree, but they are still relative closer to the CAMP compared to our study sections for they are located on the western margin of Tethys as shown in figure 1. The TJB witnessed environmental perturbations and a biological crisis at a global scale, which has been linked to the CAMP. We would like to test the hypothesis that weathering of the CAMP record is distributed at a global scale. Our study provides volcanic records in terrestrial settings from the eastern and northern margins of Tethys.

Lastly, few works have focused on rapid weathering following the CAMP eruptions. However, as the second reviewer said, “there is already strong evidence for weathering, volcanism, and carbon cycle changes in numerous settings near TJB, but the problem is that the different datasets must be correlated between sections. And this has proven quite difficult. However, the sections in the present work represent an opportunity to provide all these different proxies without worrying about correlations.” (page 16 in this file). The present study proves: 1) these intense chemical weathering records are also exhibited in the settings from the eastern and northern margins of Tethys; 2) we had the high-resolution timeframe about the onset and termination of intense chemical weathering; 3) we correlated the cause-and-effect relationships among volcanism, chemical weathering, and carbon cycles.

MINOR COMMENTS:

L. 55: just the eruption of massive lavas? It’s been argued by a few studies now that the intrusive magmas (and volatiles released from sediments that they were emplaced into) were potentially the more significant trigger (e.g., Svensen et al., 2009; Heimdal et al., 2018). And the dates of some of the low Ti sills that intrude the Amazonas Basin (which is full of evaporites as well as other sedimentary rocks) match the extinction date almost perfectly (Davies et al., 2017), whereas most/all lavas postdate the onset of the extinction.

Reply: Fair point. We modified the statements here to involve the emplacement of the sills by CAMP, and cited Davies’s paper here.

L. 63: Several workers are not wholly convinced as to how reliably quantitative the CO₂ estimates based on stomatal index data or pedogenic carbonates truly are. I think the general trends (of an increase across the TJ transition) and timing are not in doubt, just the precise values. So it may be safer to leave the 4000ppm out.

Reply: We agree with that it’s still under debate about conversion of stomatal indices to quantitative values of CO₂ concentration. However, as the reviewer said, it is not in doubt that the CO₂ increased in general across the TJB. We removed “4000 ppm” here.

L. 67–72: The idea that weathering potentially links volcanism and marine environmental perturbations in this way greatly predates the works of Penman et al. (2020) or Shen et al. (2015). The broad brushstrokes of this mechanism are mentioned by both Algeo et al. (2011, Palaeo-3) and Jenkyns (2010, G-cubed), and various aspects of it had been discussed before then (e.g., Cohen et al., 2004). Earlier citations would be better.

Reply: Fair point. We updated the references here.

L. 75–78: See earlier point. There’s lots of evidence linking CAMP volcanism with

weathering, by the fact that the basalts were intensely weathered soon after eruption. If the authors are indicating that we don't have evidence for increased weathering on a global scale, away from the CAMP basalts themselves, then they must state this more clearly.

Reply: The evidences linking CAMP with weathering are largely inferential owing to lack of a volcanic proxy in sedimentary rocks. Our study is the first to provide integrated data including volcanic records (Hg), carbon cycles (carbon isotope), and chemical weathering (Hg, CIA, and clay minerals) in two sections to investigate the cause-and-effect link of volcanic effects to the carbon cycle and chemical weathering (see the detailed replies about this issue in page 33-35 in this file). We modified this sentence to make it clearer.

L. 79: Hg isn't really that novel these days.

Reply: As in the replies above (page 33-35 in this file), we removed the statements of "a novel volcanic proxy in sediments".

L. 93–94: In so far as we can tell from the (very) limited studies on modern volcanoes. Most (if not all) of which were on arc volcanoes I believe, which are hardly analogous with LIPs.

Reply: Fair points. Yes, most of the Hg isotope studies of modern volcanism were for arc-volcanoes (e.g., Smith et al., 2008, EPSL; Sherman et al., 2009, EPSL; Yin et al., 2016, Scientific Reports). However, recent works contain many $\Delta^{199}\text{Hg}$ values from cratonic volcanism as well as mantle materials, which are similar to LIPs (e.g., Smith et al., 2008, EPSL; Deng et al., 2021, Geology). It's indeed hard to know the exact $\Delta^{199}\text{Hg}$ values of ancient LIPs. However, based on the Hg isotope records from the three marine settings (Thibodeau et al., 2016; Yager et al., 2021), the near-zero $\Delta^{199}\text{Hg}$ values associated with higher Hg enrichments near the TJB are evidence that the Hg isotopes of CAMP were likely to be near zero.

L. 95–97: And the Toarcian OAE (Them et al., 2019).

Reply: Modified. We added "The Toarcian OAE (Them et al., 2019)" here.

L. 100: See point above. I would say that all of Nevada, Astartekløft, and the Argentinian sites are distal from the CAMP. Yes, none of them are from the eastern margin of the Tethys, but that's not the only place in the world distal from the CAMP.

Reply: We are saying only that the present two section are more distal from CAMP than these sections (see detailed replies in page 33-35 in this file).

L. 110: Looking at Figure 1, Qilixia appears to be north of 30N degrees. I would suggest that this is mid latitude rather than low latitude.

Reply: The placement of each site in the geographic map of each craton is approximate. However, we modified the paleo-latitude of QLX to “low/middle latitude” rather than “low latitude” in the text as the review suggested.

L. 115–116: I’m not at all convinced by the proposed PCIE, which consists of only 2 or 3 data points, and is the same order of magnitude as many background variations, although admittedly the authors do have a question mark next to it. Also the clearest excursion of all at Haojiagou (260–360 m) is ignored. I know that it’s in Sinemurian strata, and thus too late to be connected to the TJ extinction, but to avoid confusing the reader, the authors should make clear that they are talking about CIEs close to the TJ boundary in the results section.

Reply: Organic carbon isotope records are complex and subject to many influences such as carbon sources, fractionation, depositional environments (see detailed replies in pages 33-35 of this file). So negative CIEs are complicated to explain. Yes, the samples for the PCIE are few but a similar trend of CIEs in the two study sections as well as in other sections, although varying in magnitude, is likely to represent the original signal for volcanism at a global scale. Furthermore, we provide evidence of volcanic inputs based on not only carbon isotopes, but also mercury records. We added the thickness and magnitude of each CIE in both sections to make clear that we are referring to the CIEs near the TJB (lines 114-118 and lines 133-136 for HJG and QLX, respectively).

L. 129–131: See earlier point. There is a lot of variation in the Qilixia $\delta^{13}\text{C}$ background, and the proposed CIEs aren’t really any greater in magnitude. Not convincing for me.

Reply: See the replies above as well as in page 33-35 in this file.

L. 153: See earlier point about how reliable these numbers are believed to be.

Reply: See the replies above (page 33-35 in this file).

L. 160: Wildfires are certainly a source of Hg to the surface environment in the modern day, and it has been suggested that this could also have been the case during major events (Them et al., 2019). Is there any evidence for charcoal in these terrestrial sections?

Reply: Yes, wildfires are likely to be a significant source of Hg to the environment. However, the $\Delta^{199}\text{Hg}$ values of released Hg from combustion of terrestrial plants are similar to the terrestrial inputs in river sediments. These values (both of them yield negative $\Delta^{199}\text{Hg}$) are significantly different from the $\Delta^{199}\text{Hg}$ values of atmospheric sources (positive values). Release of Hg by wildfires will not affect the main conclusions of the present work.

L. 176: See earlier point about the novelty of Hg.

Reply: Modified. we rephrased the sentence.

L. 193: Cite these other studies.

Reply: Modified. We cited a review paper (Grasby et al., 2019, ESR) here because of the NC policy of a limited number of references.

L. 200–204: Blum et al. (2014) state that the plot of ^{199}Hg vs ^{201}Hg MIF is appropriate for samples with MIF of greater than ± 0.3 . Most of the samples in this study do not. So is this interpretation valid? Also, my understanding is that the normal line of evidence for atmospheric transport and deposition of Hg is positive MIF. So how do these two datasets, in which MIF values are overwhelmingly negative, support atmospheric transport? Either the authors' interpretation is wrong, or I'm missing a step in their reasoning. Either way, their logic needs to be made clear.

Reply: We agree with Blum et al. (2014) that large MIF of ^{199}Hg and ^{201}Hg allow more precise determination of $\Delta^{199}\text{Hg}/\Delta^{201}\text{Hg}$ ratios. However, with small MIF of $\Delta^{199}\text{Hg}$ and $\Delta^{201}\text{Hg}$ ($< \pm 0.3\text{‰}$), the $\Delta^{199}\text{Hg}/\Delta^{201}\text{Hg}$ ratio can also provide important information regarding the mechanism of Hg-MIF in natural samples, as reported in many studies (e.g., Yin et al., 2016, Scientific Reports; Du et al., 2018, EST; Xu et al., 2019, CG).

In this study, our samples show $\Delta^{199}\text{Hg}/\Delta^{201}\text{Hg}$ ranging from 1.0 to 1.36, which is similar to that observed during photochemical reduction of aqueous Hg(II) driven by dissolved organic matter, ($\Delta^{199}\text{Hg}/\Delta^{201}\text{Hg} = 1.0 - 1.3$, Bergquist and Blum, 2007; Zheng and Hintlemann, 2009). Aqueous Hg(II) photoreduction produces negative $\Delta^{199}\text{Hg}$ in the gaseous Hg(0) phase, leaving the residual Hg(II) phase with positive $\Delta^{199}\text{Hg}$. Terrestrial reservoirs (e.g., soil and vegetation) mainly accumulate Hg(0), therefore are characterized by negative $\Delta^{199}\text{Hg}$. Marine reservoirs (seawater and sediments) mainly receive Hg from atmospheric Hg(II) deposition, therefore normally show positive $\Delta^{199}\text{Hg}$.

During the TJB, the marine settings have relatively positive $\Delta^{199}\text{Hg}$ (Thibodeau et al., 2016; Yager et al., 2021), which was similar to modern marine sediments (Blum et al., 2014). The $\Delta^{199}\text{Hg}$ values are lower at HJG and QLX than in relatively deep-water offshore settings (e.g., Nevada and Levanto, Thibodeau et al., 2016; Yager et al., 2021), which was likely to be result from receiving Hg dominantly from seawater (thus, positive $\Delta^{199}\text{Hg}$). However, the $\Delta^{199}\text{Hg}$ values in the present two sections are in a similar range to those of nearshore settings (St Audries Bay, Yager et al., 2021), which received more terrestrial-sourced Hg.

The present study sections are terrestrial and located far from CAMP. The Hg proportion from terrestrial inputs is higher than that from atmospheric inputs, resulting in a mixing negative $\Delta^{199}\text{Hg}$ values. Integrated use of Hg concentrations and isotopes could help to track Hg sources in sediments. In the present study, the $\Delta^{199}\text{Hg}$ values during the high-Hg

interval can be divided into two groups (Fig. 3A, 3B), one group having $\Delta^{199}\text{Hg}$ values similar to the background samples, which show the similar sources of the elevated Hg (e.g., terrestrial reservoirs) (blue arrow dashed lines in Fig. 3A, 3B). The other group have more positive values (close or a little higher than zero) evidence atmosphere sources, which could be related to volcanic eruption (purple arrow dashed lines in Fig. 3A, 3B). See the detailed replies in pages 10-11 in this file. We made this clear in lines 231-246 in the main text.

L. 212: change 'sections' to 'section'. Only one TJ section has been previously studied for Hg isotopes.

Reply: Modified.

L. 229–230: The rise in Os isotope values documented by Cohen and Coe (2007) continues into the Sinemurian. Also, whilst it might represent a long-term increase in continental weathering rates, it could equally reflect a reduction in the levels of primitive basalts being weathered (if, for example, the amount of CAMP available to be weathered was gradually declining over time, which is very likely to have occurred).

Reply: Good point. The Os isotope records, which evidence intense chemical weathering of primitive basalts of CAMP during the Hettangian and ceased at the start of the Sinemurian, also agree with the CIA records in the HJG section (higher values during Hettangian, decreased at the start of the Sinemurian).

L. 239–241: Why does elevated Hg coupled with limited MIF variation relative to background signify a large flux of terrestrial Hg to the sediments? In both Grasby et al. (2017) and Them et al. (2019), terrestrial Hg input is inferred from a peak in Hg combined with a large change in MIF to more negative values.

Reply: Great point. In Grasby and Them's works, the background sediments are marine sediments, which consist of positive $\Delta^{199}\text{Hg}$ values (Blum et al., 2014; Yin et al., 2016). Increasing of terrestrial inputs (having negative $\Delta^{199}\text{Hg}$ values) near the event horizons, result in higher Hg concentrations and more negative values in sediments. However, for the present two sections, the sediments are terrestrial for the upper Triassic background interval, yielding negative $\Delta^{199}\text{Hg}$. Increasing continental weathering near the TJB would bring a larger amount of Hg to sediments, which is characterized by higher Hg concentrations and similar Hg isotopes to the background intervals. See the replies to similar comments in pages 10-11 of this file.

L. 254: Something is missing in the brackets here.

Reply: We added the r values for the correlation co-efficient between CIA and $\Delta^{199}\text{Hg}$ for HJG.

L. 266–296: Alternatively, it could just reflect a large input of terrestrial Hg to these areas, completely independently of any volcanism (following the model of Them et al., 2019). The authors should discuss in more detail why they have ruled out this option, and cite the 2019 paper appropriately.

Reply: As shown in figure 3A, 3B, the $\Delta^{199}\text{Hg}$ values during the higher Hg interval can be divided into two groups (Fig. 3A, 3B), one group having $\Delta^{199}\text{Hg}$ values similar to the background samples, which show similar sources of elevated Hg (e.g., terrestrial reservoirs) (e.g., blue arrow dashed lines in Fig. 3A, 3B). The other group has more positive values (close to or a little higher than zero), which evidences atmospheric sources, possibly related to volcanic emissions (purple arrow dashed lines in Fig. 3A, 3B). The latter cannot be explained by elevated terrestrial Hg inputs. See the replies to similar comments in pages 10-11 and 35 of this file.

We cited Them`s paper in lines Them`s work in lines 83 and 89 in the main text.

L. 280: Change 'toady' to 'today'.

Reply: Modified.

L. 288: 'organic matter' or 'organic carbon' would be better than just 'organic'.

Reply: We deleted this sentence as suggested by the second reviewer.

L. 314: Strictly speaking, McElwain et al. (1999), calculated CO₂ concentrations based on stomatal index data, and interpreted the likely temperature change based on those values. They didn't present direct evidence of temperature changes per say. As far as I'm aware, there is actually no palaeotemperature record that directly shows warming at the onset of the TJ extinction, just lots of evidence for increased atmospheric carbon, which presumably would have caused warming.

Reply: Yes, we agree that there are no direct proxies (e.g., oxygen isotopes in conodont) for the magnitude of temperature variations near the TJB. However, as the reviewer said, there were many evidences showing increased atmosphere CO₂, which likely caused warming. This debate does not affect the main conclusions of the present work.

L. 318: Change to 'Despite the fact that...'

Reply: Modified.

L. 320: Change 'how it did in' to 'during', and 'event' to 'events'.

Reply: Modified. We rephrased them.

L. 322: 'basalt' should be 'basalts'.

Reply: Modified.

L. 336–338: How exactly does the weathering cause the oceanic perturbations that triggered biotic stress? I don't think this has been mentioned in the discussion at least. Are the authors invoking the model of weathering = nutrient runoff = eutrophication and anoxia? The problem with this is that evidence for anoxia during the TJ is patchy at best. It definitely doesn't seem to be as widespread as during the PT extinction or the later OAEs, and a lot of places only seem to start recording black shales in the earliest Jurassic, too late to have been the cause of biotic stress.

Reply: Good points. Yes, the volcanism-induced intensification of chemical weathering could increase nutrient runoff, and then induce blooms of oceanic productivity as proposed by many authors (e.g., Wignall, 2001, ESR; Algeo et al., 2011, Geology). However, besides this, increasing terrestrial runoff would also cause oceanic perturbations by increasing the harmful effects of siltation, elevated turbidity, etc., which can cause a reduction in feeding activity, osmoregulation, growth rate, body size, larval recruitment for organisms (e.g., Algeo and Twitchett, 2010, Geology).

We do not agree with the statement that anoxia during TJB was patchy at best. Black shale deposition (van de Schootbrugge et al., 2013), as well as photic zone euxinia (Richoz et al., 2012, NG; Kasprak et al., 2015, Geology) was widespread near the TJB. Furthermore, carbonate U isotopes, which serve as a global redox proxy, exhibit a ~ 0.7‰ negative excursion, equivalent to a 40-100-fold increase in the extent of oceanic anoxic deposition worldwide near the TJB (Jost et al., 2017, GGG). The increase of anoxia (40-100-fold) near TJB is even greater than that (6-fold increase) during PTB.

L. 340–342: How can this model of drawing down CO₂ over 1–3 million years conform with evidence of weathering from the two sections, given that only one of the sections (Haojiagou) has data from sediments deposited that amount of time after the event? Reword.

Reply: Fair point. We mentioned these data only from one section.

Figure 1 and caption: Change 'Lgounane' to 'Igounane'. Check throughout the manuscript.

Reply: Sorry for the typing error. Modified. We updated the section name in the main text and figures.

Reviewers' Comments:

Reviewer #1:

Remarks to the Author:

Dear authors

It is nice to see that you have taken the majority of my comments and suggestions under consideration and made the necessary changes.

I think the revised manuscript is greatly improved. However, there are still a few things that need to be considered. You will find these below.

Best regards

Sofie Lindström

There is something seriously wrong with your Fig. 1, at least in the merged PDF version, where everything except the background palaeomap from Blakey is misplaced. St. Audrie's Bay is misspelled. Stenlille-1 and Stenlille-4 are actually located very close to each other. At this scale, representing them with one dot (as in N Albert/Albert-1) would be sufficient. Also, specify that you mean Hg-isotopes.

Figures S1 and S2. Hettangian is misspelled.

Figure S1. For the Qilixia section: Why are the spore-pollen taxa presented only as genera? This is fairly useless information for biostratigraphical purposes.

Line 178. The two papers referenced here do not testify to the global recognition of these CIEs. You should refer to a paper that CIEs globally. You could e.g. use: Lindström et al. (2021) or Fujisaki et al. (2018), which presents some different correlations but at least on a global scale.

Line 568. Lindström is misspelled.

Line 642. St. Audrie's Bay is misspelled. Check throughout!

Reviewer #2:

Remarks to the Author:

This is a second review of the submitted manuscript by J. Shen and co-authors. I would like to thank the authors for their very thorough response to my previous comments. In the revised version, the introduction now clearly states what the current knowledge gaps are, and what this study actually did in order to close them. The discussion is now much easier to follow, and the different results are better tied together. The section about the LOSCAR modeling is now improved, especially in terms of understanding how the modeling was performed and what the model results suggest. Furthermore, in the first review, all three reviewers were concerned about low TOC concentrations. In the revised version my opinion is that the authors have satisfactorily addressed this, both in the discussion and supplementary file but also by clearly marking the </> 0.2 wt.% TOC data points in Fig. 2. I am happy to recommend this manuscript for publication in Nature Communications.

After reading the revised manuscript I have a few minor comments:

Lines 57-58: I would re-write this sentence. The temporal link strengthens the case for a causal link between CAMP and the ETE, but the temporal link alone is not the only reason why it has been suggested that CAMP was involved in causing the ETE.

Lines 65-68: I would delete "through weathering of silicate rocks". Because now you are saying that "...weathering... can... lower atmospheric CO2 concentrations through weathering".

71: I would delete "although it is necessary".

74: I would delete the comma

101: "cyclev": delete the v

192: I would write: "...both this and earlier studies document..."

311: I would delete "from the models".

Fig. 4: I would write what the dashed horizontal lines represent in the figure caption.

Table S1: typo: "initial seeady-state pCO₂"

Reviewer #3:

Remarks to the Author:

This is a revised version of a study investigating the relationship between changes in volcanism, the global carbon cycle, and silicate weathering during the Triassic-Jurassic extinction. I think that the manuscript has been improved since the previous submission, and a number of the points are more clearly made, particularly regarding the LOSCAR modelling. Combining multiple proxies within a single sedimentary record is an important tool for interrogating past episodes of climate change, as it negates the need to stratigraphically correlate across multiple regions. Thus, this study has the potential to be an important contribution to our understanding of the TJ extinction.

However, I still think there are things that need to be addressed with this manuscript. Most unfortunately, I think that there are points raised by both myself and another reviewer in the previous set of comments that have not been addressed to my satisfaction. Two reviewers noted in the last version that the high Hg/TOC peak at Haojiagou is largely driven by very low TOC, and the peaks at Qilixia are at least partly caused by the same issue, but whilst the authors have now added text discussing the host phase of Hg in these sediments, I cannot find a sentence anywhere in the main text simply stating that the TOC is very low in some of the samples that have high Hg/TOC and that this may hinder interpretation of these peaks. There are two short sentences in the supplementary text acknowledging this, but these need to be moved to the main manuscript. Additionally, the authors acknowledged in their response to my previous review that the $\delta^{13}\text{C}$ trends are complex to interpret. That's fine, and I agree with them. But I would then expect to find some new text in the manuscript outlining these complexities and how the authors have addressed them. But the original claim that there are three clear $\delta^{13}\text{C}$ excursions (albeit one with a question-mark attached) is still present, and the nuances undiscussed.

There are also further details in the supplementary information that I would prefer to see in the main text. The added information regarding section correlation and palynology requested by another reviewer is mentioned briefly in the Materials and Methods section, but there is important information in the supplementary text that would be good to have in the main manuscript. A number of figures pertaining to the clay-mineralogy evidence of weathering would also be good to have included. I appreciate that there are major space constraints for Nature Communications, and that asking for these details to be added, in addition to those above, would be challenging. I hate to say this, given that this is the 2nd or 3rd iteration of the manuscript and I really would understand the authors' frustration to have to start again, but I wonder if a slightly longer-format journal would allow the complexities with stratigraphic correlation and geochemical interpretations to be better explained, strengthening the arguments presented.

A final thought. The two sites studied here both have age models based on biostratigraphy and astrochronology. There are also robust age models for Levanto and St Audries Bay, which have now also been studied for $\delta^{13}\text{C}$, Hg, and Hg isotopes (I don't think the Yager et al. paper was available online when I last reviewed this manuscript). It would be interesting to have a figure comparing the trends across the four sections vs time, rather than depth. Perhaps also with the timing of CAMP activity also marked. Just an idea.

In summary, the novelty of this study is coming across more clearly now than it was in the previous

version, and the manuscript is certainly improved. But there are some comments from last time that have not been fully addressed, and I think that further details in the main text are needed regarding some issues. I recommend moderate revisions, largely consisting of more detailed explanations regarding the various complexities outlined above in the main manuscript.

MINOR COMMENTS:

L. 37: Change to 'We interpret these results...'. Do not use an unqualified 'this'. Check throughout.

L. 41–43: This initially felt counter-intuitive to me. I thought that lower latitude sites were generally more susceptible to chemical weathering, due to the warmer, and generally wetter, climate. Following a second reading, I assume that the point being made is that at higher latitudes, there will be a greater and more rapid change in temperature, and consequently also in weathering rates. I'd suggest rephrasing to make this point more clear.

L. 54: Change 'as' to 'to have been'.

L. 59–60: 'carbon' appears twice within three words, and then a third time later in the same sentence. Rephrase.

L. 61: Increased CO₂ concentrations in what? Clarify.

L. 74: Comma issue. It should either be 'Mercury (Hg) concentrations, and isotopes, are widely used...' or 'Mercury (Hg) concentrations and isotopes are widely used...'.

L. 75–76: I'm not sure that it's necessary to describe mercury as a naturally occurring element. I think that what is meant is that elemental mercury occurs naturally in gaseous form in the atmosphere. If this is the case, then I'd suggest rephrasing the start of this sentence to make it clear.

L. 80: If we're being honest, it's more true to say that during large

L. 92–93: I'm still not keen on this phraseology regarding the distality of the two new sites. I agree with the author's response to a previous comment that these sites do give a more global overview of the TJ Hg cycle, and I completely endorse that stance. But why not then simply say that. I.e., '...these records are largely concentrated around central Pangea. However, data from the eastern margin of the Tethys Ocean, which would enable a truly global-scale overview, are lacking.'

L. 97–98: Does this not depend on the specific volcanic effect? I agree that the terrestrial realm will be more impacted by volcanic warming, but presumably other effects such as aquatic acidification (heavily implicated in the TJ extinction) will be at least as equally strong in the marine realm.

L. 116: This statement is not actually true. If these three shifts in $\delta^{13}C$ at Haojiagou are considered to be excursions, then there are also excursions at 250–360 m and arguably below -100 m and -80 to -50 m. The point is that these are the three excursions (although the PCIE is dubious) around the TJ boundary.

The same is true of Qilixia. One could certainly argue that there is an excursion at around 150 m.

L. 118–119: It looks like several samples have Hg contents above 60 ppb throughout the section, with a few peaks of around 80–120 ppb, as the authors say. So I'm not sure why a 'range from 10 to 60 ppb' is stated.

L. 120–123: Yes, these Hg/TOC peaks are present, but nowhere is it stated that the higher values

within the proposed Initial CIE and Main CIE strata are from samples featuring very low TOC. I know that this is shown in Figure 2, and briefly in the supplementary information, but this is an important caveat to the peaks that should be explicitly mentioned in the main text, probably together with a citation of Grasby et al.'s 2016 recommendation of not normalizing against such low TOC values.

L. 133–136: From looking at Figure 2, it appears more like the uppermost Triassic 'background' values range from -27 to -24 per mil, rather than -25 to -23. The proposed PCIE excursion has values that fall entirely within that range. The other two CIEs are more convincing. But what concerns me most is the huge gaps in the upper Triassic dataset. How do we know that there aren't other variations in $\delta^{13}\text{C}$ within those gaps, which would make the proposed TJ excursions (especially the PCIE) look like noise? I appreciate that samples aren't always attainable, but this is a big limitation with the dataset that the authors simply must acknowledge.

L. 136: 'Mercury contents are lower at the upper Triassic'. It should be 'in the upper Triassic sediments' (or strata, either will do). And what are the Hg contents lower than? Clarify.

L. 139–141: Again, there is no mention of the fact that some (although not all, for this site) of the high Hg/TOC values are from samples featuring very low TOC contents.

Also, it is not actually true to say that the background values are 'uniformly' low and below 200 ppb/wt%. I can see three background Triassic data points that exceed 200 ppb/wt%, and a fourth that gets close. Generally low, yes, but not uniformly.

L. 143: T-Jr is used here, where it is written T-J throughout the rest of the manuscript. Correct to maintain consistency, and check for any other examples.

L. 153: I think 'infer' or 'support' would be better than 'track'. Strictly speaking, the $\delta^{13}\text{C}$ tracks carbon-cycle changes, which are assumed to have been caused by volcanism. But the volcanism (or specifically, the volcanic carbon) is just inferred to have been the source, as the authors themselves state in the following sentence.

L. 169–170: The Hg isotope record from St Audries Bay doesn't really show zero-MIF values, but rather negative ones, as might be expected for such a shallow marine environment that was rather proximal to land. Yes, the MIF gets closer to zero at the extinction interval, but it's certainly not as clear as for the other sites.

L. 176: In both sites, the Precursor CIE is indicated with a question mark, highlighting that this level isn't actually that clearly defined at either location, as I noted above. So is it really true to say that there are three clear negative excursions? The stratigraphy is key to this, and in recent years it has been shown that correctly identifying the TJ CIEs is not always straightforward or without controversy (see Lindström et al., 2017, *Palaeo-3*; and Korte et al., 2019, AGU monograph). I know that there is some information in the supplementary text on this, but even there, the logic seems to be 'here are three apparent $\delta^{13}\text{C}$ excursions around the TJ boundary, they must be the same three as recorded elsewhere'. I think that much more discussion is needed over the other potential influences on sedimentary carbon-isotope compositions and the palynological correlation of these sites with others before that interpretation can be safely arrived at, either in the main manuscript or supplementary text.

L. 179–180: I would be less worried about diagenetic overprinting than the possible influence of organic matter type (and variations thereof) on the $\delta^{13}\text{C}$ values, and indeed Hg contents for that matter. As the authors noted in their responses to the previous set of reviews, there is a complex set of possible influences on $\delta^{13}\text{C}$ trends, yet these are not discussed or even mentioned anywhere here (between lines 174–183).

L. 193: Change 'which was' to 'as'.

L. 208–212: I agree that the Hg MIF values are increasing towards less negative values at Haojiagou, but I'm not convinced that this trend is also present at Qilixia. There is an increase around the proposed PCIE level, but then one or two negative excursions (albeit consisting of only one or two data points) at and just below the Initial CIE level. So that would suggest an increasing dominance of terrestrial mercury around the extinction at that site.

L. 214–215: I think that the MIF values at St Audries Bay might be lower than those shown here. Certainly they are fairly negative.

L. 231–232: On lines 212–214 it was implied that the mercury trends were consistent with an increased flux of volcanically sourced Hg via the atmosphere, but here it is stated that the mercury is documenting increased continental weathering instead. This feels contradictory. Maybe I'm missing or misunderstanding something, but if so, the argument needs to be more clearly explained in the text.

L. 248–252: For me, it would be useful to see these figures in the main text, although I know that the length constraints for Nature Communications are tight. I'm starting to wonder if a slightly longer format journal might be better for this work.

L. 256–258: Yes, I agree with this point that this correlation suggests an elevated terrestrial Hg flux linked to increased chemical weathering. But again, this seems to be inconsistent with the earlier statement that there was evidence for an increased input of atmospheric (volcanic) Hg.

L. 270: On line 231 it was stated that the Hg was also a line of evidence for intensified chemical weathering, and now it isn't listed as one, but is back to being listed as marking volcanism. This is starting to get very confusing.

L. 274: Change to 'are likely to have'.

L. 278–279: 'Based on the current age model...'. What is this age model? If it was published in a previous study, it should be cited. If it's new to this work, either explain it, or indicate which part of the materials/methods or supplementary information an explanation can be found in.

L. 305–308: How does this result compare with Schaller et al.'s 2012 (EPSL) study into the possible drawdown of CO₂ through silicate weathering? I'm surprised that there is no mention of that work here.

L. 403: An analytical precision within ± 2.5 wt% for carbon and ± 5 wt% for sulfur sounds alarmingly imprecise for the material that is being used to check data quality. The carbon content of SDO-1 is only around 10 wt% I believe, so 10 ± 2.5 wt% would be an uncertainty of 25%. I assume that analytical precision within 2.5 % and 5 % of the measured value is actually meant.

Figure 2: I understand why the authors have included the Zhang et al. data in this figure. But for me, it just makes it look messy and harder to really look at the new data. I'd prefer it if the data were shown in a different way, or better still, not shown here, but presented together in a supplementary figure instead.

What does the red field signify in the clay mineralogy diagram? Kaolinite?

Why are the Hg values for Qilixia plotted on a logarithmic scale when no other data (not even the Hg values for Haojiagou) are not? This inconsistency makes it so much harder to compare the datasets from the two sites. Change to a linear scale.

Figure 3A–B: On what basis are the Hg/TOC values for the purple 'volcanic' and blue 'terrestrial' fields assigned? I can't find a reference of this in the main text or figure caption.

REVIEWER COMMENTS

Reviewer #1 (Remarks to the Author):

Dear authors

It is nice to see that you have taken the majority of my comments and suggestions under consideration and made the necessary changes. I think the revised manuscript is greatly improved. However, there are still a few things that need to be considered. You will find these below.

Reply: Many thanks for the positive comments to our revisions. We made modifications as indicated in our responses below.

Best regards

Sofie Lindström

There is something seriously wrong with your Fig. 1, at least in the merged PDF version, where everything except the background palaeomap from Blakey is misplaced.

St. Audrie's Bay is misspelled. Stenlille-1 and Stenlille-4 are actually located very close to each other. At this scale, representing them with one dot (as in N Albert/Albert-1) would be sufficient. Also, specify that you mean Hg-isotopes.

Reply: Many thanks for these comments. We made modifications as follows:

1) We do not know why everything except the paleomap was misplaced in the merged PDF file (in fact, we did not see any misplacements), which is likely to have been due to an error during the merging process. We will save the figures as PDF files in the next submission.

2) We have changed "St Audries Bay" to "St. Audrie's Bay" in figure 1 as well as the main text (line 679) as the reviewer suggested, but we note that the spelling "St Audries Bay" was used in some studies (e.g., Belcher et al., 2010, NG; Percival et al., 2017, PNAS).

3) We show Stenlille-1 and Stenlille-4 as one dot in the new version of figure 1 and modified them in the main text (line 681).

Figures S1 and S2. Hettangian is misspelled.

Reply: Sorry for the typo. We modified "Hetangian" to "Hettangian" in the new version of figures S1 and S2.

Figure S1. For the Qilixia section: Why are the spore-pollen taxa presented only as genera? This is fairly useless information for biostratigraphical purposes.

Reply: Yes, we agree that use of the genera of spore-pollen fossils is not optimal for biostratigraphic correlations. However, it is often not possible to correlate on the basis of spore-pollen species owing to floral endemism and/or to poor preservation of fossils in the Qilixia section (e.g., Li et al., 2020, PPP). For example, the pollen taxa *Lunatisporites rhaeticus* and *Cerebropollenites thiergartii*, which are key palynostratigraphic markers for the Triassic-Jurassic boundary in Europe, are not encountered in the Qilixia section. The palynostratigraphic framework at Qilixia was established mainly based on changes in the relative abundances of key spore-pollen genera (e.g. *Dictyophyllidites*, *Concavisporites*, *Cyathidites*, *Asseretospora*, *Quadraeculina*, *Chasmatosporites*) and the occurrences of some key species (e.g. *Lunzisorites lunzensis*, *Conbaculatisporites pauculus*, *Neoraistrickia taylorii*, *Lycopodiacidites rudis*, *Annulispora* spp., *Kyrtomisoris laevigatus*, *Classopollis* spp.) (Li et al., 2020). We have revised Fig. S1 to present the stratigraphic distributions of selected key spore-pollen species in the Qilixia section based on currently available data. In addition to the palynological fossil data, we correlated this section using carbon isotope profiles and astrochronological constraints based on Milankovitch cycles, as shown in figure S1 and lines 409-422 in the main text as well as Supplementary Note 1 in the supplementary information.

Line 178. The two papers referenced here do not testify to the global recognition of these CIEs. You should refer to a paper that CIEs globally. You could e.g. use: Lindström et al. (2021) or Fujisaki et al. (2018), which presents some different correlations but at least on a global scale.

Reply: Fair point. We added the review paper (Lindström et al., 2021) here and in lines 59, 61, 157, 163, 181.

Line 568. Lindström is misspelled.

Reply: Sorry for the typo. We modified it.

Line 642. St. Audrie's Bay is misspelled. Check throughout!

Reply: See the detailed reply to the above comment. We changed "St Audries Bay" to "St. Audrie's Bay" throughout the Ms.

Reviewer #2 (Remarks to the Author):

This is a second review of the submitted manuscript by J. Shen and co-authors. I would like to thank the authors for their very thorough response to my previous comments. In the revised version, the introduction now clearly states what the current knowledge gaps are, and what this study actually did in order to close them. The discussion is now much easier

to follow, and the different results are better tied together. The section about the LOSCAR modeling is now improved, especially in terms of understanding how the modeling was performed and what the model results suggest. Furthermore, in the first review, all three reviewers were concerned about low TOC concentrations. In the revised version my opinion is that the authors have satisfactorily addressed this, both in the discussion and supplementary file but also by clearly marking the $</>$ 0.2 wt.% TOC data points in Fig. 2. I am happy to recommend this manuscript for publication in Nature Communications.

Reply: Many thanks for the positive comments regarding our revisions to the introduction, discussion, modeling, and other sections of the manuscript. Again, many thanks for your constructive and thorough comments to the Ms, which have improved it significantly.

After reading the revised manuscript I have a few minor comments:

Lines 57-58: I would re-write this sentence. The temporal link strengthens the case for a causal link between CAMP and the ETE, but the temporal link alone is not the only reason why it has been suggested that CAMP was involved in causing the ETE.

Reply: Fair point. We modified this sentence to make the relationship between CAMP and the ETE clearer.

Lines 65-68: I would delete “through weathering of silicate rocks”. Because now you are saying that “...weathering... can... lower atmospheric CO₂ concentrations through weathering”.

Reply: Agreed. We deleted “through weathering of silicate rocks”.

71: I would delete “although it is necessary”.

Reply: Agreed. We deleted “although it is necessary”.

74: I would delete the comma

Reply: Agreed. We deleted the comma.

101: “cycle^v”: delete the v

Reply: Sorry for typo. We delete the “v”.

192: I would write: “...both this and earlier studies document...”

Reply: Modified (line 214).

311: I would delete “from the models”.

Reply: Modified. We delete “from the models”.

Fig. 4: I would write what the dashed horizontal lines represent in the figure caption.

Reply: Good point. We added a statement in the figure caption to show the horizontal dashed lines represent the background values of atmospheric pCO₂ (A) and silicate weathering flux (B) before the eruption of CAMP (lines 727-729).

Table S1: typo: “initial seeady-state pCO₂”

Reply: Sorry for the typo. We change the “seeady” to “steady”.

Reviewer #3 (Remarks to the Author):

This is a revised version of a study investigating the relationship between changes in volcanism, the global carbon cycle, and silicate weathering during the Triassic-Jurassic extinction. I think that the manuscript has been improved since the previous submission, and a number of the points are more clearly made, particularly regarding the LOSCAR modelling. Combining multiple proxies within a single sedimentary record is an important tool for interrogating past episodes of climate change, as it negates the need to stratigraphically correlate across multiple regions. Thus, this study has the potential to be an important contribution to our understanding of the TJ extinction.

Reply: Many thanks for the positive comments regarding our revisions. We agree that combining multiple proxies within a single sedimentary record is valuable, and that the multiproxy analysis of the present study has the potential to be an important contribution to T-J boundary research—especially with regard to an understanding of the cause-and-effect relationships among volcanisms, carbon perturbation, and continental chemical weathering near the TJB. Thanks again for the constructive and thorough comments on our Ms, which have improved it significantly.

However, I still think there are things that need to be addressed with this manuscript. Most unfortunately, I think that there are points raised by both myself and another reviewer in the previous set of comments that have not been addressed to my satisfaction. Two reviewers noted in the last version that the high Hg/TOC peak at Haojiagou is largely driven by very low TOC, and the peaks at Qilixia are at least partly caused by the same issue, but whilst the authors have now added text discussing the host phase of Hg in these sediments, I cannot find a sentence anywhere in the main text simply stating that the TOC is very low in some of the samples that have high Hg/TOC and that this may hinder interpretation of these peaks. There are two short sentences in the supplementary text acknowledging this, but these need to be moved to the main manuscript. Additionally, the authors acknowledged in their response to my previous review that the d13C trends

are complex to interpret. That's fine, and I agree with them. But I would then expect to find some new text in the manuscript outlining these complexities and how the authors have addressed them. But the original claim that there are three clear $\delta^{13}\text{C}$ excursions (albeit one with a question-mark attached) is still present, and the nuances undiscussed.

Reply: Please note that both the first and second reviewers opined that we have satisfactorily addressed this issue:

Reviewer #1: "It is nice to see that you have taken the majority of my comments and suggestions under consideration and made the necessary changes. I think the revised manuscript is greatly improved" (page 2 in this file).

Reviewer #2: "Furthermore, in the first review, all three reviewers were concerned about low TOC concentrations. In the revised version my opinion is that the authors have satisfactorily addressed this, both in the discussion and supplementary file but also by clearly marking the $</>$ 0.2 wt.% TOC data points in Fig. 2." (page 3-4 in this file).

We have nonetheless revised the text concerning the effects of low TOC on Hg/TOC peaks (lines 193-216) and on the complexity of interpretation of organic carbon isotope records (lines 176-192), as suggested by the third reviewer.

There are also further details in the supplementary information that I would prefer to see in the main text. The added information regarding section correlation and palynology requested by another reviewer is mentioned briefly in the Materials and Methods section, but there is important information in the supplementary text that would be good to have in the main manuscript. A number of figures pertaining to the clay-mineralogy evidence of weathering would also be good to have included. I appreciate that there are major space constraints for Nature Communications, and that asking for these details to be added, in addition to those above, would be challenging. I hate to say this, given that this is the 2nd or 3rd iteration of the manuscript and I really would understand the authors' frustration to have to start again, but I wonder if a slightly longer-format journal would allow the complexities with stratigraphic correlation and geochemical interpretations to be better explained, strengthening the arguments presented.

Reply: This is not a valid criticism. The issues about section correlations and clay minerals methods are not the dominant focus of the present work. Although there is no length limit for the Materials and Methods section in NC, citations in the main text are limited to 70. In order to respond to the reviews, we have added statements about section collection in the main text (lines 409-422). Furthermore, we moved Figure S5 to the main text (now figure 4) to support our geochemical interpretations (lines 271-282).

A final thought. The two sites studied here both have age models based on biostratigraphy and astrochronology. There are also robust age models for Levanto and St Audries Bay, which have now also been studied for $\delta^{13}\text{C}$, Hg, and Hg isotopes (I don't think the Yager et al. paper was available online when I last reviewed this manuscript). It

would be interesting to have a figure comparing the trends across the four sections vs time, rather than depth. Perhaps also with the timing of CAMP activity also marked. Just an idea.

Reply: Although it might be interesting to combine the Hg concentrations and isotopes of the four sections together in a single figure, this is not necessary to support our conclusions, and it would unnecessarily expand the present work. First, there are no chemical weathering proxies (e.g., CIA, clay minerals) available for the marine sections at Levanto and St Audrie's Bay, which would limit our ability to evaluate cause-and-effect relationships among volcanic, carbon isotope, and continental chemical weathering, as we have done in our terrestrial study sections. Second, records of volcanic effects on continental weathering are less sensitive in marine systems than on land because many marine depositional processes could alternate the proxies (such as, clay minerals, Hg isotope). We do not think that combining C and Hg data for these four sections into one figure would be useful to support the main conclusions of the present work (i.e., that intensified continental chemical weathering and carbon-cycle perturbations were linked to volcanism during the TJB).

In summary, the novelty of this study is coming across more clearly now than it was in the previous version, and the manuscript is certainly improved. But there are some comments from last time that have not been fully addressed, and I think that further details in the main text are needed regarding some issues. I recommend moderate revisions, largely consisting of more detailed explanations regarding the various complexities outlined above in the main manuscript.

Reply: Many thanks for the positive comments. We have done our best to fully address all comments and issues raised by the reviewers.

MINOR COMMENTS:

L. 37: Change to 'We interpret these results...'. Do not use an unqualified 'this'. Check throughout.

Reply: Fair point. We changed "this" to "these results".

L. 41–43: This initially felt counter-intuitive to me. I thought that lower latitude sites were generally more susceptible to chemical weathering, due to the warmer, and generally wetter, climate. Following a second reading, I assume that the point being made is that at higher latitudes, there will be a greater and more rapid change in temperature, and consequently also in weathering rates. I'd suggest rephrasing to make this point more clear.

Reply: Yes, under a rapid shift of temperature, high-latitude sites are more susceptible to temperature effects than lower latitude sites ("high-latitude amplification", Bekryaev et al.,

2010, Journal of Climate). We have slightly rephrased this sentence to express our meaning more clearly.

L. 54: Change 'as' to 'to have been'.

Reply: Modified. We changed "regarded as" to "thought to have been".

L. 59–60: 'carbon' appears twice within three words, and then a third time later in the same sentence. Rephrase.

Reply: Not necessary—the word "carbon" is used in three different contexts, i.e., "isotopically light carbon", "carbon dioxide" and "organic carbon isotopes".

L. 61: Increased CO₂ concentrations in what? Clarify.

Reply: Modified. We clarified it to "increase CO₂ concentrations in the atmosphere" in line 62.

L. 74: Comma issue. It should either be 'Mercury (Hg) concentrations, and isotopes, are widely used...' or 'Mercury (Hg) concentrations and isotopes are widely used...'.

Reply: Modified. We deleted the comma at both the second the third reviewers' suggestion.

L. 75–76: I'm not sure that it's necessary to describe mercury as a naturally occurring element. I think that what is meant is that elemental mercury occurs naturally in gaseous form in the atmosphere. If this is the case, then I'd suggest rephrasing the start of this sentence to make it clear.

Reply: This sentence has been modified for improved clarity.

L. 80: If we're being honest, it's more true to say that during large

Reply: Apologies, but we are unsure of the reviewer's meaning here.

L. 92–93: I'm still not keen on this phraseology regarding the distality of the two new sites. I agree with the author's response to a previous comment that these sites do give a more global overview of the T-J Hg cycle, and I completely endorse that stance. But why not then simply say that. I.e., '...these records are largely concentrated around central Pangea. However, data from the eastern margin of the Tethys Ocean, which would enable a truly global-scale overview, are lacking.'

Reply: Modified. We rephrased the sentence as the reviewer suggested.

L. 97–98: Does this not depend on the specific volcanic effect? I agree that the terrestrial realm will be more impacted by volcanic warming, but presumably other effects such as aquatic acidification (heavily implicated in the TJ extinction) will be at least as equally strong in the marine realm.

Reply: Agreed. We have modified this statement to make it clear that we are referring mainly to the effects of warming (see lines 97-100).

L. 116: This statement is not actually true. If these three shifts in $\delta^{13}\text{C}$ at Haojiagou are considered to be excursions, then there are also excursions at 250–360 m and arguably below -100 m and -80 to -50 m. The point is that these are the three excursions (although the PCIE is dubious) around the TJ boundary.

Reply: Fair point. Yes, we meant the three carbon excursions that are stratigraphically proximal to the TJB. We modified this sentence to make this point clear (lines 117-119).

The same is true of Qilixia. One could certainly argue that there is an excursion at around 150 m.

Reply: Fair point. Yes, we meant the three carbon excursions that are stratigraphically proximal to the TJB. We modified this sentence to make this point clear (lines 135-138).

L. 118–119: It looks like several samples have Hg contents above 60 ppb throughout the section, with a few peaks of around 80–120 ppb, as the authors say. So I'm not sure why a 'range from 10 to 60 ppb' is stated.

Reply: Modified. We changed to "Mercury concentrations (Hg) range from 3 to 101 ppb and....." in lines 119-121.

L. 120–123: Yes, these Hg/TOC peaks are present, but nowhere is it stated that the higher values within the proposed Initial CIE and Main CIE strata are from samples featuring very low TOC. I know that this is shown in Figure 2, and briefly in the supplementary information, but this is an important caveat to the peaks that should be explicitly mentioned in the main text, probably together with a citation of Grasby et al.'s 2016 recommendation of not normalizing against such low TOC values.

Reply: This criticism is not valid. We examined the effect of low TOC on the Hg/TOC peaks in lines 188-195 of the first revision. We have now modified this text further to more clearly show that low TOC is not controlling the pattern of secular Hg/TOC variation (see lines 193-216). We cited Grasby et al., 2016 in line 200.

L. 133–136: From looking at Figure 2, it appears more like the uppermost Triassic 'background' values range from -27 to -24 per mil, rather than -25 to -23. The proposed PCIE excursion has values that fall entirely within that range. The other two CIEs are more

convincing. But what concerns me most is the huge gaps in the upper Triassic dataset. How do we know that there aren't other variations in $\delta^{13}\text{C}$ within those gaps, which would make the proposed TJ excursions (especially the PCIE) look like noise? I appreciate that samples aren't always attainable, but this is a big limitation with the dataset that the authors simply must acknowledge.

Reply: We changed the background values of organic C isotopes from -25 to -23 to “-27 to -24”. As we stated in the main text, controls on variation of organic carbon isotope records are complex for terrestrial sediments (lines 177-181). We identified the PCIE based on relative negative excursions (-25.5 ‰ to -24.5 ‰ at 283.21 to 300.91 m) than its nearby underlying (-24.5 ‰ to -23.5 ‰ at 279.81 to 282.41 m) and overlying (-23 ‰ to -23.5 ‰ at 302.11 to 308.69 m) strata, as shown in figure 2G. We marked “?” next to the PCIE and added a statement (lines 179-192) to address the uncertainty in its identification.

Many thanks for acknowledging that samples are not always obtainable. We do not think that the gap (ranging from 220.22-279.81 m in a dominantly sandstone interval) would affect interpretation of the T-J boundary excursion largely because our record includes ~ 3 m of background strata (279.81 to 282.41m) between this gap and the PCIE interval.

L. 136: ‘Mercury contents are lower at the upper Triassic’. It should be ‘in the upper Triassic sediments’ (or strata, either will do). And what are the Hg contents lower than? Clarify.

Reply: This is unnecessary. Our native English-speaking coauthor, Dr. Thomas Algeo, assures us that “Upper Triassic” is sufficient. We meant that the Hg contents are lower than 400 ppb, as given in line 140.

L. 139–141: Again, there is no mention of the fact that some (although not all, for this site) of the high Hg/TOC values are from samples featuring very low TOC contents.

Reply: We discussed the influence of low TOC on Hg/TOC values in lines 193-216. See our detailed response to one of the comments above.

Also, it is not actually true to say that the background values are ‘uniformly’ low and below 200 ppb/wt%. I can see three background Triassic data points that exceed 200 ppb/wt%, and a fourth that gets close. Generally low, yes, but not uniformly.

Reply: Fair point. We changed “uniformly” to “generally”.

L. 143: T-Jr is used here, where it is written T–J throughout the rest of the manuscript. Correct to maintain consistency, and check for any other examples.

Reply: Sorry for the typo. Modified.

L. 153: I think 'infer' or 'support' would be better than 'track'. Strictly speaking, the d13C tracks carbon-cycle changes, which are assumed to have been caused by volcanism. But the volcanism (or specifically, the volcanic carbon) is just inferred to have been the source, as the authors themselves state in the following sentence.

Reply: Agreed. We changed "track" to "infer".

L. 169–170: The Hg isotope record from St Audries Bay doesn't really show zero-MIF values, but rather negative ones, as might be expected for such a shallow marine environment that was rather proximal to land. Yes, the MIF gets closer to zero at the extinction interval, but it's certainly not as clear as for the other sites.

Reply: Fair point. At St. Audrie's Bay, the MIF values are significantly negative (-0.46 ‰ to -0.24 ‰) for the background, but less negative values (-0.17 ‰ to -0.07 ‰) were associated with the Hg peaks near the TJB. We modified "Near-zero" to "Zero or near-zero". "Near-zero" encompasses values that are both a little higher and a little lower than zero.

L. 176: In both sites, the Precursor CIE is indicated with a question mark, highlighting that this level isn't actually that clearly defined at either location, as I noted above. So is it really true to say that there are three clear negative excursions? The stratigraphy is key to this, and in recent years it has been shown that correctly identifying the TJ CIEs is not always straightforward or without controversy (see Lindström et al., 2017, Palaeo-3; and Korte et al., 2019, AGU monograph). I know that there is some information in the supplementary text on this, but even there, the logic seems to be 'here are three apparent d13C excursions around the TJ boundary, they must be the same three as recorded elsewhere'. I think that much more discussion is needed over the other potential influences on sedimentary carbon-isotope compositions and the palynological correlation of these sites with others before that interpretation can be safely arrived at, either in the main manuscript or supplementary text.

Reply: We agree that the magnitude and phase of the CIEs near the TJB remain in debate. However, identification and correlation of the ICIE and MCIE in the study sections is not in doubt (as the reviewer stated on page 9-10 in this file). The existence and significance of the PCIE is less certain, but we have indicated this fully by adding a "?" after "PCIE" in figures 2, S2, S3 and discussing its uncertainty in lines 176-192.

L. 179–180: I would be less worried about diagenetic overprinting than the possible influence of organic matter type (and variations thereof) on the d13C values, and indeed Hg contents for that matter. As the authors noted in their responses to the previous set of reviews, there is a complex set of possible influences on d13C trends, yet these are not discussed or even mentioned anywhere here (between lines 174–183).

Reply: Agreed. We added a discussion of the complexity of controls on organic carbon isotope variation in lines 176-192.

L. 193: Change 'which was' to 'as'.

Reply: Modified.

L. 208–212: I agree that the Hg MIF values are increasing towards less negative values at Haojiagou, but I'm not convinced that this trend is also present at Qilixia. There is an increase around the proposed PCIE level, but then one or two negative excursions (albeit consisting of only one or two data points) at and just below the Initial CIE level. So that would suggest an increasing dominance of terrestrial mercury around the extinction at that site.

Reply: Not exactly. We meant the increase of Hg-MIF to less negative values during the mercury-enriched interval (e.g., from the PCIE to the MCIE, but not just near the ICIE) relative to the Upper Triassic background values as shown in figure 3 A and B. The relatively positive values around the PCIE (0.07 ‰, 0.08 ‰, 0.08 ‰ at 295.11, 296.91, and 298.91 m, respectively), between the ICIE (-0.06 ‰, and -0.03 ‰ at 306.69 and 308.69 m, respectively) and the MCIE (0.02 ‰, -0.04 ‰, and -0.04 ‰ for 327.78, 337.78, and 341.28 m, respectively) document greater volcanic inputs of Hg. Yes, as the reviewer suggested, the negative excursions at and just below the ICIE are evidence of greater terrestrial inputs of Hg. This variation of Hg-MIF documents the different source of Hg through the T-J transition. We changed "during the T-J transition" to "during the mercury-enriched interval"

L. 214–215: I think that the MIF values at St Audries Bay might be lower than those shown here. Certainly they are fairly negative.

Reply: Yes, the MIF values are significantly negative (-0.46 ‰ to -0.24 ‰) at St. Audrie's Bay, but less negative values (-0.17 ‰ to -0.07 ‰) were associated with the Hg peaks near the TJB. We made this clear in lines 236-239.

L. 231–232: On lines 212–214 it was implied that the mercury trends were consistent with an increased flux of volcanically sourced Hg via the atmosphere, but here it is stated that the mercury is documenting increased continental weathering instead. This feels contradictory. Maybe I'm missing or misunderstanding something, but if so, the argument needs to be more clearly explained in the text.

Reply: This is not a valid criticism because our statements are not contradictory. The Hg concentration and isotope data for our study sections provide evidence of both volcanic and terrestrial inputs. Hg isotopes, especially $\Delta^{199}\text{Hg}$ values, differ between terrestrial (negative values) and volcanism-related atmospheric sources (near zero or positive values). Integrated use of Hg concentrations and isotopes can help to track Hg sources in

sediments. In the present study, the $\Delta^{199}\text{Hg}$ values within the high-Hg interval can be divided into two groups (Fig. 3A, 3B), one group having $\Delta^{199}\text{Hg}$ values similar to the background samples, which imply a common source of Hg (i.e., terrestrial) (blue arrow dashed lines in Fig. 3A, 3B). The other group has more positive $\Delta^{199}\text{Hg}$ values (close to or a little higher than zero), which is evidence of atmosphere Hg inputs, likely from a volcanic source (purple arrow dashed lines in Fig. 3A, 3B). We made this clear in lines 226-241 and 256-270.

L. 248–252: For me, it would be useful to see these figures in the main text, although I know that the length constraints for Nature Communications are tight. I'm starting to wonder if a slightly longer format journal might be better for this work.

Reply: This is not a fair comment. Although the A-CN-K diagram is not essential to the main conclusions of our study, we moved this figure (Figure 4) to the main text from the supplementary material as the reviewer suggested. We do not agree that “a slightly longer format journal might be better for this work”. We formatted the Ms in good shape for NC.

L. 256–258: Yes, I agree with this point that this correlation suggests an elevated terrestrial Hg flux linked to increased chemical weathering. But again, this seems to be inconsistent with the earlier statement that there was evidence for an increased input of atmospheric (volcanic) Hg.

Reply: This is not a valid criticism. See replies to similar comments above (page 12 in this file).

L. 270: On line 231 it was stated that the Hg was also a line of evidence for intensified chemical weathering, and now it isn't listed as one, but is back to being listed as marking volcanism. This is starting to get very confusing.

Reply: This is not a valid criticism. See replies to similar comments above (page 12 in this file).

L. 274: Change to 'are likely to have'.

Reply: Modified.

L. 278–279: 'Based on the current age model...'. What is this age model? If it was published in a previous study, it should be cited. If it's new to this work, either explain it, or indicate which part of the materials/methods or supplementary information an explanation can be found in.

Reply: Fair point. We added “see materials and methods” here.

L. 305–308: How does this result compare with Schaller et al.'s 2012 (EPSL) study into

the possible drawdown of CO₂ through silicate weathering? I'm surprised that there is no mention of that work here.

Reply: Our model result (1-3 Myr) is in agreement with the timeframe (~ 1.5 Myr) of CO₂ concentrations falling to background values after the final CAMP eruption that was published by Schaller et al. (2012). We added a statement and cited this reference in lines 329-332.

L. 403: An analytical precision within ± 2.5 wt% for carbon and ± 5 wt% for sulfur sounds alarmingly imprecise for the material that is being used to check data quality. The carbon content of SDO-1 is only around 10 wt% I believe, so 10 ± 2.5 wt% would be an uncertainty of 25%. I assume that analytical precision within 2.5 % and 5 % of the measured value is actually meant.

Reply: Sorry for the misunderstanding. We meant the analytical precision within 2.5% and 5% of measured values. We made this clear in lines 439-441.

Figure 2: I understand why the authors have included the Zhang et al. data in this figure. But for me, it just makes it look messy and harder to really look at the new data. I'd prefer it if the data were shown in a different way, or better still, not shown here, but presented together in a supplementary figure instead.

Reply: Modified. We removed Zhang's data to the Figure S2 as the reviewer suggested.

What does the red field signify in the clay mineralogy diagram? Kaolinite?

Reply: Sorry for the missing explanation. Yes, the red field in columns F and M represents kaolinite. We have added this to figure 2.

Why are the Hg values for Qilixia plotted on a logarithmic scale when no other data (not even the Hg values for Haojiagou) are not? This inconsistency makes it so much harder to compare the datasets from the two sites. Change to a linear scale.

Reply: Modified. We used a logarithmic scale for QLX due to the large variation of Hg values (three orders of magnitude), which range from 1 ppb to 1837 ppb. We changed the logarithmic scale to a linear scale for Hg values in QLX (column H) as suggested.

Figure 3A–B: On what basis are the Hg/TOC values for the purple 'volcanic' and blue 'terrestrial' fields assigned? I can't find a reference of this in the main text or figure caption.

Reply: We have added an explanation of the basis for the assigned values in lines 702-709. The volcanic endmember is based on the most positive values of MIF ($\Delta^{199}\text{Hg} = \sim 0.1$ ‰) and most elevated Hg/TOC values (~ 600 - 800 ppb/wt.%) in the study sections, which likely reflect dominant volcanic influence. For the terrestrial endmember, we

assumed values based on the maximum Hg/TOC (~ 300 ppb/wt.% and 1000-1200 ppb/wt.% for HJG and QLX, respectively) and $\Delta^{199}\text{Hg}$ values similar to the background (i.e., -0.3‰ to -0.4‰ and -0.2‰ to 0‰ for HJG and QLX, respectively).